# Earliest evidence of elephant butchery at Olduvai Gorge (Tanzania) reveals the evolutionary impact of early human megafaunal exploitation

Manuel Dominguez-Rodrigo[1,2]*, Enrique Baquedano[1,3], Abel Moclan[1,4], David Uribelarrea[5], José Ángel Correa-Cano[1,3], Fernando Diez-Martin[1,6], Alejandro Velazquez-tello[1,3], Elia Organista[1,7], Eduardo Mendez-Quintas[1,8,9], Marina Vegara-Riquelme[1,10], Agness Gidna[11], Audax Mabulla[12]

[1]Institute of Evolution in Africa (IDEA), Rice University, Archaeological and Paleontological Museum of Madrid and Foundation of the University of Alcalá, Madrid, Spain; [2]Department of Anthropology, Rice University, Houston, United States; [3]Archaeological and Paleontological Museum of Madrid, Madrid, Spain; [4]PALEVOPRIM laboratory, UMR CNRS 7262, and Université de Poitiers, Poitiers, France; [5]Department of Geodynamics, Stratigraphy and Paleontology, Complutense University, Madrid, Spain; [6]Department of Prehistory and Archaeology, University of Valladolid, Valladolid, Spain; [7]Osteoarkeologiska forskningslaboratoriet och antikens kultur, Stockholms Universitet, Stockholm, Sweden; [8]Universidade de Vigo, MAPAS Lab, Vigo, Spain; [9]UNIARQ – Centre for Archaeology, School of Arts and Humanities, University of Lisbon, Lisbon, Portugal; [10]Area of Prehistory, Department of History and Philosophy, University of Alcalá de Henares, Alcalá de Henares, Spain; [11]Ngorongoro Conservation Area Authority, Department of Cultural Heritage, Karatu, United Republic of Tanzania; [12]Department of Archaeology, University of Dar es Salaam, Dar es Salaam, United Republic of Tanzania

*For correspondence:
mdr@rice.edu

Competing interest: The authors declare that no competing interests exist.

## eLife Assessment

In this **valuable** study, the authors present traces of bone modification on ~1.8 million-year-old proboscidean remains from Tanzania, which they infer to be the earliest evidence for stone-tool-assisted megafaunal consumption by hominins. Challenging published claims, the authors argue that persistent megafaunal exploitation roughly coincided with the earliest Acheulean tools. Notwithstanding the rich descriptive and spatial data, the behavioral inferences about hominin agency rely on traces (such as bone fracture patterns and spatial overlap) that are not unequivocal; the evidence presented to support the inferences thus remains **incomplete**. Given the implications of the timing and extent of hominin consumption of nutritious and energy-dense food resources, as well as of bone toolmaking, the findings of this study will be of interest to paleoanthropologists and other evolutionary biologists.

**Abstract** The role of megafaunal exploitation in early human evolution remains debated. Occasional use of large carcasses by early hominins has been considered by some as opportunistic, possibly a fallback dietary strategy, and for others a more important survival strategy. At Olduvai Gorge, evidence for megafaunal butchery is scarce in the Oldowan of Bed I but becomes more

frequent and widespread after 1.8 Ma in Bed II, coinciding with the emergence of Acheulean technologies, but not functionally related to the main Acheulian tool types. Here, we present the earliest direct evidence of proboscidean butchery, including a newly documented elephant butchery site (EAK). This shift in behavior is accompanied by larger, more complex occupation sites, signaling a profound ecological and technological transformation. Rather than opportunistic scavenging, these findings suggest a strategic adaptation to megafaunal resources, with implications for early human subsistence and social organization. The ability to systematically exploit large prey represents a unique evolutionary trajectory, with no direct modern analogue, since modern foragers do so only episodically.

## Introduction

Spatially concurrent lithic tools and proboscidean bones have a long history in human evolution. Such concurrency (sometimes in conjunction with other variables) has been frequently interpreted as the result of hominin butchery (*Adrien Hannus, 2018*; *Agam and Barkai, 2018*; *Agam and Barkai, 2016*; *Altamura et al., 2020*; *Barkai, 2019*; *Ben-Dor and Barkai, 2020*; *Berthelet and Chavaillon, 2001*; *Boschian and Saccà, 2015*; *Chavaillon et al., 1987*; *Chavaillon and Berthelet, 2004*; *Delagnes et al., 2006*; *Gaudzinski et al., 2005*; *Gaudzinski-Windheuser et al., 2023a*; *Gaudzinski-Windheuser et al., 2023b*; *Gomez et al., 1978*; *Goren-Inbar et al., 1994*; *Haynes, 2022*; *Konidaris et al., 2021*; *Leakey, 1971*; *Lemorini et al., 2023*; *Lev and Barkai, 2016*; *Mussi and Villa, 2008*; *Reshef and Barkai, 2015*; *Rocca et al., 2021*; *Saccà, 2012*; *Santucci et al., 2016*; *Solodenko et al., 2015*; *Villa et al., 2005*; *Yravedra et al., 2019*; *Yravedra et al., 2010*; *Rabinovich et al., 2012*). Spatial concurrency is insufficient to establish a causal relationship between tools and megafaunal remains. Given the time-averaged nature of a large part of the Paleolithic record, the deposition of tools and elephants could have occurred sequentially and unrelatedly in several sites; especially taking into account that most of these sites were in paleo-floodplains where elephants and hippos were intensively active and formed natural taphocoenoses. Examples of this abound in the paleobiological record. For example, at BK (upper Bed II, Olduvai Gorge, Tanzania), some *Pelorovis* carcasses were naturally deposited on the same alluvial bar and among the fauna processed by hominins, thereby creating an artificial spatial concurrence between independent depositional events and agencies (*Organista et al., 2016*). In the Middle Pleistocene deposits of Torralba and Ambrona (Spain), artifact-littered areas occur in spatial association with naturally deposited elephant carcasses, only a few of which were also exploited to a limited degree by hominins, as demonstrated by a few proboscidean remains bearing cut marks (*Villa, 1990*; *Villa et al., 2005*). Post-depositional/sedimentation processes also might have artificially created associations of faunal remains and unrelated lithic artifacts (*Villa, 1990*; *Villa et al., 2005*).

Direct taphonomic evidence of hominin exploitation of megafauna (In this study, megafauna refers to animals with a body mass exceeding 800 kg) (in the form of cutmarked or dynamically-percussed bone) has been so far the main argument to establish a causal link between hominin agency and megafaunal remains (*Tappen et al., 2022*). If restricted to the Early Pleistocene, most sites containing partial carcasses of megafauna (e.g. elephant, hippopotamus) either show poor preservation, do not exhibit any of these anthropogenic traces, or these modifications cannot be securely identified (*Delagnes et al., 2006*; *Domínguez-Rodrigo et al., 2007*; *Haynes, 2022*; *Haynes, 1991*; *Isaac and Isaac, 2023*; *Leakey, 1971*). In contrast, for the Middle and Upper Pleistocene, there is more abundant evidence of hominin butchery of megafaunal remains, especially of elephants (*Gaudzinski-Windheuser et al., 2023a*; *Gaudzinski-Windheuser et al., 2023b*; *Haynes, 2022*; *Yravedra et al., 2010*; *Rabinovich et al., 2012*); see review of taphonomic evidence in *Agam and Barkai, 2018*, *Haynes, 2022* and *Konidaris et al., 2021*. The question that we address here is if such a behavior has its roots in the early Pleistocene.

Once hominin agency is empirically demonstrated at taphonomically-supported butchery sites, a second inferential step is attributing carcass butchery to hunting or scavenging. Occasionally, the occurrence of embedded flint in proboscidean remains has been taken as direct evidence of predatory behaviors (*Agam and Barkai, 2018*). This evidence is marginal and chronologically restricted to the late Upper Pleistocene with composite hafted tools. Recently, a supportive argument used to detect proboscidean hunting lies on the intensity of cut marks on carcasses and mortality profiles dominated by male prime adults (*Gaudzinski-Windheuser et al., 2023a*; *Gaudzinski-Windheuser*

**Table 1.** African Early Pleistocene archaeological sites with proboscidean carcasses containing potential evidence of hominin involvement.

| | Area excavated (m2) | N° bones | N° lithics | Megafaunal taxon | Other non megafauna | Age (Ma) | Green breaks | Cut marks |
|---|---|---|---|---|---|---|---|---|
| Nyayanga 3 (Kenya) | 49 | 1580*(241 Hippo) | 42† | Hippo | Present | 3–2.6 | Unreported | Ambiguous |
| Nyayanga 5 (Kenya) | 24 | 196* | 14† | Hippo | Present | 3–2.6 | Unreported | Ambiguous |
| FLK North 6 (Tanzania) | 37 | 400 | 123 ‡ | Elephant | Present | 1.8 | Absent | Ambiguous |
| FLK North Deinotherium site (Tanzania) | - | | 23 | Elephant | Present | 1.7 | Absent | Absent |
| Barogali (Djibouti) | 37 | - | 569 | Elephant | Absent | 1.6–1.3 | Unreported | Absent |
| Nadung´a (Kenya) | 53 | 65 | 6797 | Elephant | Present | 1–0.7 | Unreported | Absent |

*Numbers in main text and appendix tables do not coincide.
†Only artifacts reported as associated to the skeletal remains (total in both excavations is 135).
‡Excluding purported manuports.

et al., 2023b). Prime adult mortality is documented also in other natural palimpsests (Haynes, 1991), in some cases probably caused by fights leading also to broken tusks (Villa and d'Errico, 2001). Ethnoarchaeological observation of modern foragers butchering elephants and hippos shows that they leave very few cut marks after bulk defleshing (Crader, 1983). Modern elephant butchery after culling events (based on a sample of >500 carcasses) also shows that bulk defleshing virtually leaves no traces on bones (Haynes and Krasinski, 2021). Only butchery aiming at the extraction of meat scraps after bulk defleshing imparts most of the cut marks documented experimentally (Haynes and Krasinski, 2021). This casts doubts on whether cutmarking intensity relates to hunting, scavenging, or a maximization strategy feasible in all scenarios. To complicate things more, there is a clear mismatch between the butchery intensity documented at several sites and the resulting tool kit associated with it. Some Middle Pleistocene elephant carcasses display abundant cut marks, but there are very few or no stone tools associated with them (Gaudzinski-Windheuser et al., 2023a), whereas at other sites elephant carcasses exhibit not a single cut mark, but they occur with hundreds or thousands of associated tools (Chavaillon et al., 1987; Delagnes et al., 2006; Yravedra et al., 2010; Table 1). In the latter case, it is probable that in some cases the bulk of tools may have been used to process other non-megafaunal fauna. In the particular case of the Nadung´a elephant (Kenya, 1–0.7 Ma), that explanation for the presence of lithics is improbable since the only other mammal carcass remains found are from one bovid and one suid, most likely a background scatter associated with the bird, fish, and reptile remains with which they are spatially associated. The question remains as to why such a divergent range of tool sets (both in quantity and typological composition) accompany similar partial megafaunal skeletal remains if they are truly functionally related to them.

To make interpretations even more challenging, many butchery sites exhibit fairly complete and (semi)articulated carcasses with limb bones intact. In some of these instances, elements were dismembered but not demarrowed or degreased (Gaudzinski-Windheuser et al., 2023a). Defleshing or demarrowing/degreasing does not require disarticulating bones, and the energy costs involved in cutting through tendons and ligaments of a proboscidean carcass are better explained as a maximizing strategy aiming at reducing the costs of element transportation (neither defleshing nor demarrowing/degreasing requires disarticulation), and/or further exploitation of within-bone fat resources. None of this is clearly documented in those sites. Fat was a crucial resource during human evolution (Ben-Dor and Barkai, 2024; Ben-Dor et al., 2011; Reshef and Barkai, 2015; Solodenko et al., 2015), and the large fat deposits stored in proboscidean carcasses rationally would have been a target of prehistoric butchery, especially in strategies of carcass maximization (as documented in the abundant cut mark record at some megafaunal butchery sites). The question of why our ancestors chose to maximize flesh extraction from proboscideans, and yet frequently abandoned their fat deposits untouched remains unanswered. Modern Bisa people (Zambia) butcher elephants and do not exploit their fat, but they follow a 'satisficing' strategy (Haynes and Krasinski, 2021), through bulk defleshing (barely impacting bone surfaces; Crader, 1983), and not a 'maximizing' strategy, as suggested for

some highly anthropogenically impacted archaeological proboscidean sites (*Gaudzinski-Windheuser et al., 2023a*). It could be argued that the meat of healthy elephants contains already a high amount of fat and that other anatomical areas (like the podal pads) contain extremely nutritious fat, or that the relevance of fat exploitation varies according to latitude and climate, but if the strategy was to maximize food resources, long bone medullary contents should also have been consumed (*Ben-Dor et al., 2011*; *Haynes et al., 2021*), as documented in sites like Castel di Guido (Italy), Preresa (Spain; *Boschian and Saccà, 2015*; *Yravedra et al., 2012*), or BK (see below). Megafaunal long bone green breakage (linked to continuous spiral fractures on thick cortical bone accompanied by modification [in some cases, intensive] of cortical and medullary surfaces of breakage planes) is probably a less ambiguous trace of butchery than 'cut marks', since many of the latter could be equifinal and harder to identify in contexts of moderate to poor preservation, especially in contexts of high abrasion and trampling (*Haynes et al., 2021*; *Haynes et al., 2020*). For example, the purported cut-marked pelvis from Fuentenueva 3 (*Yravedra et al., 2024*) exhibits intense biostratinomic—in addition to diagenetic—modifications impacted by abrasive processes, which renders the pristine experimental reference collection inadequate to confidently classify it as human-made. This is further supported by the fact that the experimental trampling collection used depicts trampling marks as extremely shallow and with wide divergent walls. This does not include the large diversity of morphologies generated by trampling marks, whose cross-sections frequently are indistinguishable from stone-tool cut marks (*Domínguez-Rodrigo et al., 2010b*). Given the abrasive sedimentary context, the abundance of abrasion marks on the affected specimens, and the lack of description of microscopic features that could unambiguously be used to identify those traces as human-made, we remain skeptical of their anthropogenic nature.

Ideally, cut marks should complement green breakage patterns to identify butchery episodes, but the virtual lack of cut marks in 'satisfying' proboscidean exploitation renders their presence/absence non-indicative of the type of anthropic action. Despite the occurrence of green fractures on naturally broken bones, such as those trampled by elephants (*Haynes et al., 2020*), and those occurring through traumatic fracturing or gnawed by carnivores (*Haynes and Hutson, 2020*), these fail to reproduce the elongated, extensive, or helicoidal spiral fractures (uninterrupted by stepped sections), accompanied by the overlapping conchoidal scars (both cortical and medullary), the reflected scarring, the inflection points, or the impact hackled break surfaces and flakes typical of dynamic percussive breakage. Evidence of this type of green breakage had not been documented earlier for the Early Pleistocene proboscidean or hippopotamid carcasses, beyond the documentation of flaked bone with the purpose of elaboration of bone tools (*Backwell and d'Errico, 2004*; *Pante et al., 2020*; *Sano et al., 2020*). Such type of dynamic breaking, in contrast, has been documented in *Pelorovis* and *Sivatherium* carcasses at the 1.3 Ma site of BK (Olduvai Gorge; *Domínguez-Rodrigo et al., 2014a*; *Labrado, 2017*; *Organista et al., 2016*). One reviewer suggested that such variables have also been documented in non-anthropogenic contexts, including helicoidal spiral fractures attributed to trampling or carnivore activity (*Haynes, 1983*), adjacent or flake-like scars produced by carnivore gnawing (*Villa and Bartram, 1996*), hackled fracture surfaces allegedly resulting from heavy passive breakage such as trampling or sediment pressure (*Haynes, 1983*), and 'impact-like' bone flakes documented in carnivore-broken bones. However, this interpretation is epistemologically problematic because it does not satisfy the fundamental criteria for valid analogy as outlined by *Bunge, 1981*, namely substantial, structural, and environmental affinity. Specifically, the cited examples involve agents, materials, and contexts that differ markedly in composition, mechanical properties, and loading regimes from those considered here. Experimental and actualistic studies demonstrate that carnivores—rather than trampling—are also capable of producing spiral fractures and overlapping bone scarring, but these observations are restricted to faunal remains of substantially smaller body size than elephants, which carnivores can gnaw (*Haynes, 1983*; see also *Appendix 1—figures 30–36*). To date, no carnivore has been documented as producing comparable fracture morphologies or surface damage on late juvenile and adult elephant bones. Consequently, the proposed analogy is not supported. Moreover, *Haynes, 1983* provides no empirical evidence that sediment pressure or trampling can generate hackled fracture surfaces. Such features are instead associated with dynamic loading conditions, whereas passive breakage processes have not been shown to produce these types of modifications (*Lyman, 2006*). This reasoning also applies to impact flakes on elephant limb bones, which can only be produced by the sole modern agent documented to dynamically fracture non-infantile green proboscidean long bones: humans.

Regardless of the type of butchery evidence—and with the taphonomic caveat that no unambiguous evidence exists to confirm that megafaunal carcasses were hunted or scavenged other than hominins accessed them in different taphonomically defined stages (i.e. early or late)—the principal reasons for exploring megafaunal consumption in early human evolution are its origin, its episodic or temporally patterned occurrence, its impact on hominin adaptation to certain landscapes, and its reflection on hominin group size and site functionality. If hominins actively sought the exploitation of megafauna, especially if targeting early stages of carcass consumption, the recovery of an apparent surplus of resources reflects a substantially different behavior from the small-group/small-site pattern documented at several earlier Oldowan anthropogenic sites (*Domínguez-Rodrigo et al., 2019*) -or some modern foragers, like the Hadza, who only exploit megafaunal carcasses very sporadically, mostly upon opportunistic encounters (*Marlowe, 2010*; *O'Connell et al., 1992*; *Wood, 2010*; *Wood and Marlowe, 2013*). Determining when the process of becoming megafaunal commensal started has major implications for human evolution, since it has clear implications for our understanding of past social group sizes and social dynamics.

The multiple taphonomic biases intervening in the palimpsestic nature of most of these butchery sites often prevent the detection of the causal traces linking megafaunal carcasses and hominins. Functional links have commonly been assumed through the spatial concurrence of tools and carcass remains; however, this perception may be utterly unjustified as we argued above. Functional association of both archaeological elements can more securely be detected through objective spatial statistical methods. This has been argued to be foundational for heuristic interpretations of proboscidean butchery sites (*Giusti, 2021*). Such an approach removes ambiguity and provides a better argument for spatial functional association, as demonstrated at sites like Marathousa 1 (*Konidaris et al., 2018*) or TK *Sivatherium* (*Panera et al., 2019*). This method will play a major role in the present study.

Here, we present the discovery of a new elephant butchery site (Emiliano Aguirre Korongo, EAK), dated to 1.78 Ma, from the base of Bed II at Olduvai Gorge (Tanzania). It is potentially the oldest unambiguous proboscidean butchery site at Olduvai. In combination with its study, we have carried out extensive survey at a landscape scale on one stratigraphic section of Bed I (Oldowan) and three stratigraphic sections of Bed II: lower Bed II-Lower Augitic Sands (1.8.1.69 Ma), upper middle Bed II (from tuff IIB to IID, 1.5 Ma), and upper Bed II (overlying Tuff IID, 1.3 Ma), which are known to contain Acheulian materials. The goal was to find megafaunal fossils bearing traces of hominin modification across the landscape. Starting at 1.78 Ma, Olduvai Gorge records extensive evidence of hominin involvement with megafaunal carcasses (*Domínguez-Rodrigo et al., 2014c*; *Domínguez-Rodrigo et al., 2009b*), namely Proboscidea and Hippopotamidae, followed closely by Giraffidae and big Bovidae. The nature of this interaction is elaborated and analyzed in this paper. We have used a spatial approach, combined with a more general taphonomic analysis, and an analysis of hammerstone-broken bone specimens occurring widespread on the three targeted Bed II stratigraphic intervals. Also provided is a technological study of the lithic artifacts.

## Results

### The Emiliano Aguirre Korongo (EAK) site

EAK is located in the area of the confluence of the two gorges, next to geo-locality 45 a, between the FLK-N and FLK-NN sites (*Hay, 1976*; *Figure 1*). Stratigraphically, the archaeological remains rest on Tuff IF (1.78 M.a), that is at the beginning of Bed II (*Deino, 2012*). As in the whole area of the gorge, the lowermost Bed II is similar to the uppermost Bed I, which is composed of clays deposited in the margins of the lake. The drying period that characterizes Bed II had not yet taken place during this moment. The sedimentary context of the Olduvai basin in this area is dominated by low energy processes, with predominantly clay deposition, controlled by small seasonal rises and falls in lake level. The lake had a very high pH and high salinity in the central zone (*Deocampo et al., 2017*), but this value is lowered at the margins by the arrival of runoff water. EAK is located roughly coinciding with the FLK fault zone (*Figure 1*). The bone and lithic remains lie on a thin layer of clay (<5 cm) on top of tuff IF. They are also covered by the lacustrine clay of the lowermost Bed II. It is therefore the oldest documented archaeological site formed in Bed II in Olduvai.

During the recent Holocene, the EAK site was affected by a small landslide which displaced the site 6.25 m vertically and approximately 12 m downslope. Although the stratigraphic sequence remains

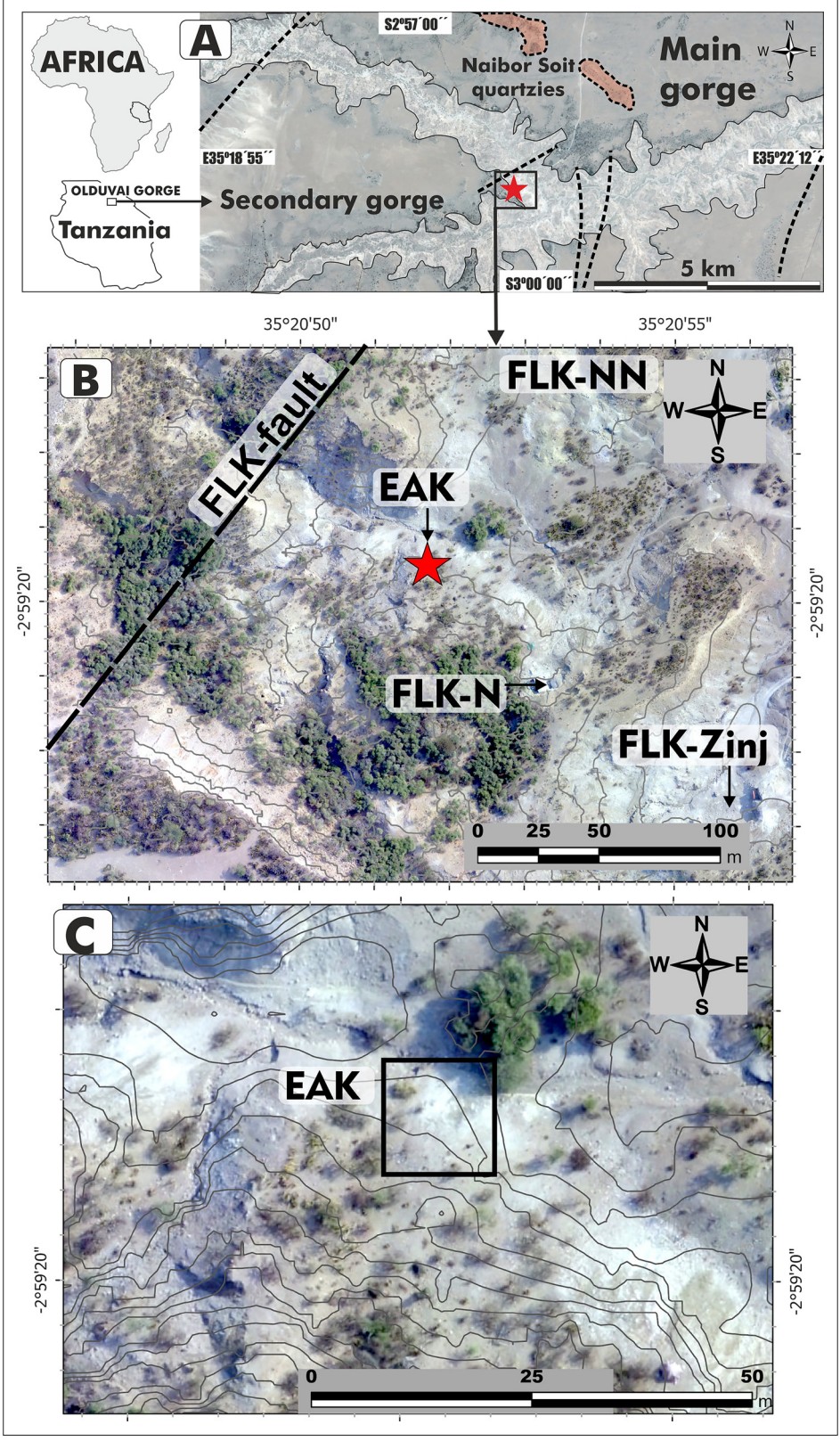

**Figure 1.** Location of EAK in the junction of the main and secondary branches of Olduvai Gorge, main faults and Naibor Soit quartzites (**A**), within the area where the Bed I sites cluster at the junction (**B**), and the specific EAK locus with 1 m contour lines (**C**).

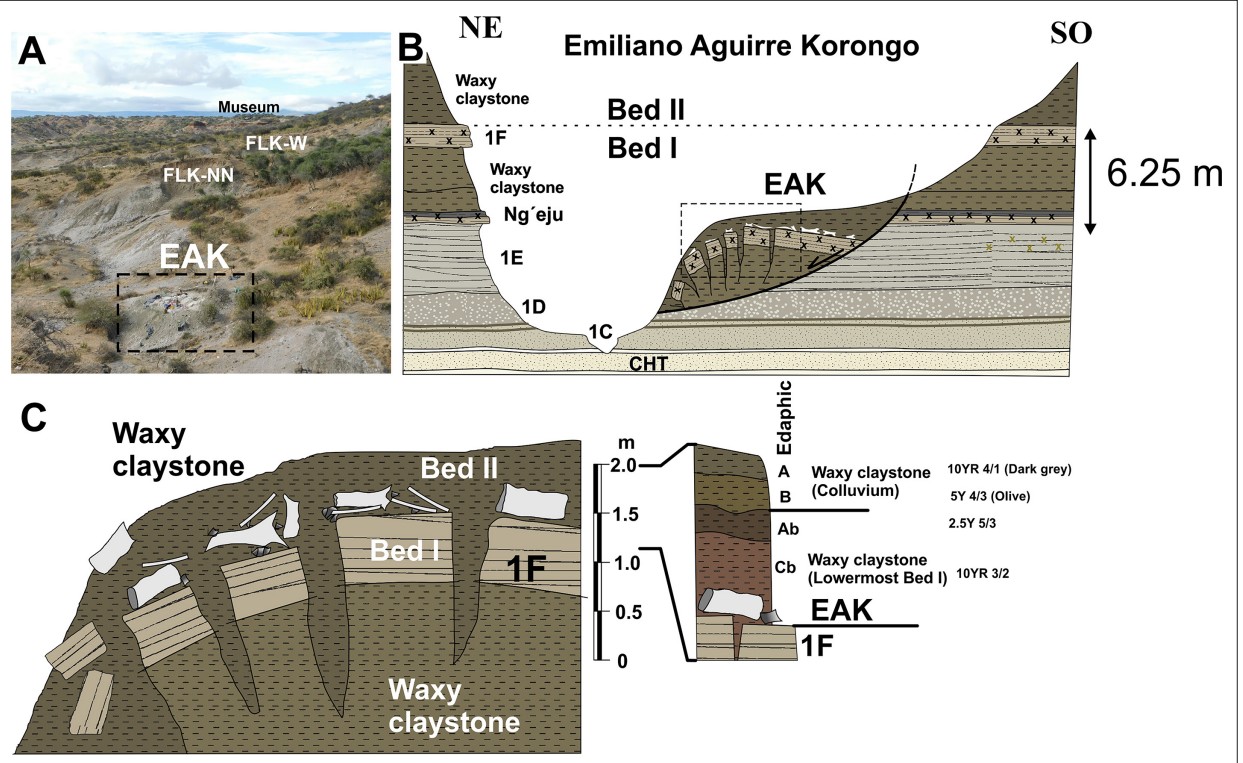

**Figure 2.** Stratigraphy of EAK. (**A**) Location of EAK on the south side of the Main Gorge (view to the east). (**B**) Stratigraphic section of the gully (Korongo) showing the correlation between the different marker tuffs of Bed I on either side. The landslide displaced the boundary between Bed I and Bed II, including the following sequence: waxy claystone, tuff 1 F, waxy claystone. Vertical displacement is 6.25 m. (**C**) Detail of the stratigraphy of the site, showing how the archaeological level rests directly on Tuff 1 F and was covered by the clay of the lowermost Bed II. The landslide has created vertical cracks that separate both the Tuff 1 F and the archaeological level itself. The archaeological remains have moved along with the tuff blocks. The original clay of the lowermost layer II protected the fossils from erosion after the landslide. An incipient soil formed on this surface and was subsequently buried by colluvium. At present, erosion is acting on the part closest to the stream, affecting some of the fossils.

clearly recognizable, Tuff I-F, which serves as the paleosurface of the site, is fragmented into numerous blocks up to 1 m in diameter (*Figure 2*). The archaeological deposit has moved together with the blocks and their original spatial distribution has been altered by this post-sedimentary deformation.

Olduvai Gorge is located in a sedimentary basin bounded to the south by several volcanoes, and to the north by large outcrops of metamorphic rocks. These two groups are the sources of raw material (basalts and quartzites, respectively) during practically the entire Pleistocene at Olduvai. Both the aforementioned basement rocks and the Plio-Quaternary and Quaternary sedimentary fill are affected by these major east-west faults known as 1st, 2nd, 3rd, 4th, KK, FLK, and 5th faults. It should be noted that the FLK fault can only be identified in the landscape through the deformation of the southern escarpment of the gorge, at the beginning of the gully that gives its name to this deposit (EAK). It is a well-known fault because it defines the FLK-5 fault block, described stratigraphically by *Hay, 1976*. Although it is an escarpment degraded by the current geodynamics, it affects the most recent aeolian deposits of Olduvai (Naisiusiu). This geomorphological feature indicates that the FLK fault remained active at least until the Upper Pleistocene to Holocene. The exact direction of the fault is unknown, although the stratigraphy suggests that it must be SW-NE (*Figure 1*), and its vertical drop is several tens of meters. The movement of the fault has probably triggered local earthquakes of varying magnitude. It is well known that earthquakes are one of the main causes of landslides (*Keefer, 1984*; *Ugai et al., 2013*; *Wasowski et al., 2011*), and in this particular case, they may have triggered several landslides affecting deposits from the base of the present gully - that is from the upper part of Bed I to the uppermost Bed (*Figure 2*). The fluvial erosion of the stream at the base of the slopes also destabilized the base of the slopes in this gully, favoring this type of gravitational process. In the case of EAK, this is a very small landslide, although much larger landslides have been recorded in the surrounding area, affecting Bed II units such as the Lower Augitic Sandstone (LAS), which slides over the waxy clay of

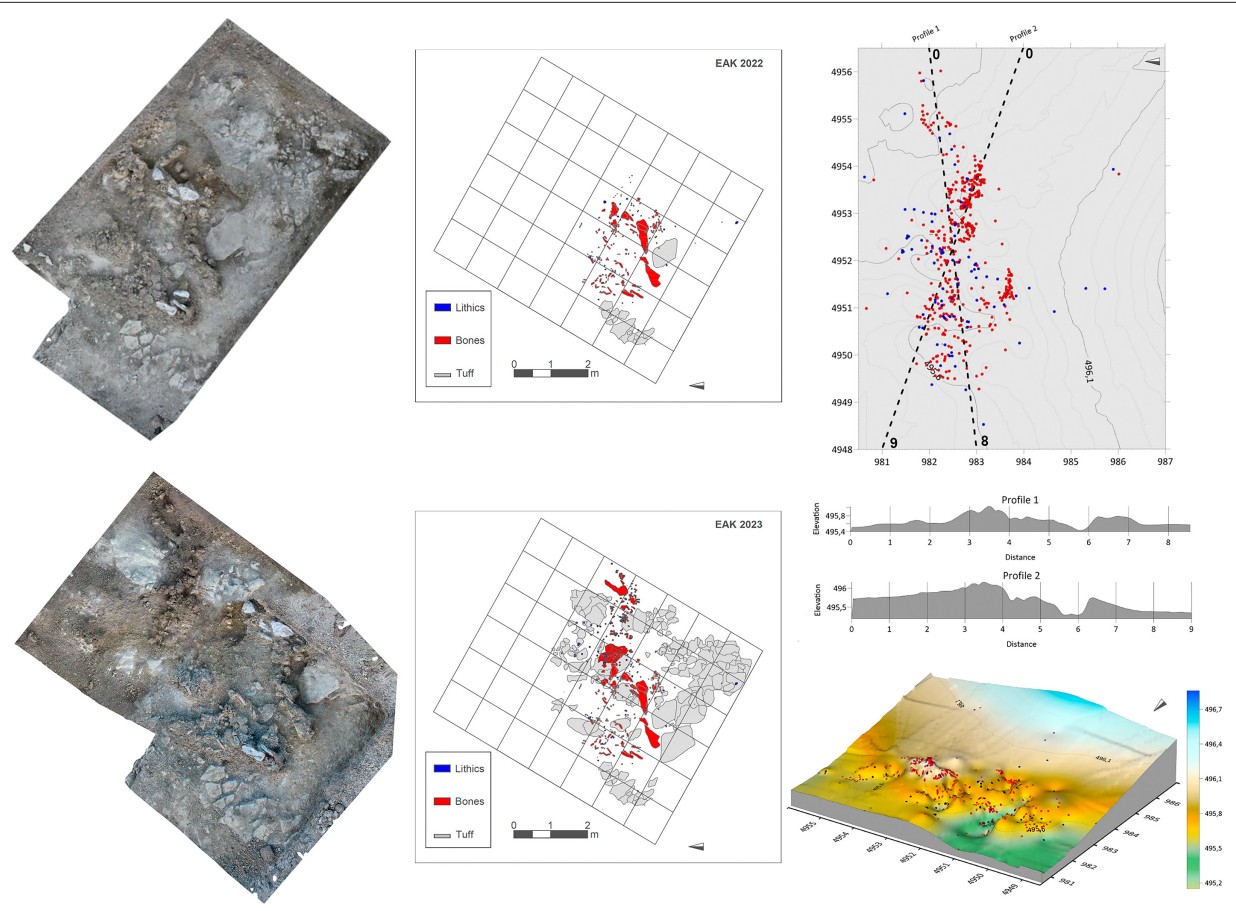

**Figure 3.** Photogrammetry (left) and planimetry (right) of the EAK site in 2022 and 2023. The site-wide distribution of materials, lithics (blue) and bones (red), topographic profiles, and the 3D terrain model are shown.

the lowermost Bed II. Interestingly, there is a similar archaeological site on the Colorado Plateau in northern New Mexico (USA); in this case, a mammoth presenting a similar post-depositional history (*Rowe et al., 2022*).

In June 2022, a partial elephant carcass was found at EAK on a fragmented stratigraphic block situated topographically below FLK North, containing most of the sequence from Tuff ID to the carbonated root clays overlying Tuff IF that constitute the base of Bed II. The carcass must have been eroding over the years, since proboscidean bone fragments and lithics had been found over more than 100 m repeatedly along the seasonally active stream at the bottom of the gulley. In January of the same year, we had not documented anything eroding on the same block, but after the rainy season, several bones emerged, making it obvious that the site contained remains of a proboscidean carcass (*Figure 3*). Excavation started right after the discovery and uncovered remains from a single *E. recki* juvenile individual. We initially set up a 4 x 2 m trench, followed by an extension consisting of two additional trenches (*Figure 3*). The northern side presented an erosive front, probably responsible for several of the scattered elephant bones found along the bottom gully. The remaining excavated partial carcass contained the complete pelvis associated with both hindlimbs (*Appendix 1—table 1*). These included both femora and tibiae, plus one set of tarsal elements (including calcaneus, astragalus) and a fully articulated foot (*Appendix 1—figure 1*). In addition, several rib fragments belonging to a minimum of 8 ribs, as well as a portion of an ulna, were also discovered (*Appendix 1—figure 2*). It is interesting to note that no complete vertebrae were discovered (only one fragment was identified), despite the presence of a portion of the axial skeleton. The skull was positioned upside down, and the projecting tusks were partially preserved (*Figure 3*). In addition to the elephant carcass, a background scatter of a few bones, including a *Parmularius* M², a vertebra, rib, tibia, and metatarsal of a small-sized artiodactyl, was found.

A total of 80 stone tools, mostly flakes and flake fragments, were spatially distributed in tight connection with 153 bone remains, pertaining to 46 elements of a proboscidean carcass (*Figure 3*; *Appendix 1—table 1*). Their preservation is exceptional, and the edges remain sharp. The close spatial association must be functionally linked. Although the cortical bone in several elements was affected by diagenesis, preventing the preservation of any potential bone surface modifications, the good preservation in the remaining assemblage did not yield any marks created either by hominins or other carnivores. Ongoing restoration work will provide an accurate estimate of well-preserved and modified fractions of the assemblage. Two elephant bone specimens showed attributes of green breakage, inferred to be of anthropic origin (see Appendix 1). The tibiae were well preserved, but the epiphyseal portions of the femora were missing, probably removed by carnivores, which would also explain why a large portion of the rib cage and almost all vertebrae are missing. Further support for this likelihood can be found in the deletion of bones from three feet, which are one of the richest sources of fat and affected early in the process of carnivore ravaging (*Haynes, 1991*; *Haynes, 1988*; *Haynes and Hutson, 2020*). The absence of the forelimbs could also be explained if carnivores had dragged them away from the core axial area; however, the fact that the carcass was detached from its original depositional locus and gravitationally moved (with its overlying sediment) after faulting might also explain that the missing carcass parts could still be in their original depositional spot. Our testing of the area where the block was supposed to have been originally formed yielded no results, mainly because of the thick overburden of Ndutu sediments in the area. The fact that the carcass was moved while encased in its sedimentary context, along with the close association of stone tools with the elephant bones, is in agreement with the inference that the animal was butchered by hominins. A more objective way to assess this association is through spatial statistical analysis.

## The spatial statistical analyses of EAK

The analysis of orientation (azimuth) and slope (plunge; *Appendix 1—figure 28*) has shown that the null hypothesis of spatial isotropy cannot be rejected, whether the entire sample is analyzed jointly, or the elephant remains are differentiated from the lithic ones. This result is supported using Rayleigh, Kuiper, and Watson tests, but also by the visualization of the stereograms and rose diagrams, which indeed show the existence of orientations in all directions. Regarding the slope, the stereograms indicate the absence of strong imbrications of remains.

The Spatial Point Pattern (SPP) of the elephant bones has an intensity of 7.74 specimens per square meter. However, this intensity varies significantly across the analyzed surface, with a very high-intensity area in the center of the excavated area and very low intensity surrounding this zone. This pattern is evident both when analyzing the intensity itself and when examining the hot spot maps (*Appendix 1—figure 24a–c*). The use of MAD and DCLF tests has rejected the null hypothesis of spatial randomness with the inhomogeneous $K$ and $L$ functions, as well as with the $K$scaled and $L$scaled functions (p-value = 0.025), while it could not be rejected with the $F$, $G$, and $J$ functions (p-value = 1). These results align with the use of spatial functions (*Appendix 1—figure 24d–k*), as the $F$, $G$, and $J$ functions do not reject the random model, whereas the $K$, $L$, $K$scaled, and $L$scaled functions reject randomness and suggest a clustered SPP. Additionally, the inhomogeneous $K$ and $L$ functions indicate that this clustering occurs only at short distances and that at distances greater than 1 m, the SPP follows a regular pattern. The pair correlation function also appears to indicate the presence of a clustered pattern at very short distances. According to the nearest-neighbor cleaning test, most of the faunal assemblage is highly likely to be part of the cluster (*Appendix 1—figure 24l–n*).

When performing different regression models, we found that the linear regression had an AIC of –861.53, the quadratic regression –1990.689, the cubic regression –2076.40, and the 4th-degree polynomial regression –2100.39. However, we used the quadratic regression for the analysis because the cubic and 4th-degree polynomial regressions were found to be unreliable due to issues related to sample size and collinearity. Therefore, we performed modeling using the quadratic regression (*Appendix 1—figure 24o–r*), reinforcing the interpretation of a very high-intensity cluster in the central part of the excavated area. Furthermore, it is proposed that the accumulation should not extend significantly beyond the excavated area. This supports that site erosion may have affected the preserved concentration less than initially suggested.

The analysis of the SPP for lithic remains shows an intensity of 1.67. Although the sample size is much smaller, it exhibits a pattern similar to that of the bone remains. It also presents intensity and

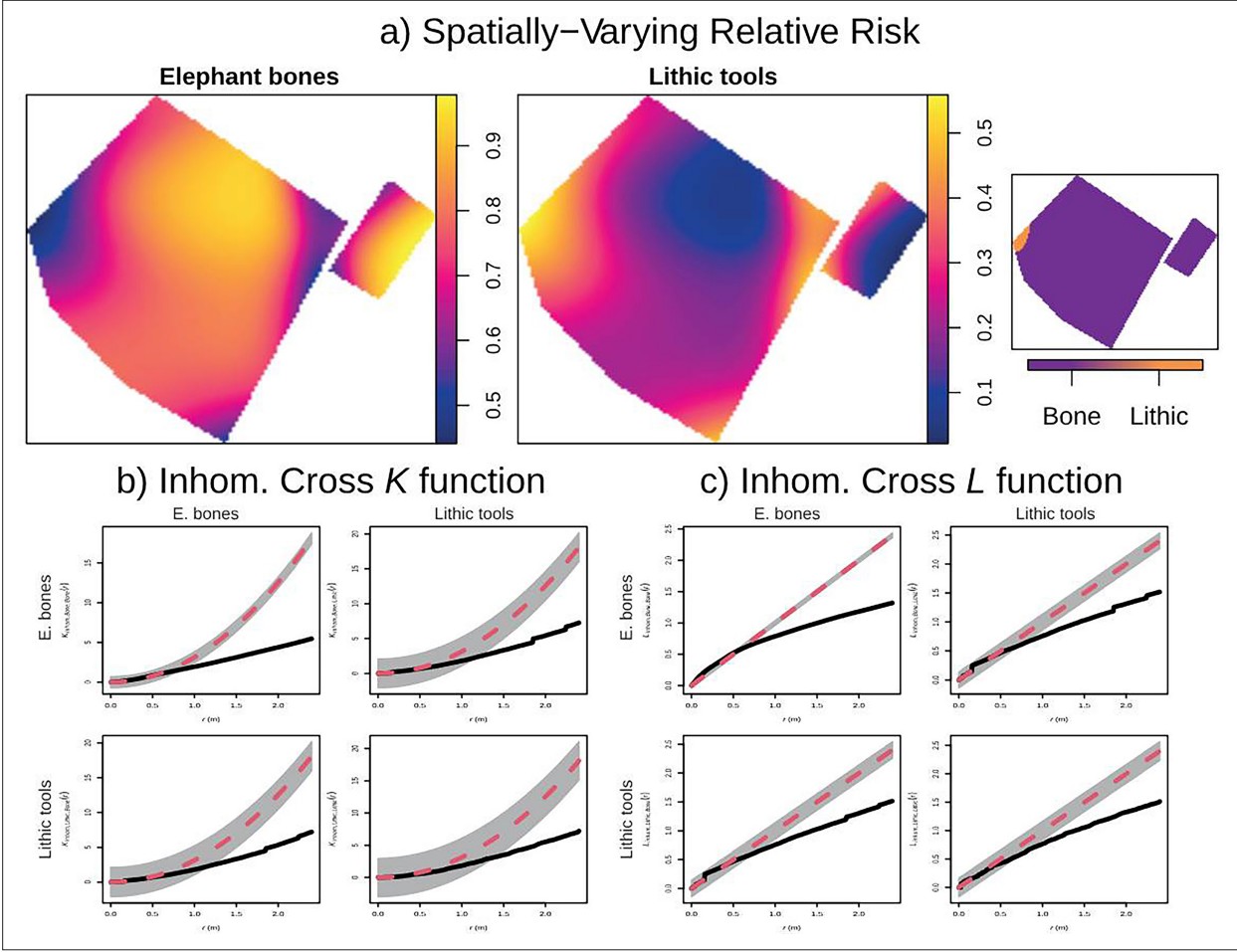

**Figure 4.** Relative risk (i.e. probability of occurrence) of bones and lithic artifacts at EAK, with second-order functions showing regular correlation between both types of items.

hot spot maps (*Appendix 1—figure 25a–c*), with very high intensity in the central area of the spatial window and very low intensity in the surrounding areas. The DCLF and MAD tests yield the same results as those for the elephant remains. However, the functions (*Appendix 1—figure 25d–k*) show some differences. The *F*, *G*, *J*, and pair correlation functions do not reject the null hypothesis, while the *K*, *L*, *K*scaled, and *L*scaled functions do reject it. In this case, the inhomogeneous functions suggest the presence of a regular pattern, while the scaled functions indicate a clustered pattern. Once again, the nearest-neighbor cleaning test (*Appendix 1—figure 25l–n*) suggests that the lithic remains in the high-intensity area are highly likely to be part of a cluster.

The regression analyses in this case show much higher AIC values compared to the faunal remains. However, as before, the 4th-degree polynomial regression has the lowest AIC (–87.99), followed by the cubic regression (–83.02), the quadratic regression (–71.73), and finally the linear regression (79.5). However, once again, the cubic and 4th-degree polynomial regressions cannot be used due to collinearity and sample size issues, making the quadratic regression the most appropriate choice. Modeling with quadratic regression (*Appendix 1—figure 25o–r*) further reinforces the presence of a large cluster and the possible absence of remains outside the analyzed window. Interestingly, the area of maximum intensity does not exactly coincide with that of the elephant remains but is slightly shifted to the left (west) within the spatial window.

The analysis of the SPP with marks (elephant bones vs. lithic tools) shows that, as expected due to the imbalance in sample size, there is always a higher probability of finding elephant remains than lithic remains (*Figure 4a*), with the only exception being at the far left (north) of the spatial window. The analysis of spatial marks reveals a significant segregate correlation between fauna and lithics when using the *K*cross and *L*cross functions (*Figure 4b–c*). The pattern is of regular segregation,

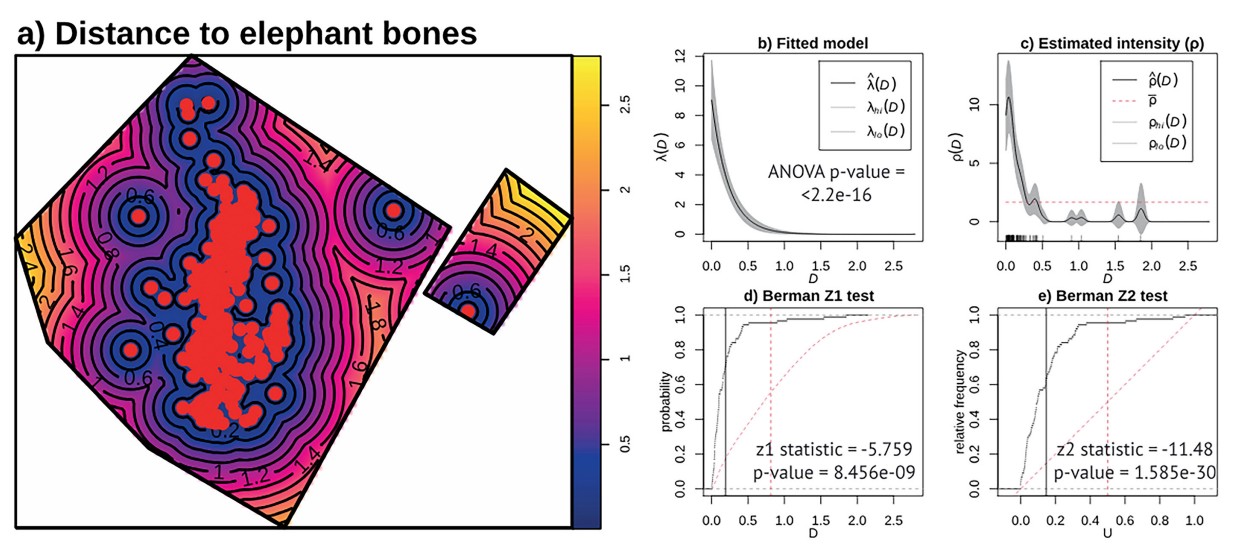

**Figure 5.** Correlation between lithics and bones at EAK (using the elephant bones as a covariate) with the $\rho$ ('rhohat') function and a Z 1 and Z 2 Berman–Lawson–Waller.

meaning that although lithics and bones form part of the same cluster, they tend to occupy different positions, with lithics clustered in the periphery, as would correspond to their use and discard upon the processing of the elephant carcass, which occupies the central part of the cluster.

Finally, analyzing the bones as a covariate has further reinforced the presence of high lithic tool intensity in the center of the spatial window and the relation between fauna and lithic artifacts (*Figure 5*). First, modeling intensity based on the covariate and the estimated intensity ($\rho$) shows that intensity is very high when lithic remains are close to elephant remains, but it drops sharply when moving just 50 cm away. This finding is further supported by the ANOVA test on the model (p-value $\leq 2.2 \times 10^{-16}$) and the results of the Berman $Z_1$ (p-value $\leq 8.456 \times 10^{-9}$) and $Z_2$ (p-value $\leq 1.585 \times 10^{-30}$) tests. This also shows that the functional connection between the elephant bones and the tools has been maintained despite the block post-sedimentary movement. This is further supported by the technological analysis (Appendix 1).

## Traces of megafaunal exploitation along the paleolandscape at Olduvai Gorge

### Bed I

During our survey of the Bed I targeted strata in 2024, no significant traces of megafaunal green bone breakage were observed, with the exception of one locality (*Appendix 1—figure 3*). In previous years, we had observed few megafaunal fossils, mostly restricted to hippopotamus scattered remains. They were not particularly abundant, despite the clearly lacustrine and alluvial depositional settings sampled at the junction.

### Bed II

Out of the three targeted units (lower, intermediate, and upper), the lower unit yielded the highest concentration of elephant and hippopotamus remains, followed by the upper unit (*Appendix 1—figure 3*). The distribution shows a clustered pattern, despite the virtual continuity of most of the sequence in larger areas. The junction area contains the highest density of megafaunal remains for the lower unit. *Appendix 1—figure 3* shows the thorough overlap of those elephant and hippopotamus carcasses with areas containing high density of stone artifacts. All the carcasses documented in the lower unit but one (EAK) occur within the Lower Augitic Sands (LAS). This unit contains a landscape-scale concentration of lithic artifacts spreading from HWKE to FLK West (*Domínguez-Rodrigo et al., 2023*; *Uribelarrea et al., 2017*; *Uribelarrea et al., 2019*).

The distribution of carcasses in the intermediate unit is less dense and mostly restricted to a small portion of the junction. This small area also shows a concentration of lithic remains, but its extent is still unknown since only two localities have been excavated [one to the west of FLK West (*Fujioka et al., 2022*) and BBS] and not much surface material is found that could be securely attributed to this unit in between both sites. BBS (Bob Brain Site) is a recent site discovered in middle Bed II on the same stratigraphic unit as the site named by *Fujioka et al., 2022* as FLK West, recently renamed T69 Complex. In the side gorge, SHK represents this unit, with well-documented presence of megafauna and cut-marked hippopotamus remains (*Diez-Martín et al., 2017*; *Domínguez-Rodrigo et al., 2014c*).

The other large cluster of elephant and hippopotamus carcasses is documented in the upper section of the side gorge (*Appendix 1—figure 3*). There, most of them concentrate around two sites (BK and SC). These sites are also relevant because they contain other smaller megafauna, such as *Sivatherium* and *Pelorovis* in larger numbers (*Domínguez-Rodrigo et al., 2014a*; *Organista et al., 2019*; *Organista et al., 2016*; *Organista et al., 2017*). It is at these two sites and the landscape surrounding them that the highest densities of lithic artifacts are also found in the upper section of Bed II (*Diez-Martín et al., 2009a*).

The clustering of the elephant (and hippopotamus) carcasses in the areas containing the highest densities of landscape surface artifacts is suggestive of a hominin agency in at least part of their consumption and modification. The presence of green broken elephant long bone elements in the area surveyed is only documented within such clusters, both for lower and upper Bed II. This constitutes inverse negative evidence for natural breaks occurring on those carcasses through natural (i.e. non-hominin) pre- and peri-mortem limb breaking (*Haynes et al., 2021*; *Haynes et al., 2020*; *Haynes and Hutson, 2020*). In this latter case, it would be expected for green-broken bones to show a more random landscape distribution and occur in similar frequencies in areas with intense hominin landscape use (as documented in high density artifact deposition) and those with marginal or non-hominin intervention (mostly devoid of anthropogenic lithic remains).

This indicates that from lower Bed II (1.78 Ma) onwards, there is ample documented evidence of anthropogenic agency in the modification of proboscidean bones across the Olduvai paleolandscapes. The discovery of EAK constitutes in this respect the oldest evidence thereof at the gorge. The taphonomic evidence of dynamic proboscidean bone breaking across time and space supports, therefore, the inferences made by the spatial statistical analyses of bones and lithics at the site.

## Discussion

Evidence shows that the oldest systematic anthropogenic exploitation of proboscidean carcasses is documented (at several paleolandscape scales) in the Late Pleistocene sites of Neumark-Nord (Germany; *Gaudzinski-Windheuser et al., 2023a*; *Gaudzinski-Windheuser et al., 2023b*). Evidence of at least episodic access to proboscidean remains goes back in time (see review in *Agam and Barkai, 2018*; *Ben-Dor et al., 2011*; *Haynes, 2022*). Redundant megafaunal exploitation is well documented at some early Pleistocene sites from Olduvai Gorge (*Domínguez-Rodrigo et al., 2014a*; *Domínguez-Rodrigo et al., 2014c*; *Organista et al., 2019*; *Organista et al., 2017*; *Organista et al., 2016*). At the very same sites, the stone artifactual assemblages, as well as the site dimensions, are substantially larger than those documented in the Bed I Oldowan sites (*Diez-Martín et al., 2024b*; *Diez-Martín et al., 2009b*; *Diez-Martín et al., 2014*; *Diez-Martín et al., 2017*). Whether this reflects larger hominin groups or more prolonged and time-averaged intervention in more productive environments remains to be determined (it will likely vary according to site).

A recent discovery of a couple of hippopotamus partial carcasses at the 3.0–2.6 Ma site of Nyayanga (Kenya), spatially concurrent with stone artifacts, has been argued to be causally linked by the presence of cut marks on some bones (*Plummer et al., 2023*). The only evidence published thereof is a series of bone surface modifications on a hippo rib and a tibial crest, which we suggest they might also be the result of byproduct of abiotic abrasive processes; the marks contrast noticeably with the well-defined cut marks found on smaller mammal bones in the same localities (*Plummer et al., 2023*: *Figure 3C and D*) associated with the hippo remains (*Plummer et al., 2023*). The two trenches from which both hippo remains derive are located in the stratigraphic unit NY1, consisting of a channel containing cobble conglomerates and coarse sands in addition to silts. Gravel and coarse sand can generate the same types of marks seen in the figures of the Nyayanga hippo bones (*Domínguez-Rodrigo et al., 2009a*; *Domínguez-Rodrigo et al., 2010b*). Hippopotamidae are the most abundant

taxa in the Nyayanga excavations, indicating a productive taphocoenotic environment, in which hippopotamus remains co-occur with a large diversity of mammal taxa, most of which could have been naturally deposited and mixed. No green fractures have been reported on the hippopotamus bones as found in some other mammal fauna. Although Nyayanga could potentially be one of the earliest examples of megafaunal butchery, the taphonomic evidence remains ambiguous. Butchery experiments show that complete butchery of animals is reflected in a large percentage of remains bearing traces of butchery marks. In T3, these are documented in <0.9% and in T5 in 1%. This is a magnitude of 10–30 times lower than expected using experimental butchery frameworks (even if using only well-preserved bone specimens; *Domínguez-Rodrigo, 1997*; *Domínguez-Rodrigo et al., 2014b*; *Pizarro-Monzo et al., 2021*).

The 1.8 Ma *Elephas recki* of FLK North (level 6, Olduvai Bed I) has the potential of being the oldest proboscidean butchery site (*Leakey, 1971*). It was argued that cut marks existed on its complete bones (*Bunn, 1982*; *Delagnes et al., 2006*), but a subsequent taphonomic study of those marks suggested that they might have been made by trampling (*Domínguez-Rodrigo et al., 2007*). Additionally, it is unclear if the spatial association of stone tools and elephant bones is functional or artificial. *Leakey, 1971* did not report on the third dimension of her excavations, and this is crucial. In our excavations of FLK N6, we documented that the level was a deposit spanning more than half a meter depth and fossils and stone tools were vertically scattered throughout (*Domínguez-Rodrigo et al., 2010a*). It would have been essential to document that the FLK N6 tools associated with the elephant were either on the same depositional surface as the elephant bones and/or on the same vertical position.

The ambiguity about the FLK N6 elephant renders EAK, potentially, the oldest proboscidean butchery evidence at Olduvai, and also probably one of the oldest in the early Pleistocene elsewhere in Africa. FLK North Deinotherium occupied a position above the EAK stratigraphic level, and evidence of hominin intervention there is missing (*Leakey, 1971*). A cut-marked bone fragment of a large animal (probably a hippopotamus) was documented at El-Kherba (Algeria) dated to 1.78 Ma (*Sahnouni et al., 2013*). These findings are suggestive of the beginning of megafaunal exploitation by humans in similar times both in East and North Africa.

Our landscape study of the megafaunal remains at Olduvai shows that elephants and hippopotamuses overlap on the same clustering areas. In all of them, the geology is indicative of extensive alluvial zones. At the base of Bed II, the gorge junction is characterized by extensive wetlands (*Ashley, 2003*; *Ashley et al., 2009*; *Deocampo et al., 2002*; *Hay, 1976*; *Liutkus and Ashley, 2003*). In the overlying LAS, wetlands are crossed by important fluvial systems (*Domínguez-Rodrigo et al., 2023*; *Uribelarrea et al., 2017*). In the middle and upper Bed II fluvial and wetland systems intertwined as the lake was reduced and moved away from its former shoreline in the Bed I-Bed II transition (*Arráiz et al., 2017*; *Garrett, 2017*; *Uribelarrea del Val and Domínguez-Rodrigo, 2017*). In all cases, these wet environments must have been preferred places for water-dependent megafauna, like elephants and hippos, and their overlapping ecological niches are reflected in the spatial co-occurrence of their carcasses. Both types of megafauna show traces of hominin use through either cutmarked or percussed bones, green-broken bones, or both (Appendix 1).

Green-broken elephant bones are documented at Olduvai at a landscape scale during the LAS deposition (1.7 Ma; *Diez-Martín et al., 2015*; *Domínguez-Rodrigo et al., 2023*; *Uribelarrea et al., 2017*). This record is now added to the newly discovered EAK site, showing tight spatial association (both vertically and horizontally) between bones and stone tools. This trend of spatial association of stone artifacts and proboscidean bones continues through the deposition of Bed II. Although no cut marks have been documented on the proboscidean bones that we identified (part of which had cortical surfaces impacted by biostratinomic and diagenetic modification), the presence of green-broken bones at the sites, the butchery of carcasses as indicated by spatial statistical analyses at sites like EAK and TK (*Domínguez-Rodrigo et al., 2014a*; *Organista et al., 2016*; *Panera et al., 2019*), and the presence of hammerstone-broken bones on the surrounding paleo-landscapes of these and other sites support the occasional exploitation of megafaunal remains by humans. In most green-broken limb bones, we document the presence of a medullary cavity, despite the continuous presence of trabecular bone tissue on its walls. We do not know if the bone breakage was intended to obtain marrow or to create raw material for making artifacts or tools with functional purposes or both. We have been able to identify only one potential bone tool in our sample (*Figure 6*, *Appendix 1—figures 4–23*); the scarcity of bone implements supports the former option. The exploitation of megafauna is

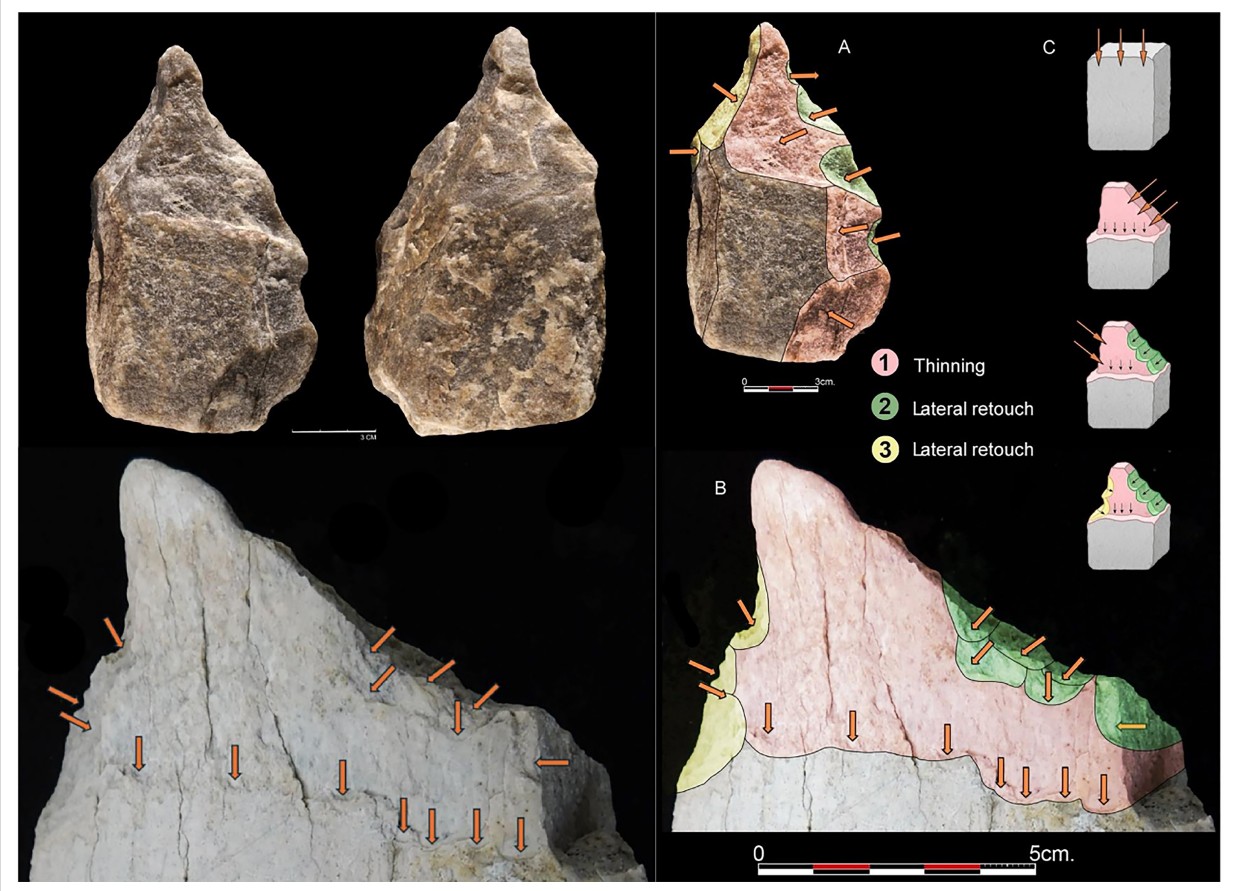

**Figure 6.** Left: Intentional shaping of points in a quartzite LCT from FLK West, and on a proboscidean femur shaft. Arrows indicate conchoidal scars probably caused by use (given their location away from the edge) and their reflection and stepped morphology by the pointed area. Notice the polishing at the point, which contrasts with the unpolished state of the remainder of the artifact. More complete cortical and medullary views of this artifact can be seen in **Appendix 1—figures 6–8**. Right: Comparison of the configuration strategies identified on the quartz LCT and the LAS proboscidean femur shaft. (**A**) LCT on a quartz slab from FLK West, Level 6; (**B**) The proboscidean femoral shaft from LAS in the vicinity of EAK; (**C**) Diacritical diagram showing point conceptualization sequencing identified in the previous specimens: 1. First thinning work (oblique to the distal end in the quartz LCT and vertical in the bone shaft); 2. Right-sided retouch; 3. Left-sided retouch. Series 2 and 3 determine the final shaping of the distal tip.

not anecdotal, since at sites like BK or TK (upper Bed II), the taphonomic analysis of other megafaunal remains (namely, *Sivatherium* and *Pelorovis*) clearly shows intensive exploitation of complete carcasses in both of these taxa, especially at BK (**Domínguez-Rodrigo et al., 2014a**; **Organista et al., 2016**; **Panera et al., 2019**). The food surplus generated by the exploitation of some of these animals would have been a significant advantage for the survival and adaptation of the hominin groups.

These results of widespread conspicuous evidence of hammerstone-breaking behaviors during Bed II partially confirm the recent report on bone artifacts discovered in Bed II by another research team (**de la Torre et al., 2025**). Excavations at the T69 Complex area unearthed a wealth of megafaunal (proboscidean and hippopotamus) remains displaying hammerstone green breakage and hominin modifications. These remains seem to be spatially connected with a highly productive paleoland-scape, since at its northern end, it connects with BBS, a recently discovered site with megafaunal (Hippopotamus) remains. Green breakage at BBS also reflects hominin intervention in the palustrine environment congregating so many megafaunal carcasses. This unit would be equivalent to our second unit in Bed II.

We are not confident that the artifacts reported by dela Torre et al. at the T69 Complex are indeed tools. They are similar to the green-broken specimens that we have reported here. They are characterized by repeated number of flake scars along the edges; however, this is not a diagnostic feature, since multiple scars along edges also indirectly result from the repeated battering until the final breakage of the bone during butchery (**Haynes et al., 2021**; **Haynes et al., 2020**; work in progress). In large

bovids, it can also be derived from the gnawing of durophagous carnivores (***Villa and Bartram, 1996***; ***Appendix 1—figure 32***). Carnivores capable of breaking elephant bones and producing similar break shapes go back to at least the early Middle Miocene (***Appendix 1—figure 33***), although no carnivore today can effectively break the dense long bone mid-shafts of adult elephant bones. Another argument used by the authors is that the purported tools are longer and bear more flake scars compared to other faunal fragments (approximate average 13 vs 4). Again, this is documented among durophagous carnivores (***Villa and Bartram, 1996***; see examples in Appendix 1). A higher scar number in the T69 artifact collection could result from a correlation of different variables (i.e. longer dimension of the compared specimens, or different carcass sizes) and be, therefore, unrelated to anthropic intentionality. A third argument provided is that the items share consistent technological features, such as the production of morphologically similar, elongated, pointed, and notched bone tools, suggesting a patterned behavior. Again, this is equifinal with unintentional breaking of megafaunal remains during butchery (***Haynes et al., 2021***; ***Haynes et al., 2020***; ***Appendix 1—figures 6–19***; experimental work in progress). There are several arguments that could be used to support this alternative:

1. The scars in the T69 artifacts exhibit an asymmetric distribution, characterized by stepped or reflected, incomplete fractures predominantly on the medullary surface, with minimal invasiveness. This contrasts with the expected conchoidal and complete scarring distributed equally across both cortical (i.e. periosteal) and medullary surfaces, as would be anticipated in cases of intentional flaking. Given that bone is more easily flaked than metamorphic rock, and that bifacial knapping of stone typically produces symmetrical, conchoidal scars on both surfaces, this would also be expected in the T69 bone specimens.

2. This bilateral symmetry principle is observed in Middle Pleistocene bone handaxes, where intentional modification is inferred based on the equal treatment of both sides and evidence of artificial shaping (***Zutovski and Barkai, 2016***). It is also documented on the similarly aged 1.4 Ma bone handaxe from Konso (***Sano et al., 2020***), and a purported handaxe from FC (Bed II, Olduvai; ***Backwell and d'Errico, 2004***), but it is absent from the reported items in T69, as well as those of the present study. Why, otherwise, would scars on the medullary surface be mostly limited to the wall of the breakage plane?

3. The repeated presence of incomplete, stepped scars on the medullary surfaces of the artifacts found at T69 aligns more closely with damage resulting from deliberate battering associated with breakage rather than subsequent edge retouching. The latter process would be expected to generate more controlled and complete conchoidal scars indicative of intentional shaping (***Backwell and d'Errico, 2004***; ***Haynes et al., 2021***; ***Haynes et al., 2020***). This reflected scarring is also documented in Bos/Bison bones broken by hyenas, caused by redundancy in gnawing and the resulting overlapping micro-cracking leading to final collapse (***Villa and Bartram, 1996***; ***Appendix 1—figure 31***).

4. Scar frequency is not a taphonomically valid criterion to attribute agency, since elephant bones can be broken in multiple ways, thus impacting experimentally derived scar patterns. Sequential battering of initial crack lines with handheld hammerstones can generate overlapping notches and scars (work in progress). This is not necessarily reproduced when other experimental studies use alternative methods (such as throwing or bashing against immobile anvils) (***Backwell and d'Errico, 2004***; ***Haynes et al., 2021***).

5. In several of the artifacts from T69, most of the overlapping scarring occurs in association with notches (and, in some cases, associated percussion marks) derived from dynamic loading. This reinforces that they may have resulted from repeated battering during bone breaking.

6. The T69 Complex bone artifacts and those reported here are large-sized. If their potential function was for heavy-duty activities, those should have left any visible traces (at the macro and microscales) on the scarred edges. These are absent. A faunal assemblage that has good preservation of cut and percussion marks should certainly have good preservation conditions for microwear use and macro-damage scars.

7. If applying the same criteria defining bone tools at T69 Complex to Bos/Bison modified bones, hyenas would also create similar artifacts (***Appendix 1—figure 32***).

A final argument used by the authors to justify the intentional artifactual nature of their bone implements is that the bone tools were found in situ within a single stratigraphic horizon securely dated to 1.5 million years ago, indicating systematic production rather than episodic use. This is taphonomically unjustified. In a productive paleosurface where megafauna is abundant and hominin intervention is intense, the presence of this type of highly scarred green-broken bones can be fairly common, since they result from butchery and carnivore post-depositional ravaging on non-proboscidean megafaunal

bones (*Domínguez-Rodrigo et al., 2014a*; *Domínguez-Rodrigo et al., 2014c*; *Domínguez-Rodrigo et al., 2009b*; *Villa and Bartram, 1996*). Here, we have shown that most green-broken elephant bones cluster in three main areas of the Olduvai gorge.

Leakey collected more than a hundred purported bone tools from the same locations of Bed II where we collected our hammerstone-broken bones (*Leakey, 1971*; *Appendix 1—figure 33*). All of them derive from Bed II (*Backwell and d'Errico, 2004*). A recent taphonomic review cautiously identified six potential bone tools in that collection (*Pante et al., 2020*). Some of these authors, using the much smaller T69 Complex artifact sample, now argue about systematic bone tool production. Bone artifacts (i.e. hominin-made items) that are used as tools should display evidence thereof through the macroscopic scarring, the polishing of the actively used surface, and the presence of microabrasive modifications on it (*Backwell and d'Errico, 2008*, *Backwell and d'Errico, 2001*; *d'Errico and Backwell, 2009*; *LeMoine, 1994*). None of that has been documented in the T69 assemblage. We do not deny that the items reported by *de la Torre et al., 2025* have the potential of having been used as tools, but proof is currently missing. The same applies to our reported sample (*Figure 6*; *Appendix 1—figures 4–25*). Only one artifact shows modifications that are potentially derived from its use as a tool. If confirmed, it is interesting to note the convergence in the point shape targeted in such an implement and those intentionally shaped in the earliest Acheulian implements documented at Olduvai Gorge (*Figure 6*, *Appendix 1—figures 6–8*). If such a bone implement was a tool, it would be the oldest bone tool documented to date (>1.7 Ma).

## Conclusions

Here, we have reported a significant change in hominin foraging behaviors during Bed I and Bed II times, roughly coinciding with the replacement of Oldowan industries by Acheulian tool kits -although during Bed II, both industries co-existed for a substantial amount of time (*Domínguez-Rodrigo et al., 2023*; *Uribelarrea et al., 2017*; *Uribelarrea et al., 2019*). This correlation does not necessarily imply causation, given that most tools documented around some of these butchery sites (e.g. EAK) are typologically Oldowan (mostly flakes and fragments). The traditional Acheulian large cutting tools do not seem to have played any role in megafaunal butchery, since they are most commonly undocumented or under-represented at early butchery sites, and where they are predominant or very common (e.g. TK), they are not functionally associated with faunal exploitation (see the examples of typological changes at TK, when *Elephas* and *Sivatherium* carcasses are exploited; *Panera et al., 2019*). In contrast, the occurrence of these megafaunal exploitation areas in three Bed II paleolandscape sections containing extremely large sites (e.g. FLK West in the LAS strata; SHK and BK in middle and upper Bed II respectively in the secondary gorge) attests to larger group sizes, which may be in connection with the intensity of exploitation of megafaunal resources. This was argued to be the main trigger in the thorough exploitation of a complete *Sivatherium* carcass at BK (*Domínguez-Rodrigo et al., 2014b*), another one at TK (*Panera et al., 2019*), and several *Pelorovis* individuals at BK (*Organista et al., 2016*). The documentation of *H. erectus* around this time, with a much bigger anatomy than previous and sympatric hominins, suggests that a meat-based diet was behind these changes (*Domínguez-Rodrigo et al., 2021*). Having repeated access to high-yielding megafaunal taxa would have enabled access to a large supply of food for a large group for a substantial amount of time. The same as modern Hadza can fend off predators like lions from large kills, those early hominins might have sustained a similar eco-trophic position, especially given their bigger size. An additional important resource, fat, might have also pushed hominins to target proboscideans, since their long bones contain large amounts of fat, rich in nutritional EPA. This resource is currently under no competition, since modern hyenas cannot break open elephant long bone shafts from adult individuals.

The evidence presented here, together with that documented by *de la Torre et al., 2025*, represents the most geographically extensive documentation of repeated access to proboscidean and other megafaunal remains at a single fossil locality. The transition from Oldowan sites, where lithic and archaeofaunal assemblages are typically concentrated within 30–40 m² clusters, to Acheulean sites that span hundreds or even over 1000 m² (as in BK), with distinct internal spatial organization and redundancy in space use across multiple archaeological layers spanning meters of stratigraphic sequence (*Domínguez-Rodrigo et al., 2014a*; *Domínguez-Rodrigo et al., 2009b*; *Organista et al., 2017*), reflects significant behavioral and technological shifts. This pattern likely signifies critical innovations in human evolution, coinciding with major anatomical and physiological transformations in

early hominins (*Dembitzer et al., 2022*; *Domínguez-Rodrigo et al., 2021*; *Domínguez-Rodrigo et al., 2012*). The causal relationships of these events that appear correlated in the archaeological record remain to be established.

## Methods
### Excavation of EAK

The systematic excavation of the stratigraphic layers involved a small crew. Materials exceeding 2 cm in size were meticulously plotted using laser theodolites (TOPCON total stations). Each trench was subjected to stereo-photography upon exposure, enabling the creation of photogrammetric 3D reconstructions of the paleosurface and deposited materials prior to recovery. The orientation and inclination of the artifacts were recorded using a compass and an inclinometer, respectively. Sediments removed from the trenches were sieved using 5 mm and 3 mm meshes.

The photogrammetric reconstructions of each trench were subsequently georeferenced and the resulting orthophotos were imported into AutoCAD, facilitating the detailed mapping of archaeological materials. The decision to conduct photogrammetry by individual trenches was made to minimize the exposure time of the excavated materials. Prolonged exposure could accelerate desiccation, potentially causing the most fragile specimens to become brittle and fragmentary. To mitigate this risk, excavation was limited to small, manageable areas, with small trenches proving ideal for efficient excavation due to the limited overburden. Each trench was excavated until the entire archaeological horizon was exposed, followed by comprehensive photogrammetric documentation prior to 3D plotting. Throughout the excavation, professional restoration experts provided necessary treatment for the bones. This was crucial for the recovery and identification of many bone fragments.

### Restoration of the EAK elephant bones

The main interventions were carried out in two locations: in situ and in the field laboratory at the Emiliano Aguirre Research Station. For the consolidation and extraction of remains at the site, interventions were performed using Paraloid B-72 diluted in acetone at different concentrations, applied exclusively to fossils exhibiting powdering. This methodology strengthened the bones prior to extraction, ensuring better handling and reducing the risk of fractures. The use of Paraloid B-72 is a standard practice in the conservation of osseous remains due to its stability and compatibility with the material, though its limited penetration in some bone types has been noted (*Daura et al., 2017*). During this process, bandaging with sterile gauze impregnated with Paraloid B-72 was selectively applied to necessary areas to further protect fragile sections. Additionally, superficial bandaging was conducted on larger fossils that visibly presented potential fractures and retained their cortical surface, aiming to prevent deterioration during the extraction process.

The greatest challenge was extracting long bones due to their size, weight, and fragility. These bones exhibited a greater presence of cortical surface, although they were not completely preserved, as they were fragmented but still retained significant areas suitable for study.

Another significant challenge was the extraction of tusk remains, as the ivory exhibited a high degree of degradation and fragmentation. To recover these, multiple applications of consolidant were performed, allowing certain tusk blocks to be extracted in a controlled manner.

Larger fossils were extracted in block to minimize the risk of structural damage. The extraction process involved:

1. Initial consolidation by applying Paraloid B-72 dropwise to the surface, along with bandaging with sterile gauze in necessary areas to provide additional protection.
2. Surface protection with a sacrificial layer, avoiding direct contact with other reinforcing materials.
3. Creation of a counter-mold using plaster bandages, reinforced with interwoven steel rods to provide rigidity and stability.
4. Insertion of metal spatulas beneath the remains to lift and safely transport them.

In the field laboratory, cleaning of the bone remains was carried out, along with adhesion of fragments and their consolidation when necessary. The primary priority was preparing the remains for study by the rest of the team, opting for a purely conservative intervention, prioritizing detailed cleaning of some fragments for identification. Additionally, reassembly of larger fragments was performed, using Paraloid B-72 diluted in acetone at a higher concentration to ensure effective adhesion. For bone

remains, mechanical and chemical cleaning procedures were applied depending on the nature of the adhered deposits.

This restoration procedure enabled the preservation of many bone fragments. Cleaning them properly will require much more time. In their current status, several specimens still have some adhering sedimentary matrix. This has prevented us from thoroughly examining all bone surfaces looking for BSM or conducting comprehensive identification. Some elements and fragments remain unidentified given also the lack of a proper comparative proboscidean collection. Despite this, most of the assemblage has been successfully studied.

## Taphonomic analysis of EAK

Skeletal part profiles were determined using the Number of Identifiable Specimens (NISP) and the Minimum Number of Elements (MNE) (*Appendix 1—table 1*). Given that with the exception of a few faunal specimens, the majority of the assemblage belongs to a single elephant individual, we did not use hierarchical units like MNI or MAU. To determine the MNE, ribs have been divided according to *Rodríguez-Hidalgo et al., 2017* system: portion 1 belongs to the epiphysis which includes the head, neck, and costal tubercle; portion 2 belongs to the coastal angle; portion 3 belongs to the proximal body; portion 4 is composed of the distal body and portion 5 belongs to the sternal zone. The high frequency of fragmentation documented in the EAK's ribs and the lack of landmarks made us determine the rib's MNE through portions 1 and 2, in addition to a side-by-side comparison of rib shaft fragment widths (portions 3 and 4).

In the present study, the presence of abrasion and polishing due to water dynamics (i.e. resedimentation, transport, lag) was taken into account. The first issue we analyzed regarding the spatial distribution of archaeological remains was the orientation and slope of the items. First, we created stereograms and rose diagrams for the sample of elephant bone remains and lithic artifacts, as well as for both sets combined. These graphs were generated using OpenStereo software. Next, we conducted Rayleigh, Watson, and Kuiper tests to determine whether the isotropy/anisotropy tendency could be rejected (*Fisher, 1995*).

An evaluation of the cortical surfaces was made, followed by an analysis of bone surface modifications; namely, cut marks, tooth marks, percussion marks, and natural marks (i.e. biochemical and abrasion marks). Marks were identified by using hand lenses under strong direct light (60 W) following the methodological and mark diagnostic criteria specified in *Blumenschine, 1995* and *Blumenschine and Selvaggio, 1988* for tooth and percussion marks, *Domínguez-Rodrigo et al., 2009a*; *Domínguez-Rodrigo et al., 2010b* for cut marks, and *Domínguez-Rodrigo and Barba, 2006* for biochemical marks. The analysis of bone breakage at EAK is based on the identification of green and dry breakage planes following *Villa and Mahieu, 1991* criteria.

## Lithic analysis of EAK

A total of 80 lithic specimens have been recovered from EAK. The technological analysis of these materials follows established classification criteria and methodological frameworks (*Diez-Martín et al., 2009a*), categorizing artifacts into the following groups:

a. Unmodified Specimens – Naturally occurring cobblestones lacking evidence of anthropogenic modification.
b. Percussion Tools – Includes hammerstones or cobbles exhibiting diagnostic battering, pitting, and/or impact scars consistent with percussive activities.
c. Cores – Handheld cores are classified based on key technological attributes: (i) the number of reduction surfaces (unifacial, bifacial, multifacial), (ii) the number of striking platforms (unipolar, bipolar, multipolar), and (iii) the spatial organization of knapping sequences (linear, opposed, orthogonal, centripetal). Bipolar cores are identified based on diagnostic features associated with anvil-supported percussion.
d. Detached Products – This category encompasses both complete and fragmented flakes, derived from either bipolar or freehand reduction. Technological attributes recorded for detached products include core reduction stage (assessed through the presence and extent of cortical coverage on dorsal surfaces and striking platforms), platform morphology, and dorsal scar patterning.
e. Knapping By-Products – Includes all non-diagnostic lithic debris resulting from core reduction and flake detachment. This category incorporates non-diagnostic flake fragments with evidence

of direct handheld percussion, shatter, blocky/angular undetermined detached fragments, core-like fragments, and debris ≤20 mm in size.

## The spatial statistical analyses of EAK

The spatial point patterns (SPP) of faunal and lithic remains have been analyzed using the 'spatstat' library (*Baddeley et al., 2015*) in R (https://www.r-project.org/). This methodological approach is of great interest to paleontological and archaeological contexts, as it allows for the analysis of aspects such as the degree of randomness of the SPP, potential relationships between different types of materials, or enabling predictions about unexcavated areas. We conducted the analysis in three different ways after selecting the spatial window, that is the analyzed excavated area (52.56 m$^2$).

First, we analyzed the SPPs without marks (i.e. raw SPPs, without considering any specific spatial variable) for the elephant remains and lithic remains independently. In these initial cases, we created intensity maps, hot spot maps, and hot spot maps with a 99% Confidence Interval (C.I.). Next, we applied a chi-squared test with four different grid configurations (5×5, 8×8, 10×10, 12×12; *Moclán et al., 2023*) to determine whether the SPPs are homogeneous or inhomogeneous. These analyses showed that, in all cases, the SPPs can be considered inhomogeneous (p-value <0.05). Therefore, we proceeded to use tests to analyze the type of SPP. We employed the Pair Correlation, *K*, *L*, *F*, *G*, and *J* functions in their inhomogeneous versions, as well as the *Kscaled* and *Lscaled* functions to analyze the typology of the SPPs (i.e. random, cluster, or regular). Both versions of the functions were used because the sample size does not allow us to analyze the type of inhomogeneity through a studentized permutation test. Similarly, we used the DCLF (Diggle-Cressie-Loosmore-Ford) and MAD (Maximum Absolute Deviation) tests with the *K*, *L*, *F*, *G*, *J*, *Kscaled*, and *Lscaled* functions to check the typology of the point patterns (i.e. random vs. non-random). Finally, we used the 'nnclean' function (*Baddeley et al., 2015*) to detect possible clustering of materials through nearest neighbor clutter removal. Lastly, we performed spatial modeling using linear, quadratic, cubic, and 4th-degree polynomial regressions to delve deeper into the characteristics of the SPPs. The models were evaluated using the Akaike Information Criterion (AIC) index, and subsequently, we compared their validity through ANOVA. Finally, we selected the best-performing model (i.e. the one with the lowest AIC values and that does not show issues of collinearity or suffer from the small sample size) and generated four maps: (1) estimated intensity of the point process based on the fitted model, (2) simulation of a new point pattern using the fitted model with the Metropolis-Hastings algorithm, (3) a difference map subtracting the intensity of the simulated new point pattern from the 'real' intensity of the spatial point pattern (*Domínguez-Rodrigo et al., 2024*), and (4) an extrapolated estimation of potential points within a larger spatial window (308.23 m$^2$) than the original one.

The second analysis consists of examining the degree of statistical correlation between two marks, that is spatial variables, in this case, fauna vs. lithic tools. To do this, we first calculated the relative risk to assess the likelihood of each type of material appearing in the analyzed window. Then, we applied a chi-square test to verify once again, using the same grid configuration, whether the samples are homogeneous or inhomogeneous. The chi-square test again indicates that the spatial distribution is inhomogeneous. Therefore, we used the *K·*, *L·*, *Kcross*, and *Lcross* inhomogeneous functions to analyze whether there is spatial correlation between the two material types and, if so, what kind it is.

Finally, given the size disparity between the elephant remains and the lithics, we analyzed the relationship between both types of materials by considering the location of the elephant remains as a covariate. First, a distance map was generated to the centroids of the bone remains. ANOVA was then conducted to assess the fitted effect of the covariate in the SPPs. This was followed by the application of the $\rho$ function ('rhohat' function) to investigate the influence of the distance to the bones. This test is similar to the previous one but provides a more detailed view of the potential variations in the intensity of remains with respect to distance from the bones. Finally, a $Z_1$ and $Z_2$ Berman–Lawson–Waller test was performed.

## Megafaunal exploitation along the paleolandscape sequence of Olduvai Gorge

An intensive survey seeking stratigraphically associated megafaunal bones was carried out in the months of June 2023 and 2024. We targeted the Bed I and Bed II exposures of the main Gorge (from FLK NNN to the KK fault), and the side gorge, from the junction until BK. Linearly, this comprises over

6 km of outcrops. We targeted this area because it contains the highest density of fossils of all the gorge. We focused on proboscidean bones and used hippopotamus bones, some of the most abundant in the megafaunal fossils, as a spatial control. Bones were counted as separate carcasses when enough distance existed in between findings (i.e. localities). Stratigraphic association was carried out by direct observation of the geological context and with the presence of a Quaternary geologist during the whole survey. When fossils found were ambiguously associated with specific strata, these were excluded from the present analysis.

For the sake of analysis, and given that we were not focusing on time-resolution landscapes, we used four time intervals: Bed I, Bed II lower (from Tuff IF to LAS), Bed II intermediate (from Tuff IIB to Tuff IIC), and Bed II upper (from Tuff IID to the interface with Bed III). Only in one case, Bed II lower, were most of the remains circumscribed to a single stratigraphic unit (Lower Augitic Sands [LAS], which mark the beginning of Hay's middle Bed II). The goals of this survey were: (a) collect a spatial sample of proboscidean and megafaunal bones enabling us to understand if carcasses on the Olduvai paleolandscapes were randomly deposited or associated to specific habitats; (b) given the water dependence of elephants, we wanted to understand if they showed the same type of ecological association to water-dependent megafauna, like hippopotamus, as their modern counterparts (using taphocoenoses as indicators of past megafaunal biocoenosis); (c) approach potential anthropogenic agency in proboscidean exploitation by comparing carcass dispersal and accumulation and productivity of anthropogenic landscapes (i.e. concentrations and dispersals of lithic artifacts); (d) seek anthropogenic traces of proboscidean exploitation in the form of cut and percussion marks and green bone breakage; (e) associate this latter point with the hominin use of space, by determining if these taphonomic evidences occurred randomly or clustered and spatially associated with stone tools.

## Acknowledgements

We thank the Spanish Ministry of Science and Innovation for funding this research (PID2023-146260NB-C2), and the Spanish Ministry of Culture for their funding through the program of Archaeology Abroad. We thank the Commission for Science and Technology (COSTECH), the Ngorongoro Conservation Area Authorities (NCAA), the Division of Antiquities, and the Tanzanian Ministry of Natural Resources and Tourism for their permission to conduct research in Tanzania. AM is funded by a postdoctoral contract grant from the Fyssen Foundation. We are grateful to the Tanzanian co-workers at Olduvai Gorge. We are deeply indebted to the constructive comments made by G Haynes, R Barkai and one anonymous reviewer to an earlier version of this manuscript. We also thank a second anonymous reviewer, despite our disagreement with some of his/her comments. We also appreciate the professional editorship of Yonatan Sahle and Detlef Weigel.

## Additional information

### Funding

| Funder | Grant reference number | Author |
|---|---|---|
| Ministerio de Ciencia, Innovación y Universidades | PID2023-146260NB-C2 | Manuel Dominguez-Rodrigo |
| Fondation Fyssen | | Abel Moclan |

The funders had no role in study design, data collection and interpretation, or the decision to submit the work for publication.

### Author contributions

Manuel Dominguez-Rodrigo, Conceptualization, Data curation, Formal analysis, Funding acquisition, Investigation, Methodology, Writing – original draft, Project administration; Enrique Baquedano, Resources, Funding acquisition, Validation, Project administration, Writing – review and editing; Abel Moclan, David Uribelarrea, Formal analysis, Validation, Investigation, Writing – review and editing; José Ángel Correa-Cano, Validation, Investigation, Methodology, Writing – review and editing; Fernando Diez-Martin, Resources, Validation, Investigation, Writing – review and editing; Alejandro

Velazquez-tello, Investigation; Elia Organista, Eduardo Mendez-Quintas, Marina Vegara-Riquelme, Agness Gidna, Validation, Investigation, Writing – review and editing; Audax Mabulla, Project administration

**Author ORCIDs**
Manuel Dominguez-Rodrigo (iD) https://orcid.org/0000-0002-7233-331X
Audax Mabulla (iD) https://orcid.org/0000-0001-9591-5497

Reviewer #1 (Public review): https://doi.org/10.7554/eLife.108298.5.sa1
Reviewer #2 (Public review): https://doi.org/10.7554/eLife.108298.5.sa2
Author response https://doi.org/10.7554/eLife.108298.5.sa3

## Additional files

**Supplementary files**
MDAR checklist

**Data availability**
All data is included in the paper.

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

# Appendix 1

## EAK excavation and landscape distribution of megafaunal green-broken bones

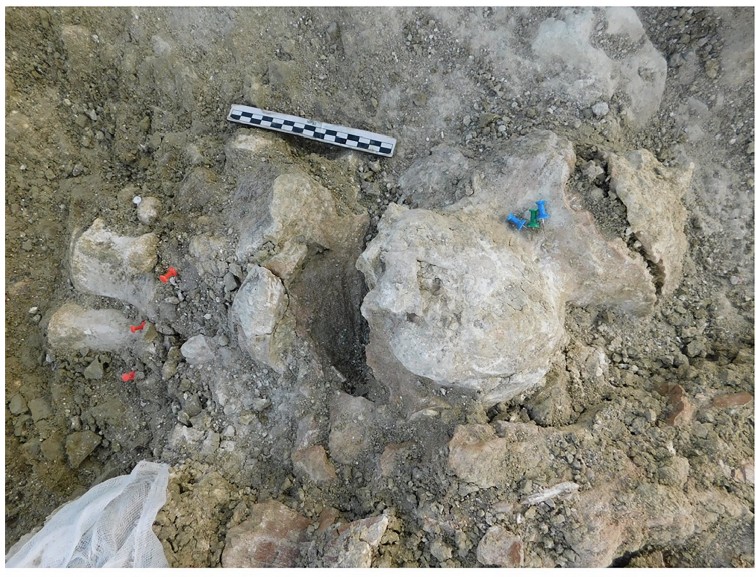

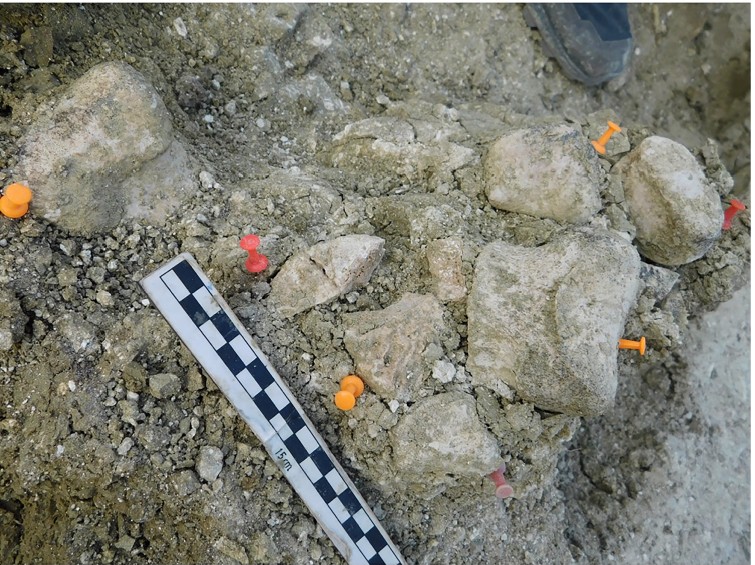

**Appendix 1—figure 1.** Excavation at EAK of the partial articulated rear foot of the juvenile elephant, displaying the calcaneum with unfused epiphysis (upper) and articulated phalanges (lower).

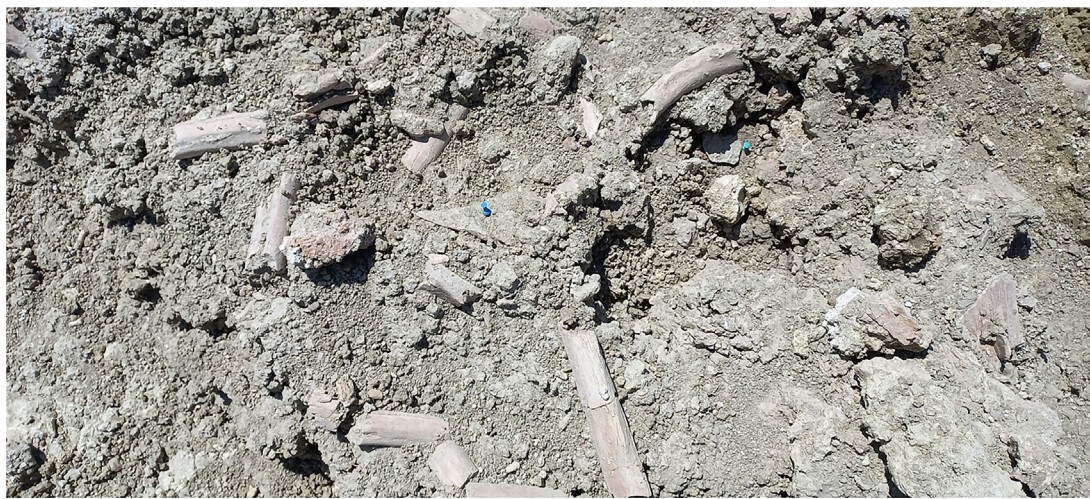

**Appendix 1—figure 2.** Fragmented and anatomically connected ribs from the proboscidean individual at EAK.

**Appendix 1—table 1.** Skeletal representation of the elephant carcass at EAK.

|  | NISP | MNE |
| --- | --- | --- |
| Cranium | 12 | 1 |
| Mandible |  |  |
| Teeth | 44 | 5 |
| Vertebrae | 1 | 1 |
| Ribs | 51 | 8 |
| Pelvis | 11 | 1 |
| Scapula |  |  |
| Humerus |  |  |
| Radius |  |  |
| Ulna | 1 | 1 |
| Carpals |  |  |
| Metacarpals |  |  |
| Femur | 5 | 2 |
| Tibia | 2 | 2 |
| Patella | 1 | 1 |
| Calcaneum | 2 | 1 |
| Astragalus | 1 | 1 |
| Tarsals | 5 | 5 |
| Metatarsals | 5 | 5 |
| Phalanges | 12 | 12 |

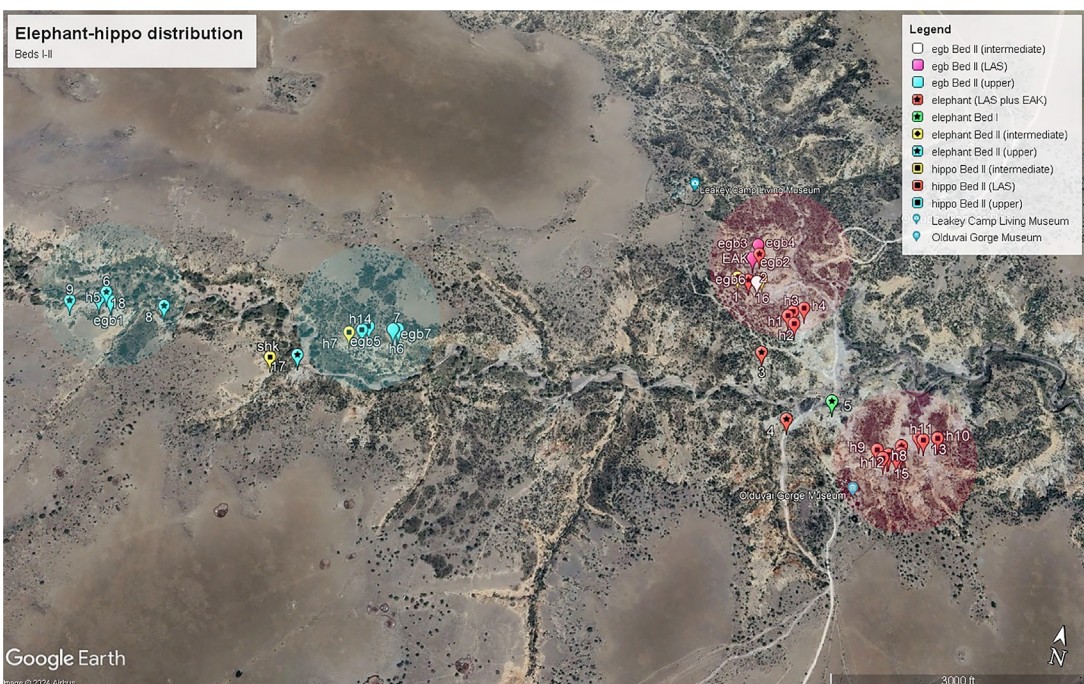

**Appendix 1—figure 3.** Distribution of elephant and hippopotamus carcasses at the main section of Olduvai Gorge, documented in the survey conducted in June 2024. The lower (red), intermediate (yellow), and upper (blue) stratigraphic units display a clustered distribution of carcasses. Notice the spatial overlap of those megafaunal clusters with the dense landscape-scale concentrations of stone artifacts (in faded red for the LAS lower unit and faded blue for the upper unit). Elephant carcasses appear numbered and hippopotamus carcasses start with 'h' before the number. EGB, elephant green-broken bone. The megafaunal site of TK Sivatherium is outside the map.

## Stratigraphically associated green-broken proboscidean bones (see *Appendix 1—figure 3*)

A small collection of green-broken elephant bones was retrieved from the lower and upper Bed II units. They appear here as examples of green-broken megafaunal bones found in the three paleolandscape units targeted. One of them, a proximal femoral shaft found within the LAS unit, has all the traces of having been used as a tool (*Figure 6*). It displays green breaks on all its sides, with conchoidal fractures observed both on the cortical and medullary surfaces (*Appendix 1—figures 5–7*). This indicates multiple dynamic loading events, reflected in abundant step fractures, ripple marks, and multiple, same-direction curved outlined step micro-conchoidal scars on the same front of its proximal end. This, together with their association to a polished tip bearing more abrasion than the rest of the bone, is suggestive of the bone specimen having been used as a tool (*Figure 6*). An equally compelling case of anthropogenic agency was found on another specimen found a few meters away (probably suggesting butchery of the same individual elephant), with a complete and smooth set of two spiral overlapping green fractures (*Appendix 1—figure 4* middle), and a series of step fractures on the breakage plane of the alternative break (*Appendix 1—figure 4* lower), finalized with a percussion mark on the cortical surface of the spiral break (*Appendix 1—figure 4* upper).

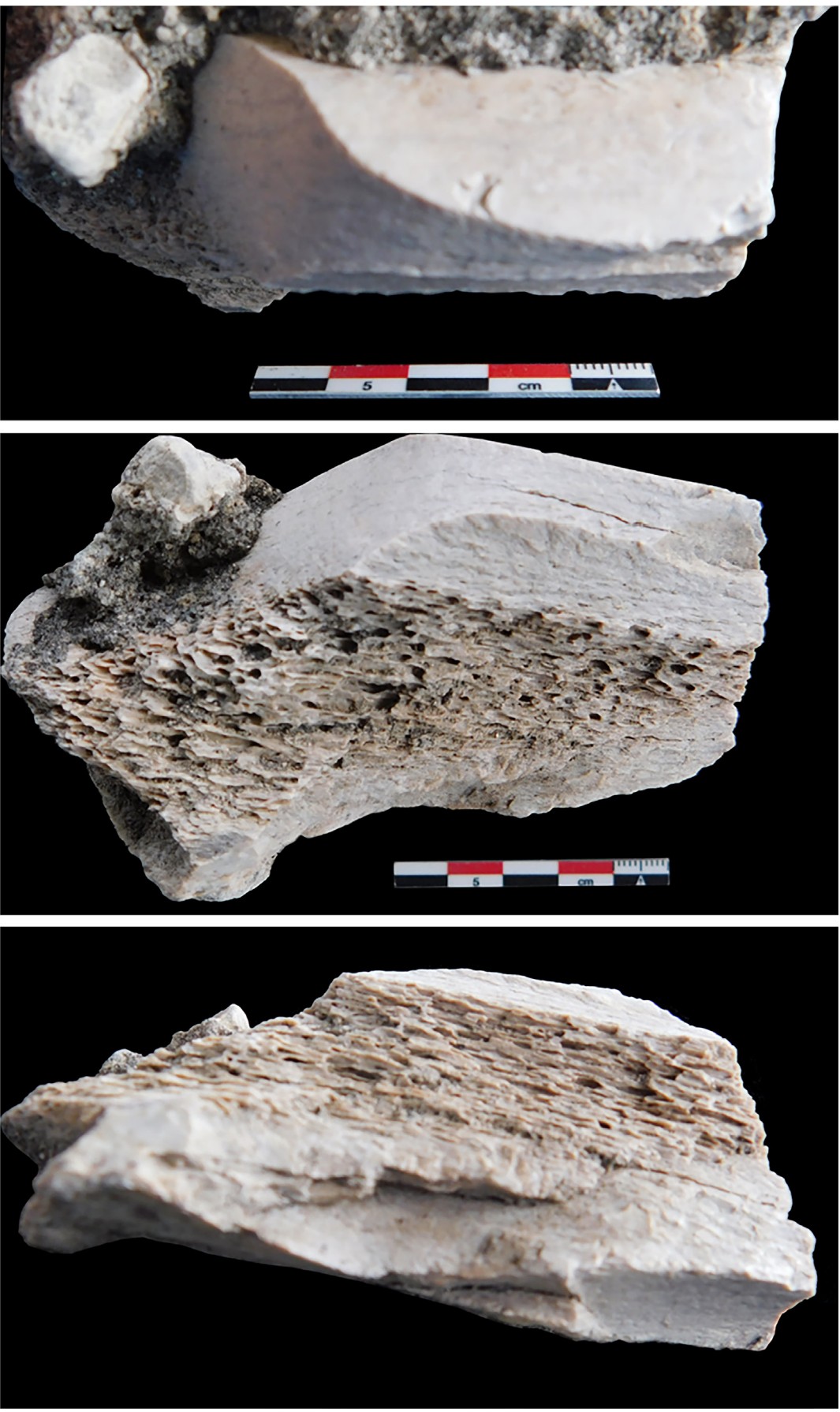

**Appendix 1—figure 4.** Example of elephant green-broken limb bone shaft displaying percussion mark (upper) and overlapping reflected/stepped scars on its medullary surface resulting from hammerstone breaking.

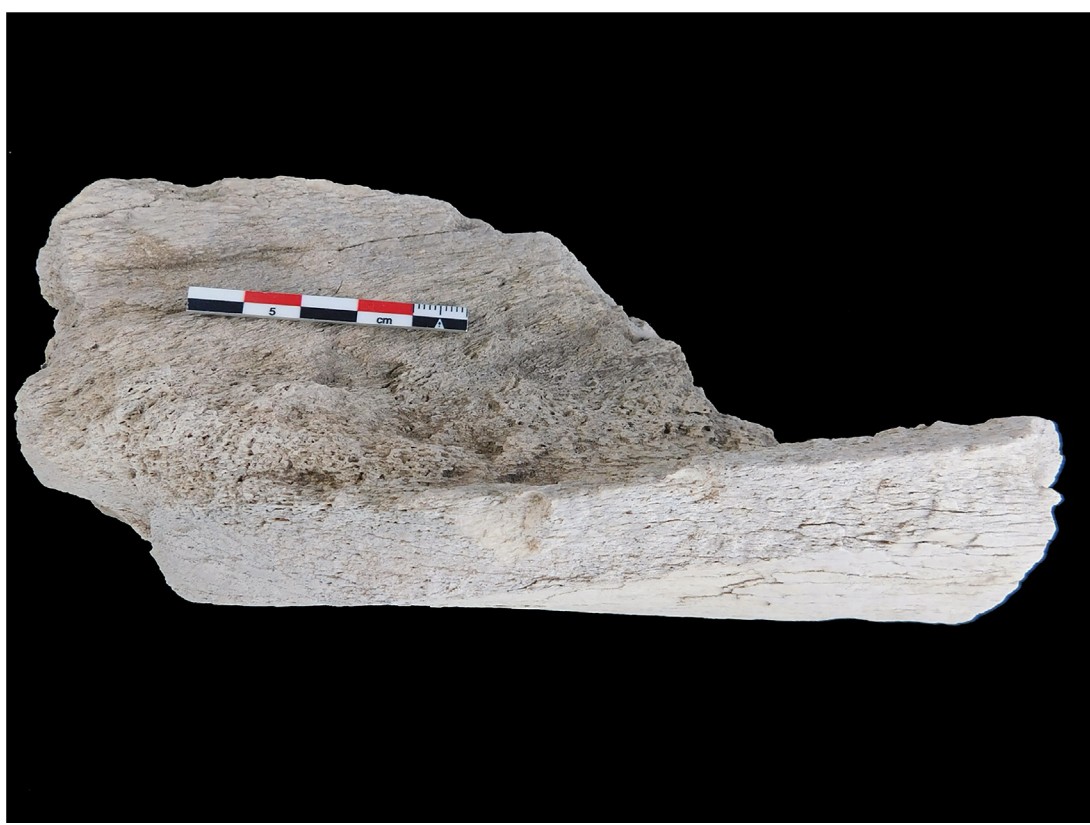

**Appendix 1—figure 5.** EGB1: proboscidean limb bone shaft with green breaks found at BK (upper unit).

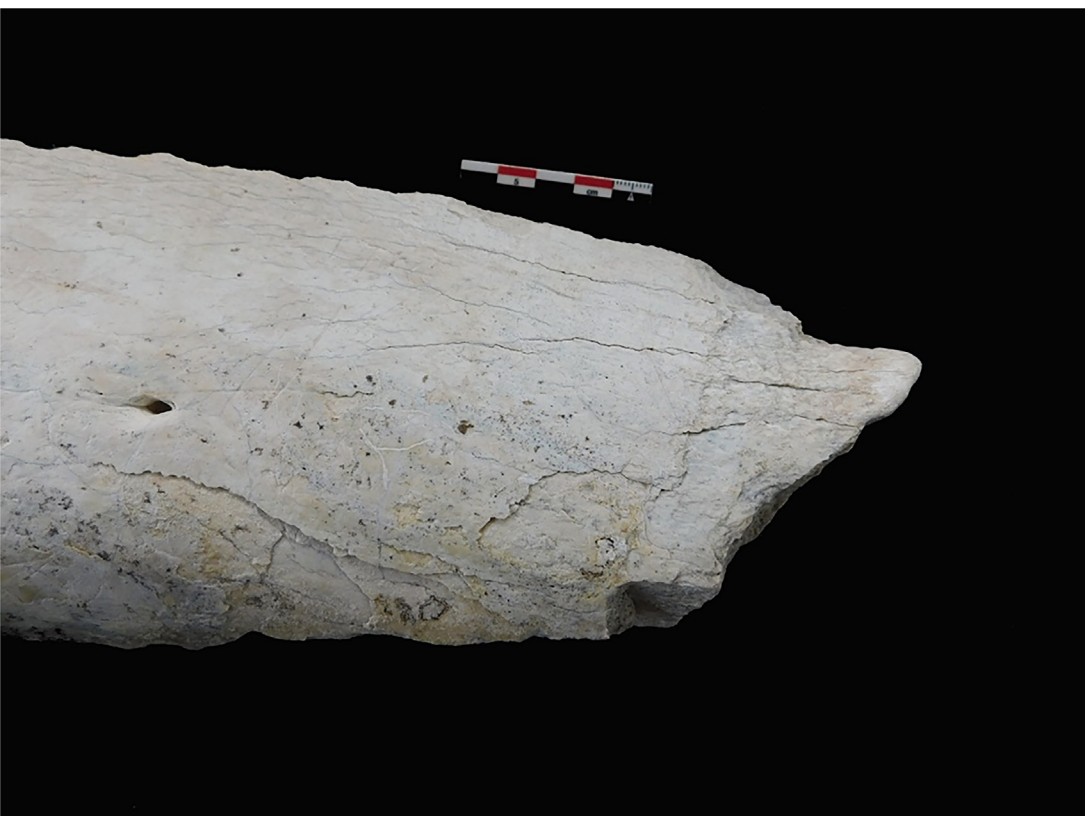

**Appendix 1—figure 6.** EGB 2: proboscidean femoral shaft with green breaks and a distal portion showing a combination of curved scars and a polished tip, most likely polished anthropogenically.

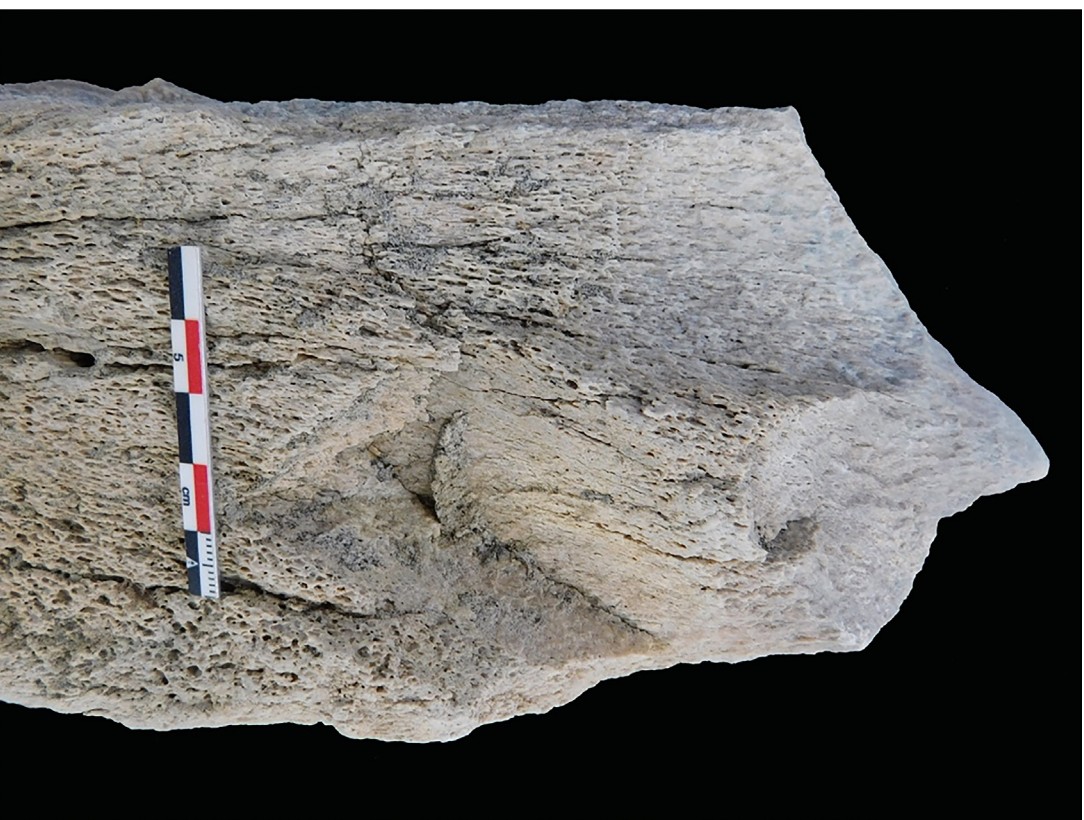

**Appendix 1—figure 7.** EGB 2: medullary view of the same proboscidean specimen displaying green breaks and step fractures on its proximal portion.

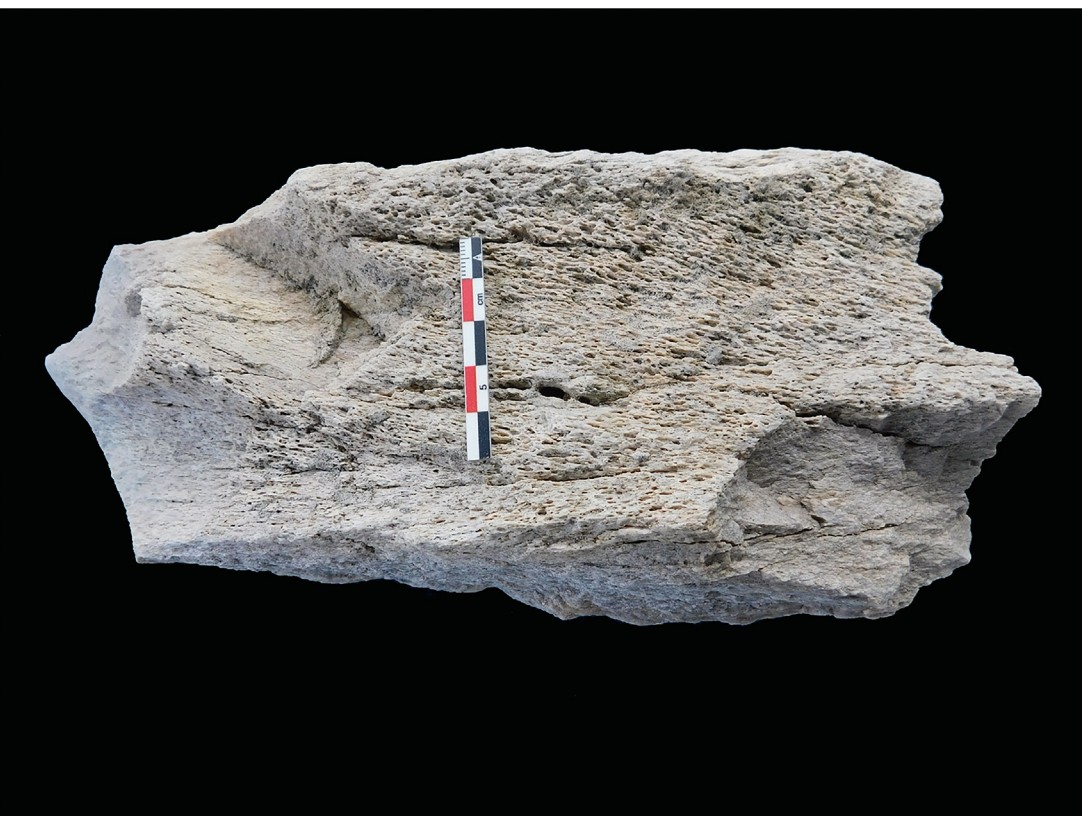

**Appendix 1—figure 8.** EGB 2: same specimen showing green fractures on all its sections, as seen from a medullary perspective. All breaks show sharp edges in contrast with the polished tip.

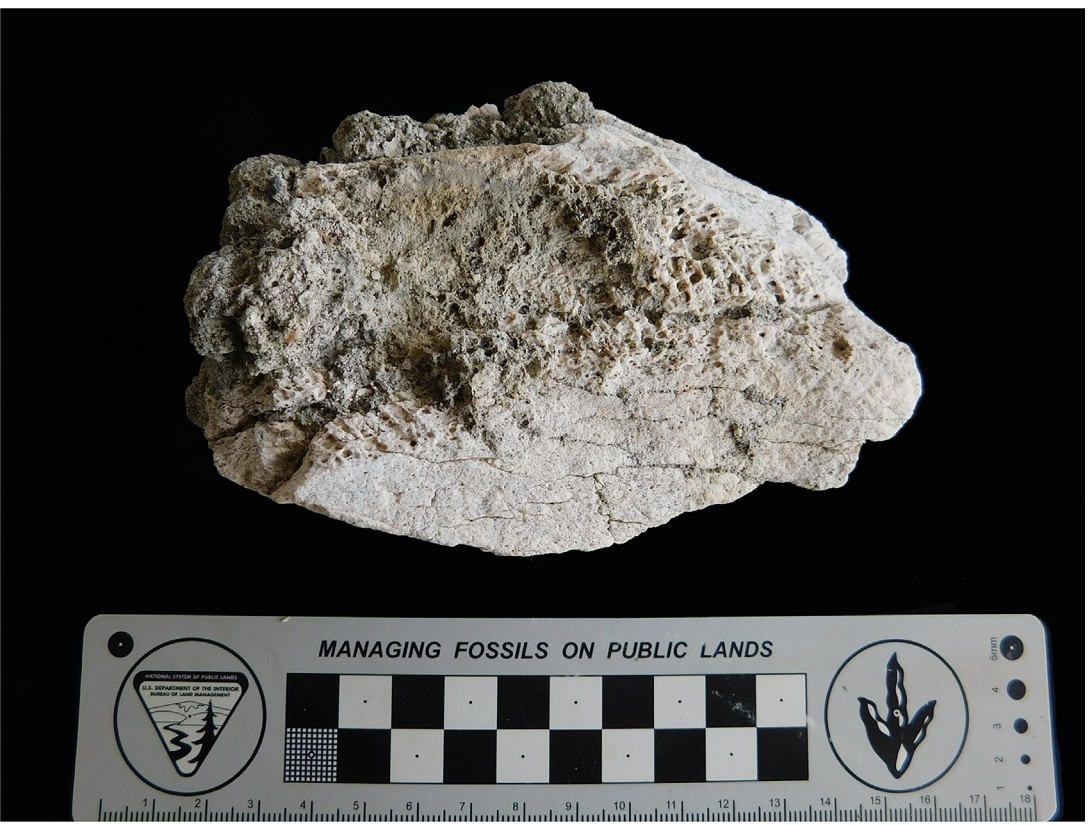

**Appendix 1—figure 9.** EGB3: proboscidean limb shaft with smooth green breaks. It was found in the LAS lower unit.

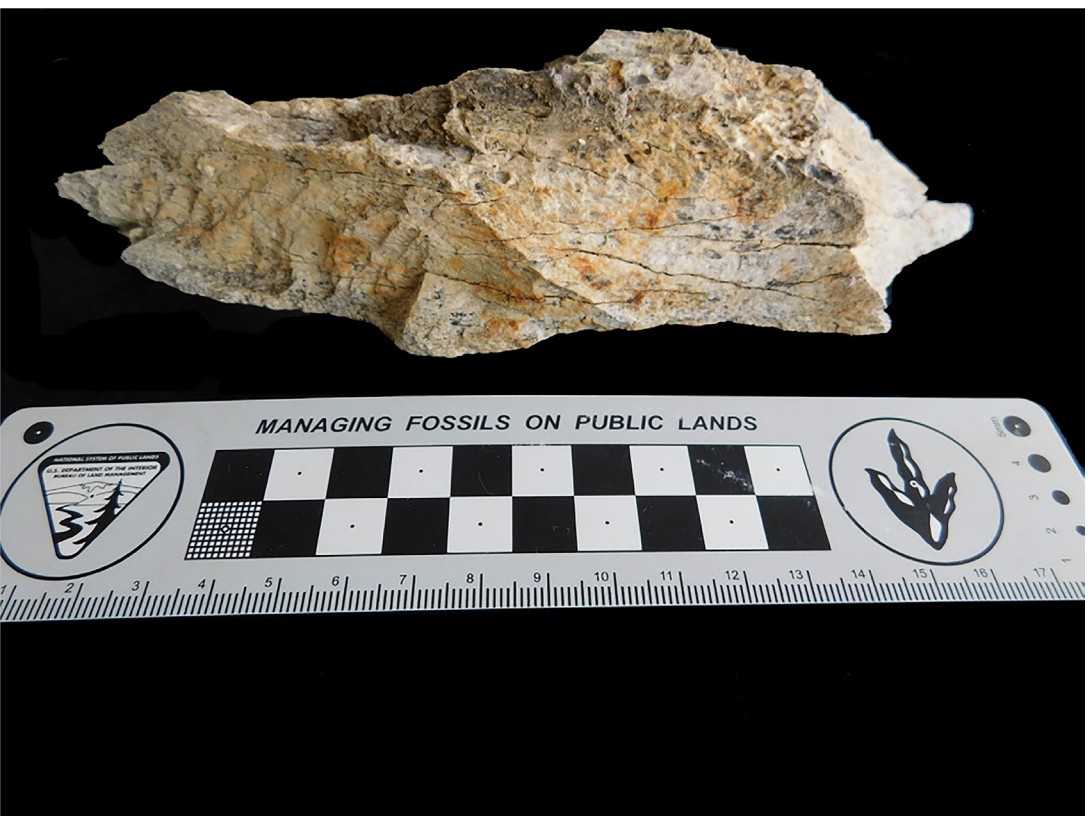

**Appendix 1—figure 10.** EGB4: proboscidean limb shaft fragment with green breaks showing multiple overlapping conchoidal scars and hackle marks typical of dynamic loading. It was found in the LAS lower unit.

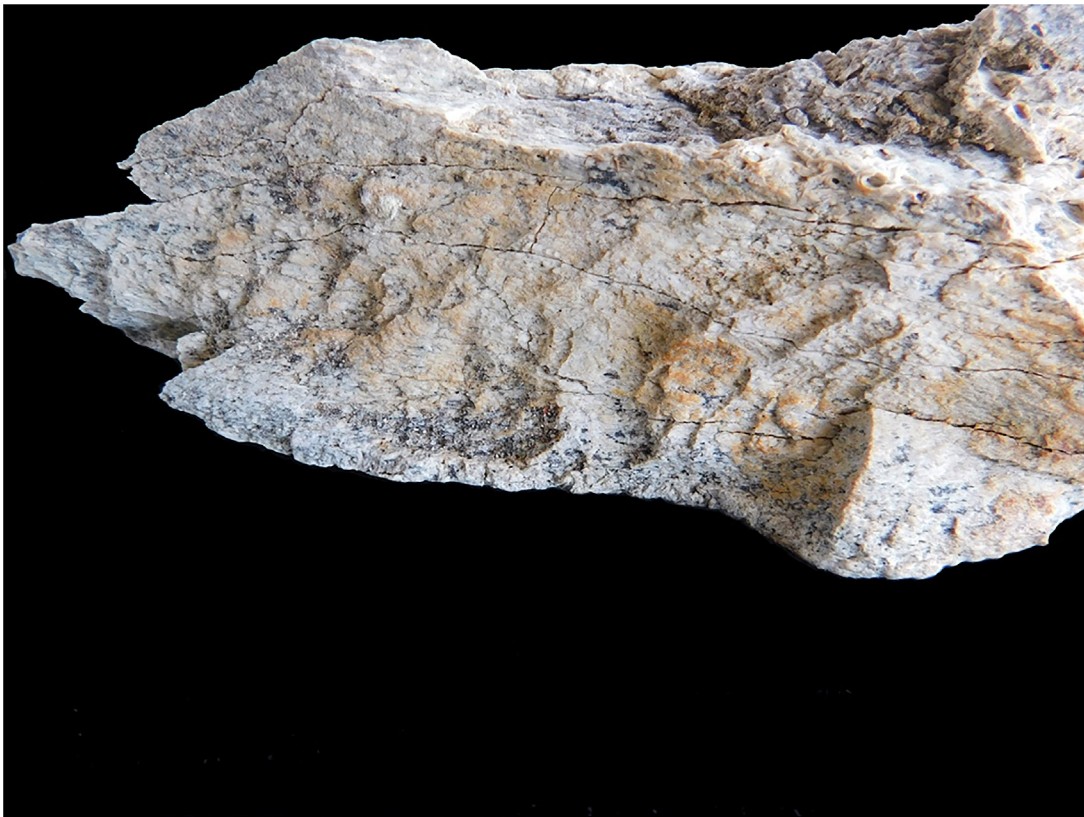

**Appendix 1—figure 11.** EGB4: detail of the hackle marks of the same specimen, typical of dynamic loading.

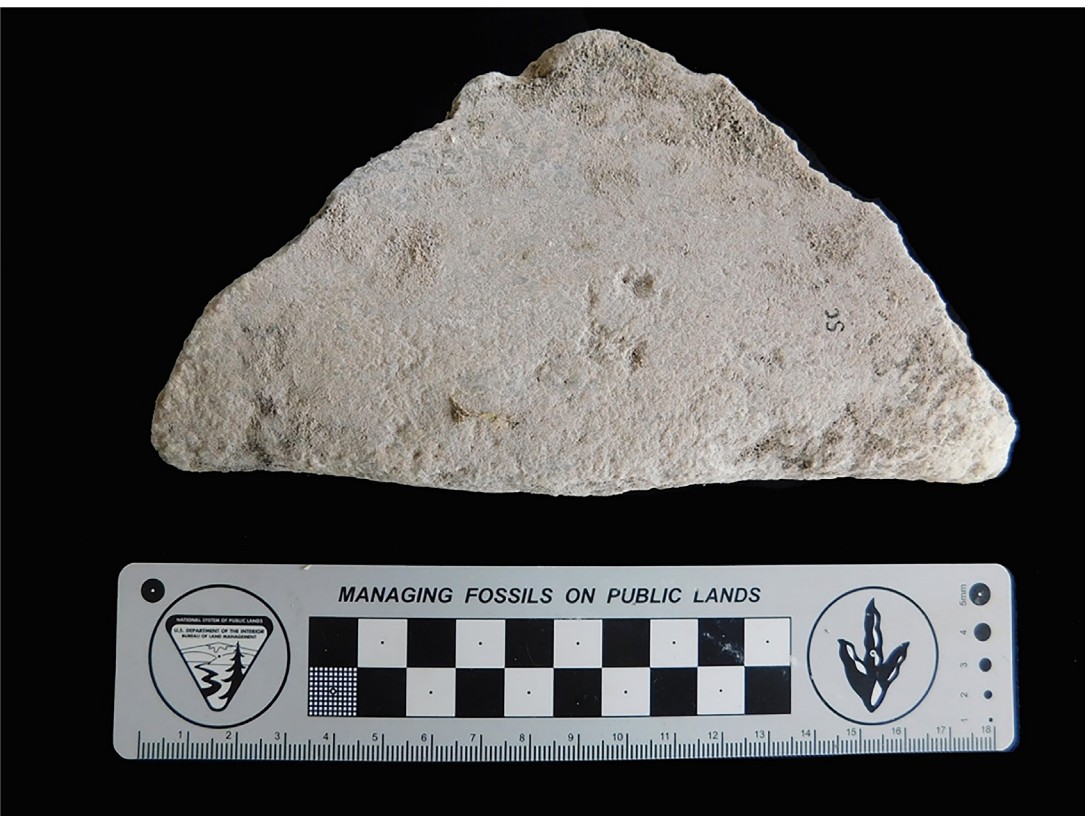

**Appendix 1—figure 12.** EGB5: Probable proboscidean flat bone found at SC (upper unit). It has attributes of green breakage and some pitting on its cortical surface, possibly resulting from percussion.

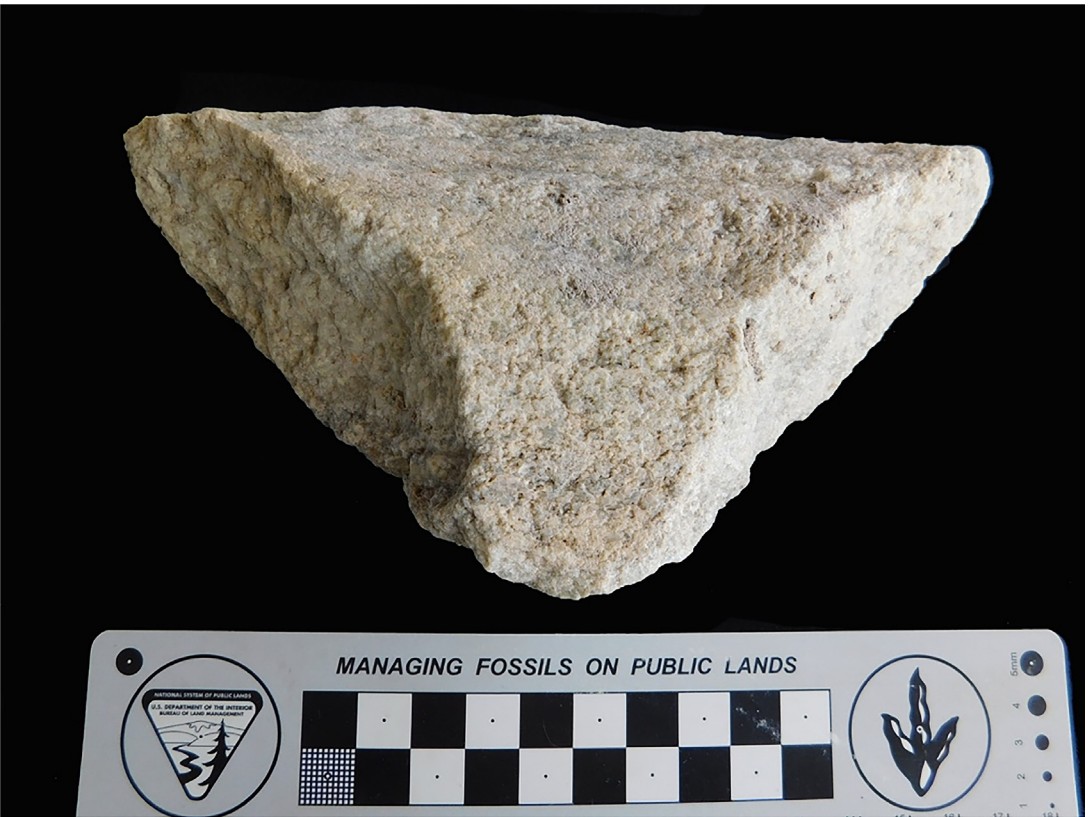

**Appendix 1—figure 13.** EGB 5: same specimen seen from the other side.

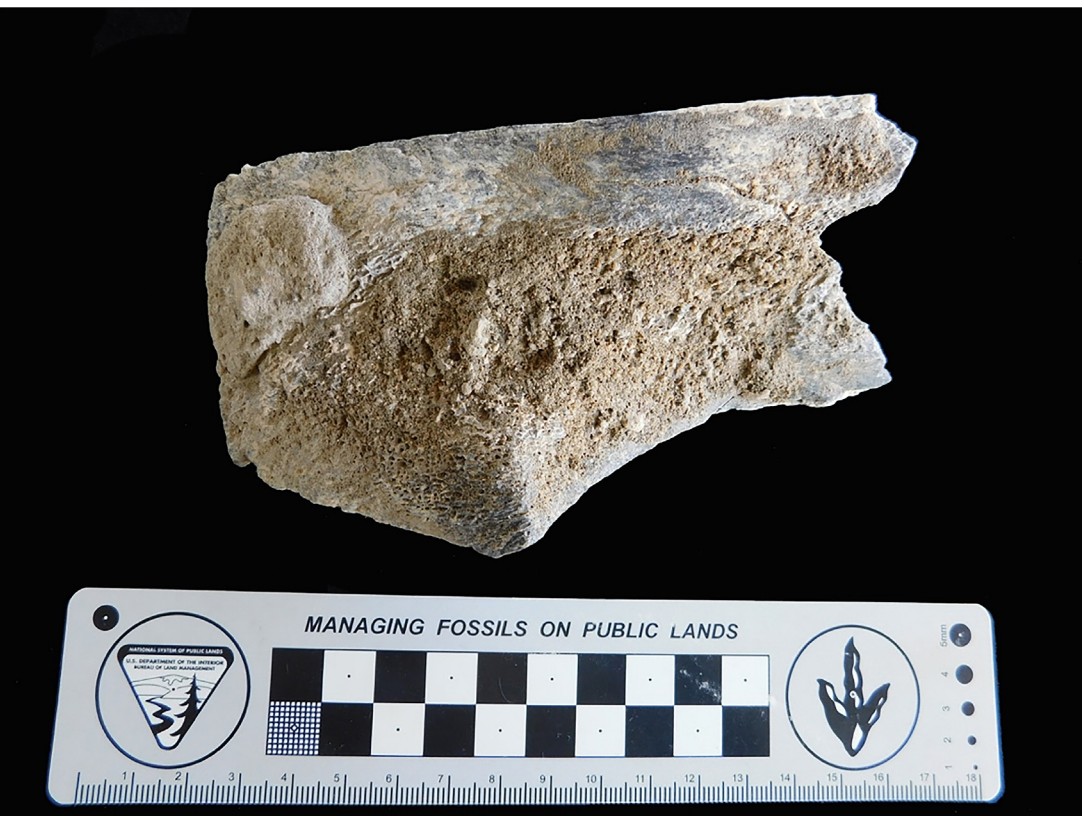

**Appendix 1—figure 14.** EGB 6: medullary view of possible proboscidean limb shaft fragment with green break morphology, found in the intermediate unit in the vicinity of FLK west.

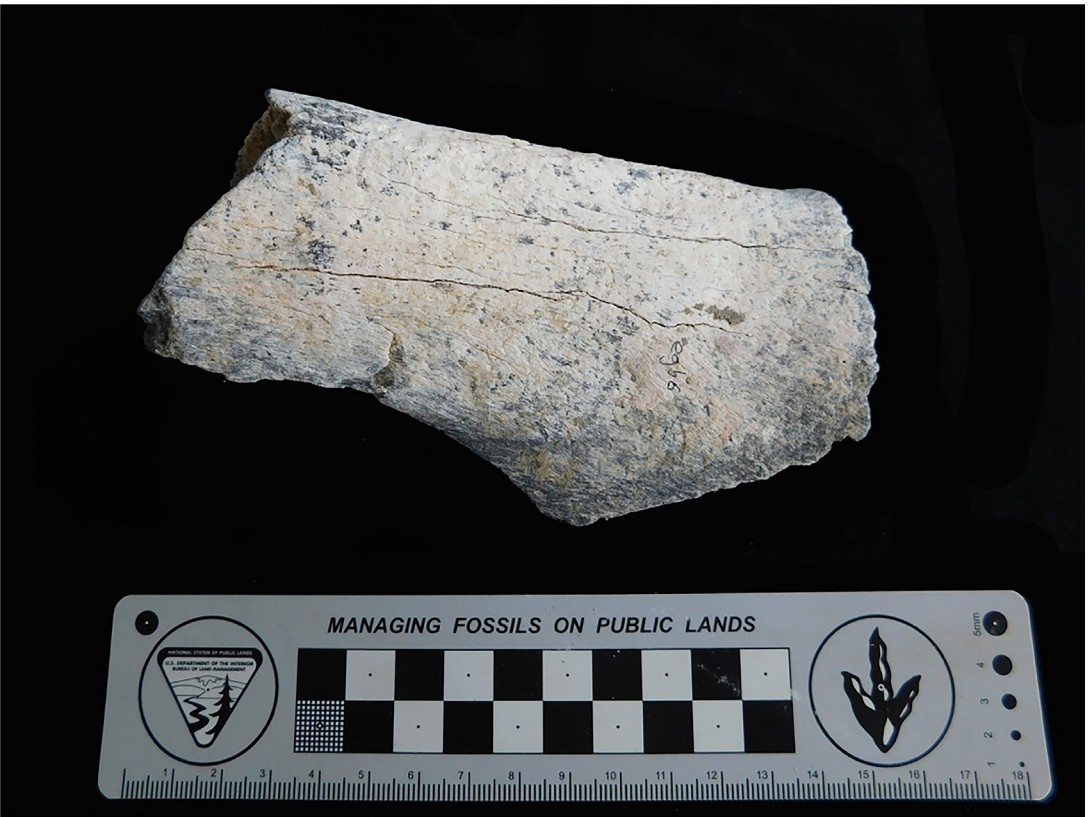

**Appendix 1—figure 15.** EGB 6: same specimen seen from its cortical side. Notice the pitting on its surface near the notch.

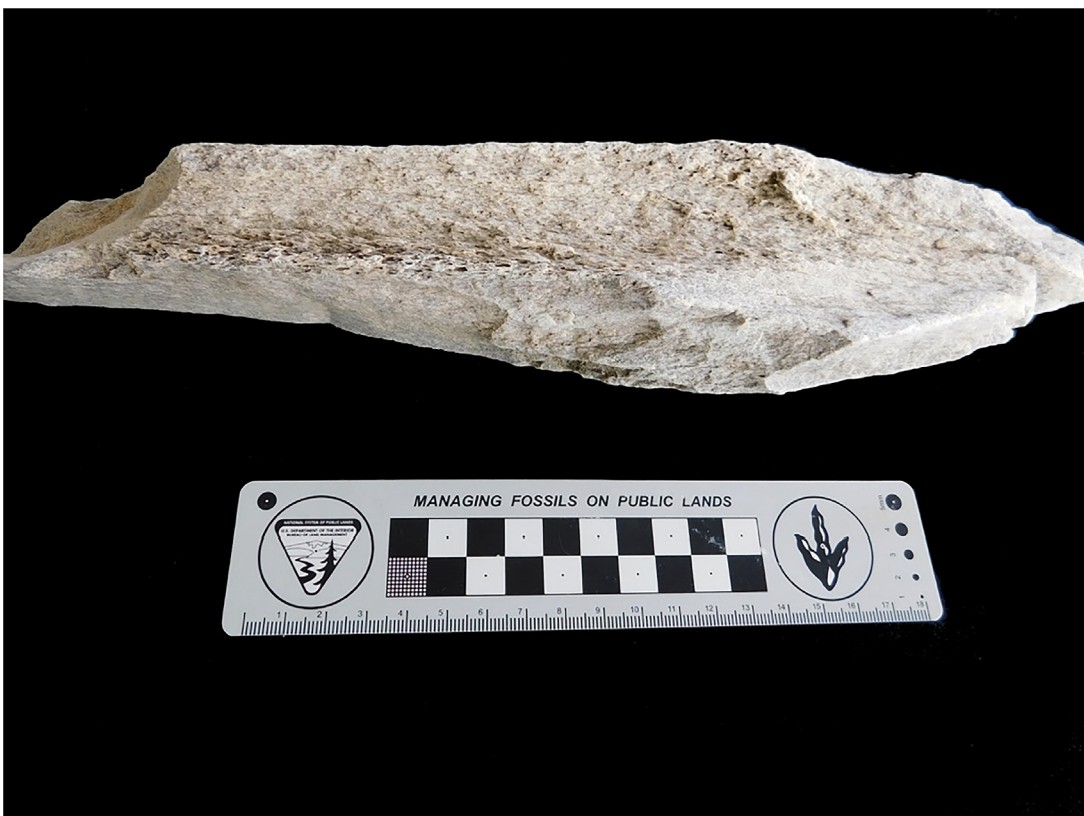

**Appendix 1—figure 16.** EGB7: proboscidean limb shaft fragment (probably from a femur) with green break attributes, showing multiple overlapping conchoidal scars (center) found at SC (upper unit).

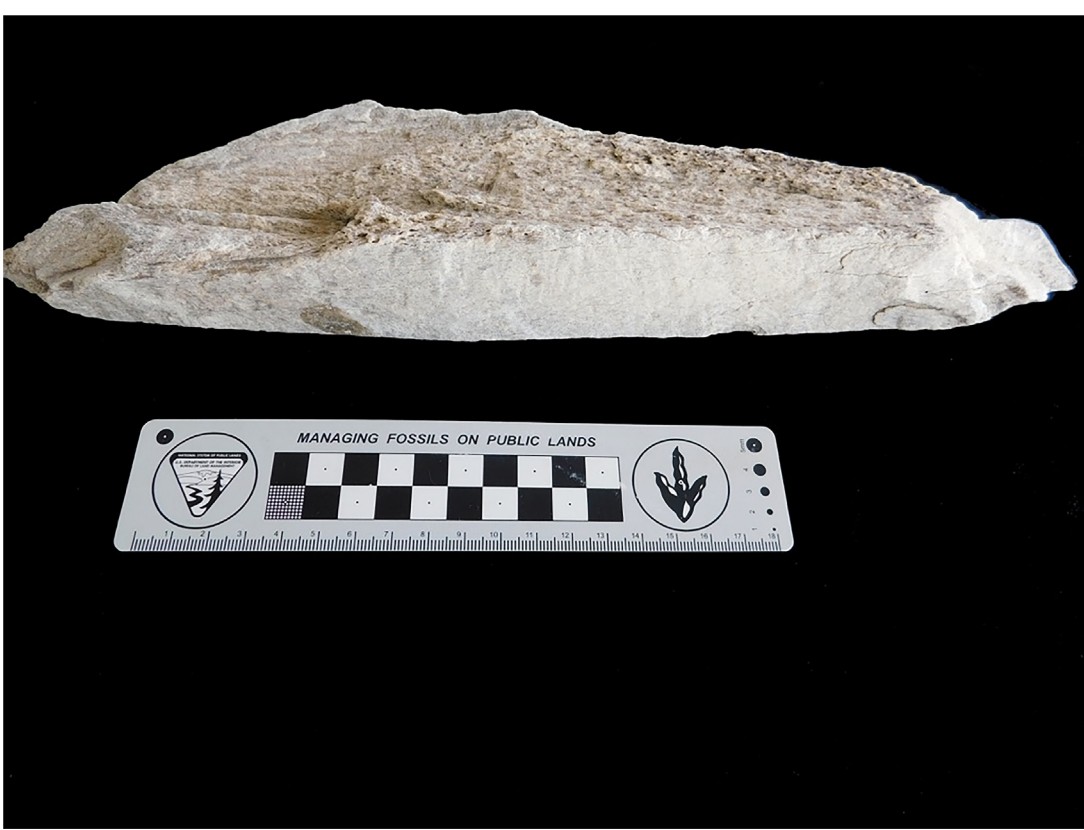

**Appendix 1—figure 17.** EGB7: other side of the same proboscidean limb shaft fragment with attributes of green-bone breakage, such as smooth fracture surface and conchoidal scars, found at SC (upper unit).

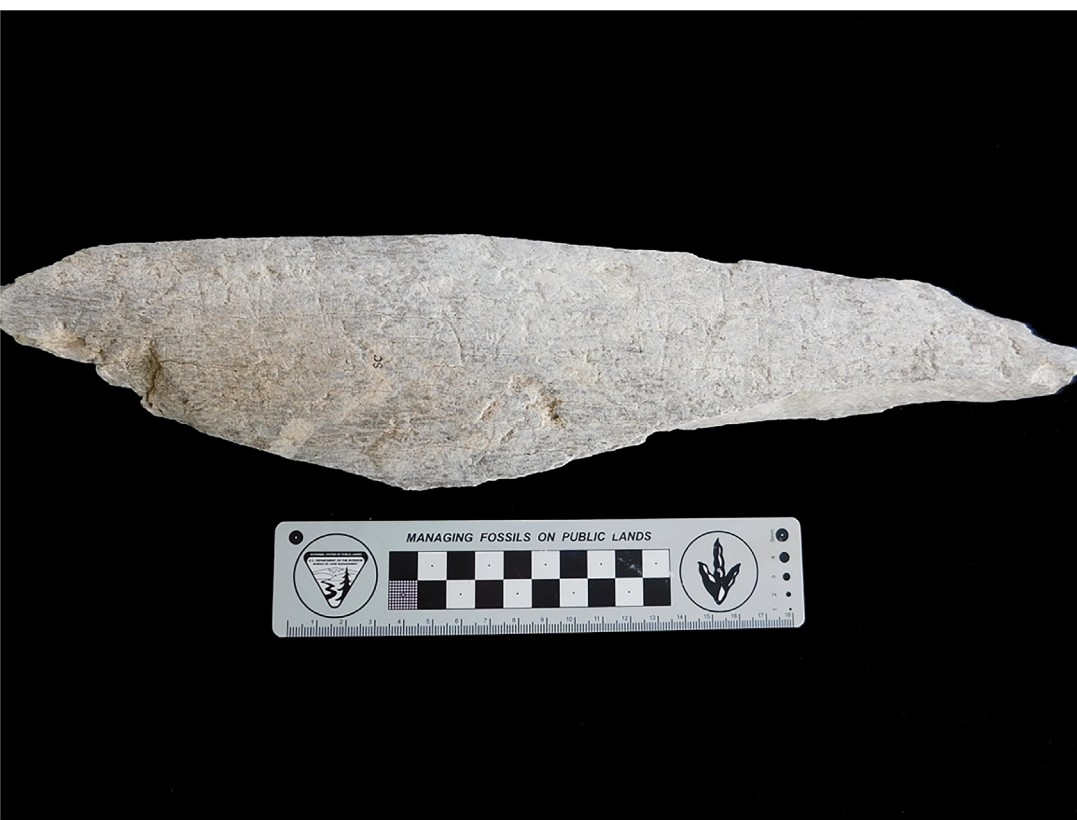

**Appendix 1—figure 18.** EGB7: same proboscidean limb shaft fragment, cortical view. Note the marks from abrasive agencies, plus some pitting by both lower edges of breakage planes suggestive of the potential dynamic loading effector.

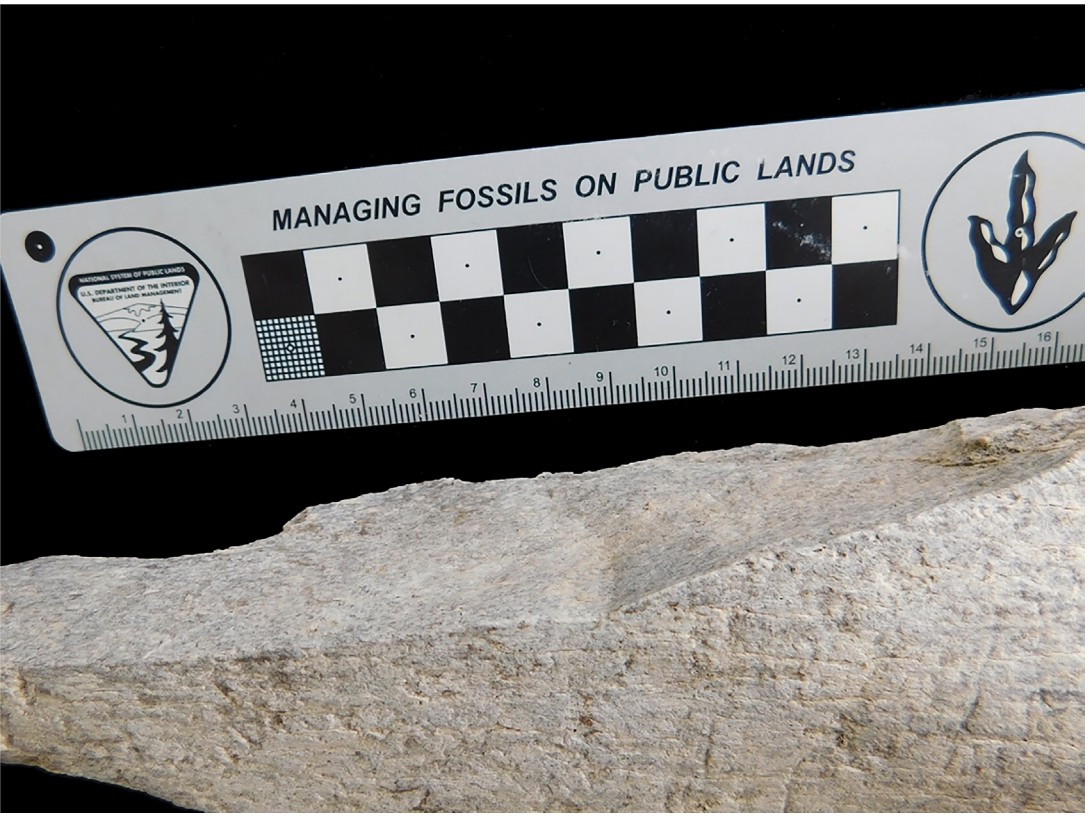

**Appendix 1—figure 19.** EGB7: detail of the same limb shaft fragment showing the longest smooth fracture surface typical of a green break with inflection point in the middle.

## Green-broken specimens with uncertain stratigraphic attribution

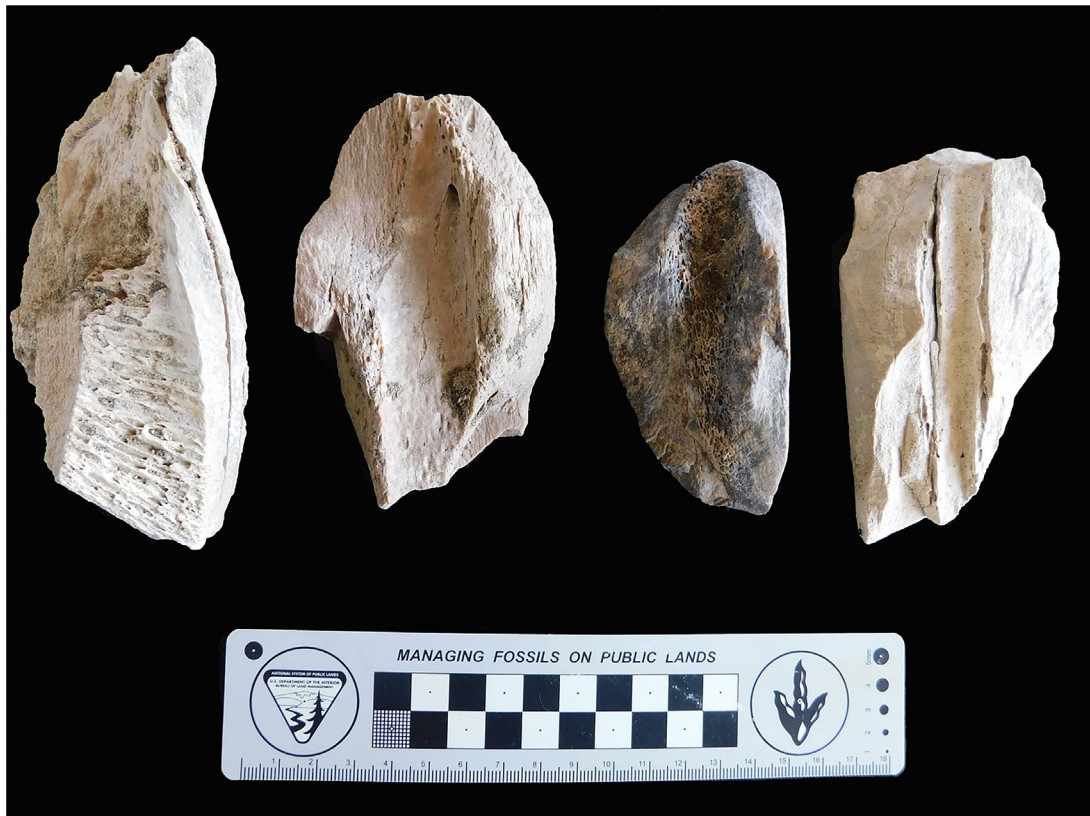

**Appendix 1—figure 20.** Example of other megafaunal long bone shafts showing green breaks.

## Green-broken bones at EAK

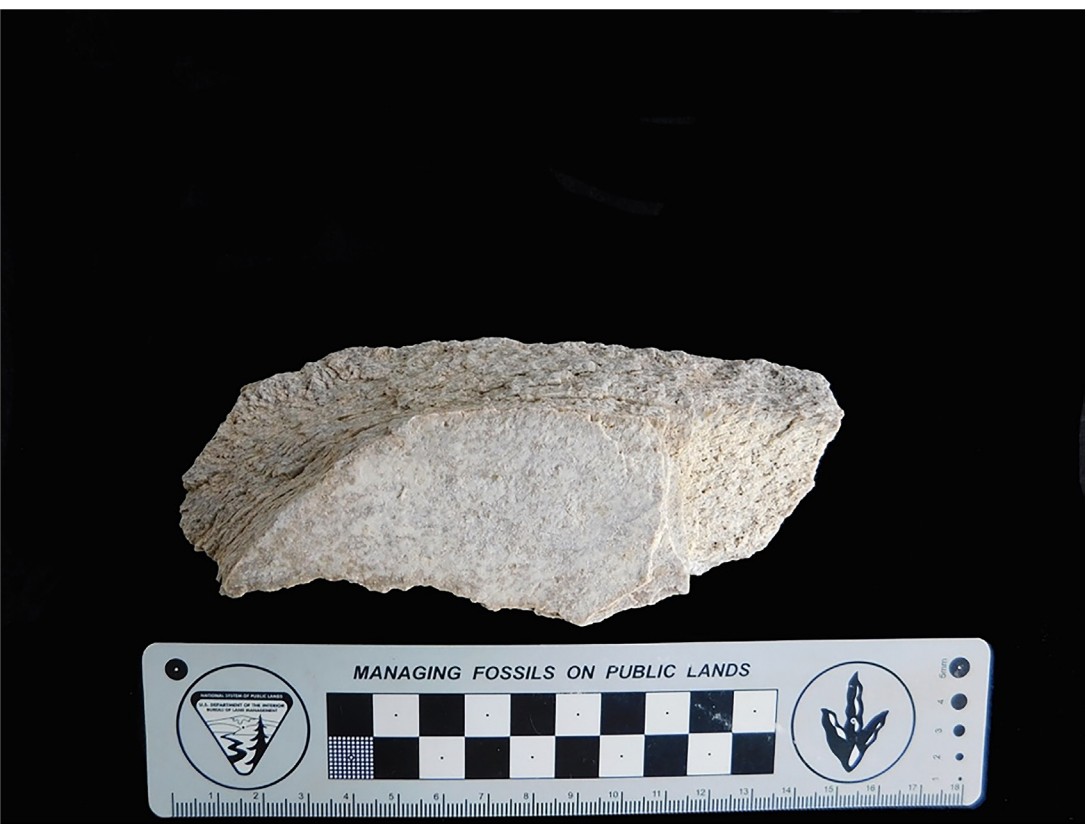

**Appendix 1—figure 21.** Elephant-sized flat bone fragment with green-bone curvilinear break.

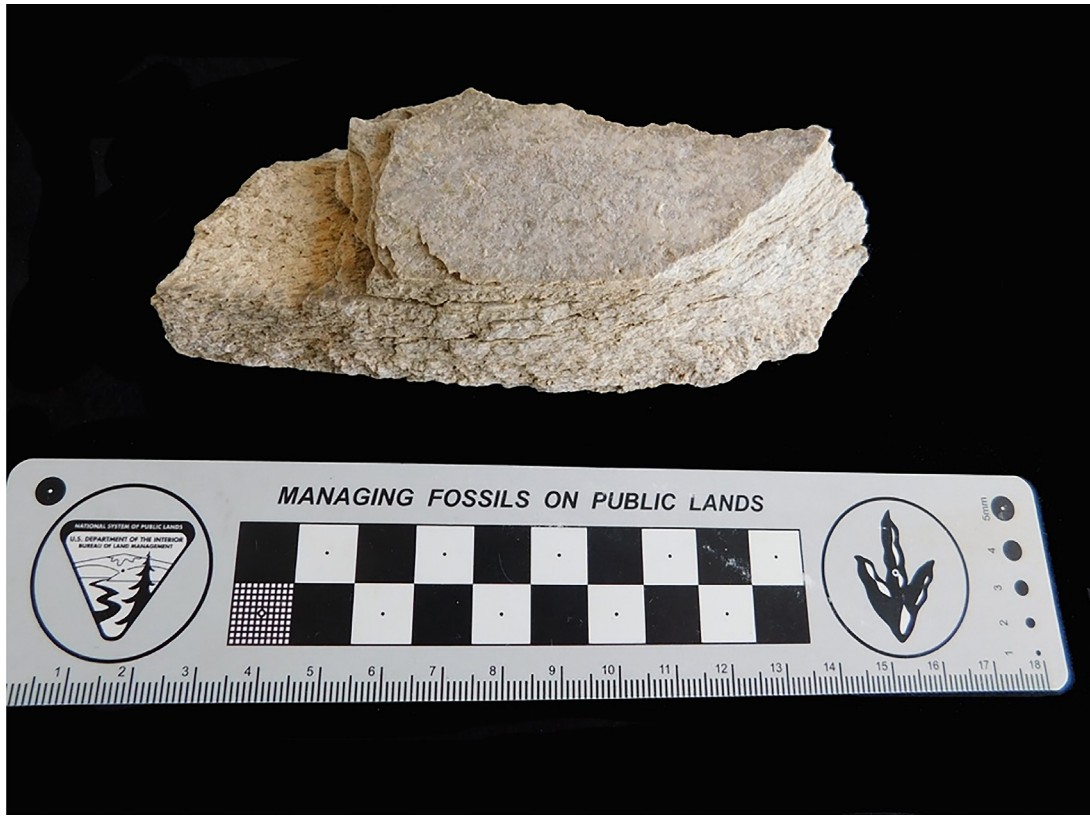

**Appendix 1—figure 22.** Same specimen with a different view.

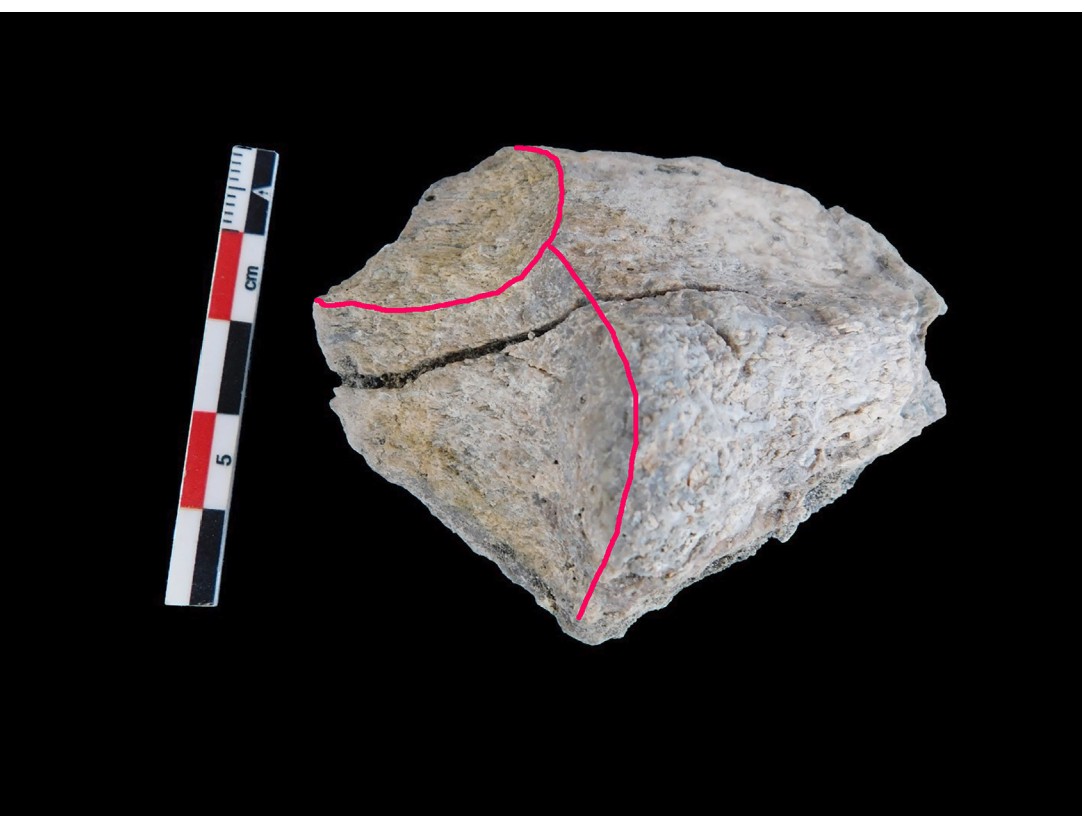

**Appendix 1—figure 23.** Proboscidean limb bone fragment found at EAK, with morphology of a green-bone break. Red lines show the outline of the overlapping scars.

The spatial statistical analyses of EAK

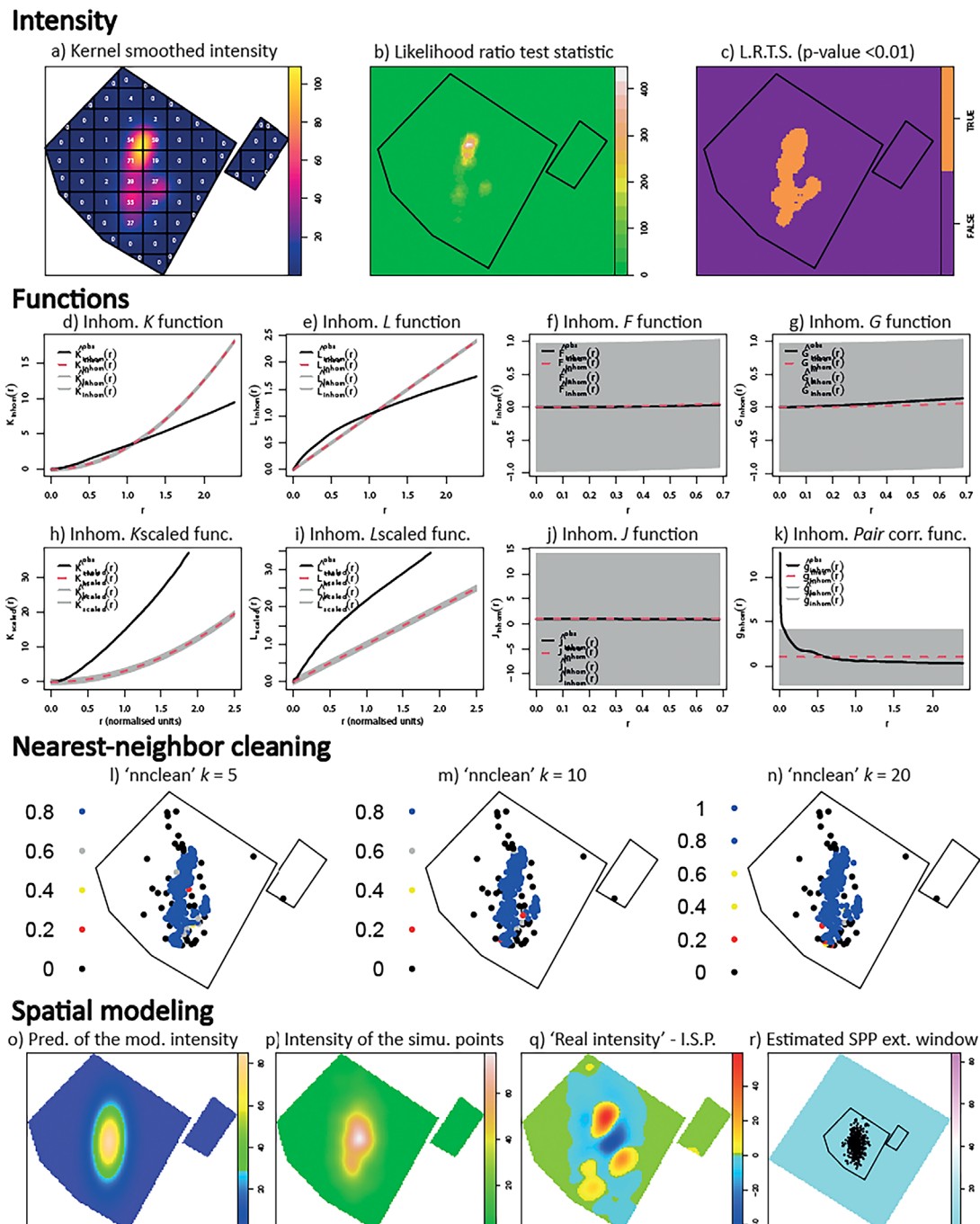

**Appendix 1—figure 24.** Mapping of intensity, nearest-neighbor cleaning, and spatial modeling of the SPP of bones at EAK, including graphs of the inhomogeneous second-order functions.

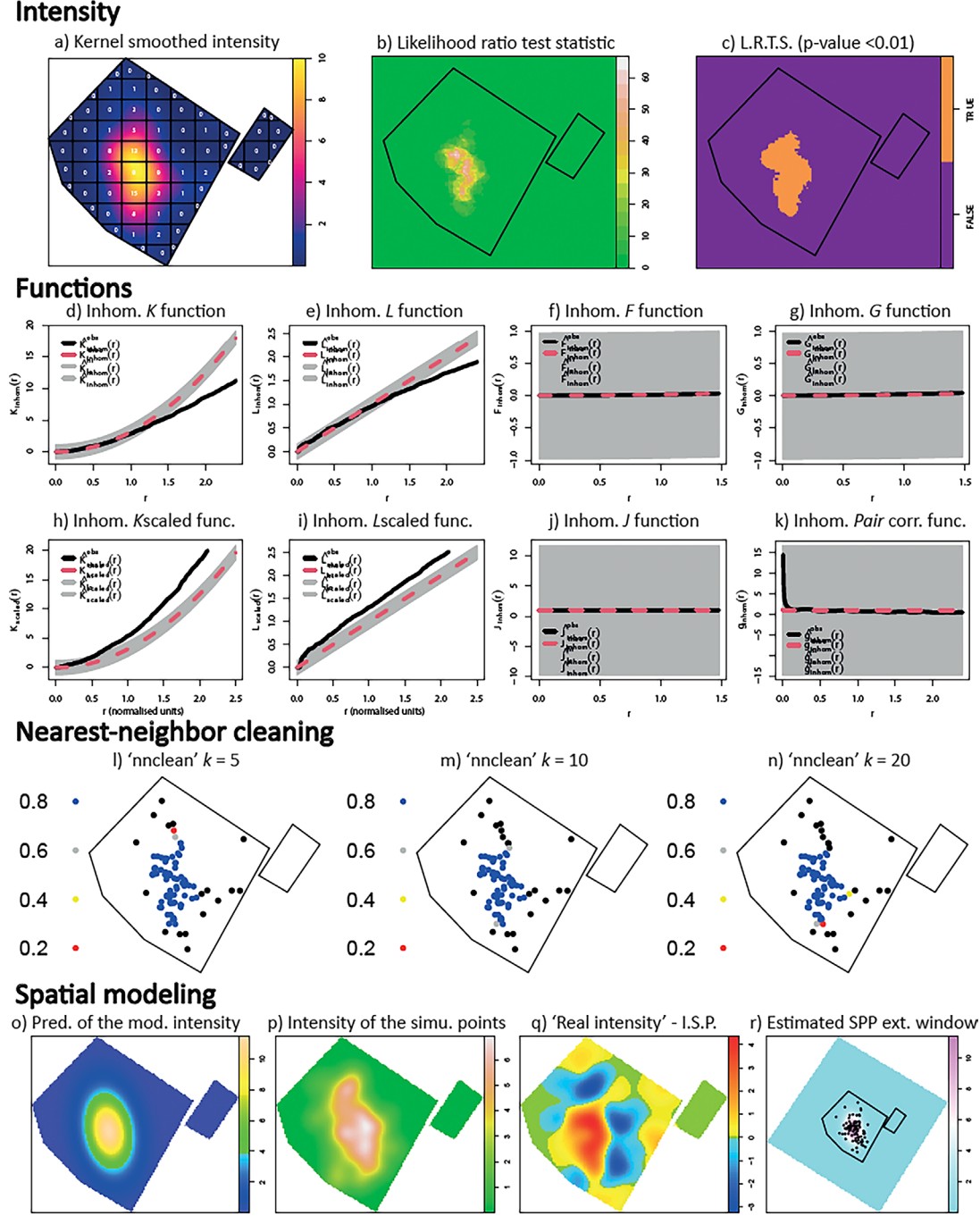

**Appendix 1—figure 25.** Mapping of intensity, nearest-neighbor cleaning, and spatial modeling of the SPP of lithics at EAK, including graphs of the inhomogeneous second-order functions.

## The technological analysis of EAK

### Materials and methods of the lithic analysis

A collection of 80 lithic specimens has been retrieved from EAK. For the study of these materials, we have taken into consideration the following technological categories and procedures implied in the analysis (*Diez-Martín et al., 2024a*; *Diez-Martín et al., 2017*; *Diez-Martín et al., 2009a*; *Diez-Martín et al., 2009b*; *Diez-Martín et al., 2011*): (a) Unmodified specimens are cobblestones devoid of any sign of anthropogenic manipulation; (b) percussion materials, include hammerstones or cobbles showing battering and/or scarring on their surface; (c) cores. Handheld cores have been

classified according to the combination of the following traits: number of reduction surfaces (unifacial, bifacial, and multifacial); number of striking platforms (unipolar, bipolar, multipolar); arrangement of the identified knapping series (lineal, opposed, orthogonal, centripetal). Bipolar cores have been defined through the technical traits related to anvil percussion; (d) Detached products include both diagnostic bipolar and handheld complete and broken flakes. Technical traits recorded in flakes include reduction stages (according to the variable retention of cortex on dorsal areas and striking platforms), striking platform type, and dorsal scar arrangements; (e) Waste material includes the by-products of the knapping processes: non-diagnostic flake fragments with signs of direct handheld percussion, shatter or blocky/angular undetermined detached fragments, core-like fragments, and ≤20 mm debris.

## Results

All the specimens but one have been preserved in mint fresh conditions, lacking any signs of post-sedimentary alteration (R0). An indeterminate shatter fragment shows severe abrasion on surfaces and edges (R3). The bulk of the sample is constituted by quartz specimens (96%), while the sum is formed by few lavas (two basalt cobbles and a phonolite flake). *Appendix 1—table 2* shows the distribution of lithics sorted by category and raw materials. In general, the EAK collection is a meager sample with a limited expression of knapping behaviors, as all the categories with technological significance are represented by few specimens. Non-diagnostic waste is the most significant category, mostly ≤20 mm debris and flake fragments.

In the following description, lithic categories are grouped by operational sequences (techno-economic actions sharing common goals):

1. Unmodified material. Natural and non-modified materials are represented by a low-quality vesicular basalt amorphous cobble (87x59 × 52 mm, 296 g) showing no sign of modification on its surfaces and, thus, devoid of anthropogenic use.
2. Percussion activities. Percussive actions are represented by a single high-quality basalt hammer-stone. The specimen is oval and shows significant signs of battering on its whole perimeter (proximal and distal ends, and lateral areas; *Appendix 1—figure 26.2*).
3. Production of small and medium-sized flakes. Fractions of the flake production sequences have been documented on quartz and, very partially, on phonolite:
   a. Cores represent 5% of the total sample. Two knapping methods have been identified on quartz blanks. Direct handheld percussion is represented by an exhausted multifacial specimen showing at least 10 negative scars detached from multiple planes. Considering its advanced stage of exploitation, the last flakes produced from this core are significantly small (mean length and breadth 24x28 mm) (*Appendix 1—figure 26.4*). The bipolar on anvil technique is represented by three cubic quartz specimens in an advanced phase of reduction (mean 66x52 × 41 mm, 225 g), most probably tabular slab fragments, showing opposed bipolar scars detached from platform and base that, in two cases, show circular or semi-circular core rotation (*Appendix 1—figure 26.3* and *Appendix 1—figure 26.5*). Two specimens show signs of frosting on their platform areas. When properly identified in negative scars (n=4), flakes detached from bipolar (mean 34x40 mm) cores are larger (significantly wider) than those observed in the freehand multipolar specimen (mean 24x28 mm) and one of the flakes removed from a bipolar core can be included within the medium-sized flake group (breadth 63 mm; *Appendix 1—figure 26.1*).
   b. Detached material. Most of the complete and broken diagnostic flakes retrieved have been assigned to direct handheld percussion (83%), while the sum is represented by specimens showing clear traits of bipolar knapping. As for the former method, the few broken specimens show longitudinal axial snaps (a very common fracture accident observed in the Naibor Soit raw material), although a side-proximal snap has also been identified. The mean size of broken flakes is 32x23 × 10 mm and 8 g. Most complete flakes retrieved fall within the small size group (≤50 mm of maximum length), as mean flake size is 33x29 × 11 mm. With a length of 58 mm, only one retrieved flake exceeds this range and, thus, can be included in the medium-sized flake range (*Appendix 1—figure 27.2*). Size and mass parameters of complete flakes are detailed in *Appendix 1—table 3*. The phonolite flake included in this category fits well within the quartz size pattern and with the

most representative technical traits described for quartz specimens (*Appendix 1—figure 27.1*). Regarding their position in the reduction sequence, 90% of the flake sample is included in the full production stage, this is to say, flakes with no cortex retention on both dorsal surface and butt (Type 6). The remaining flakes partially retain cortex on the dorsal surface (Type 5, n=1) or striking platform (Type 3, n=1). Striking platforms are clearly dominated by unifaceted butts (73.7%), although other non-cortical butts are also represented (lineal and point striking platforms). A single cortical butt has been identified in a specimen with no cortex on the dorsal area (Type 3). Mean breadth and thickness of striking platforms is 19x8 mm (range 11–31 mm and 6–12 mm, respectively). Flake morphology in the frontal plane is preferentially rectangular (33%), quadrangular (28%), and triangular (22%) (*Appendix 1—figure 27.3*). With respect to dorsal patterns, mean negative scars observed in dorsal areas are 2.17 (interval 1–5). The directional arrangements of flakes, excluding those showing a single dorsal scar (n=4), follow a variety of patterns, although the most numerous are bidirectional opposed (longitudinal and lateral; *Appendix 1—figure 27.4* and *Appendix 1—figure 27.6*) and orthogonal (partial or full; *Appendix 1—figure 27.5*), both patterns accounting for 44% of the identified cases, respectively. These dorsal models fit well within the reduction strategies observed in the core sample. Natural and potentially usable cutting edges on flakes tend to be located on the side (60%), being specifically lateral (45%) or bilateral. Distal (35%) and transversal (5%) locations are also documented. Mean cutting edge length per flake is 32 mm (30 mm in lateral specimens, 58 mm in bilateral, and 27 mm in distal). The case of the single phonolite flake deserves a comment. As no volcanic cores have been retrieved from the site, this specimen might well have arrived at the site already produced from another area. This impression is reinforced by the fact that, in agreement with the dimensional pattern observed among the quartz flake collection, this small product shows a potentially usable cutting edge of 92 mm (summing bilateral and distal edges), far beyond mean edge length in quartz or even the maximum length recorded in specimens detached from this raw material (56 mm). A small collection of flakes clearly detached by means of anvil technique has been retrieved (with identification of platform and base areas, opposed detachments, and irregular ventral surfaces). Mean size of these flakes is 36 x 30 × 15 mm and 17 g, which shows that they tend to be particularly longer and thicker than freehand flakes. One specimen shows frosting on the basal area.

c. Waste constitutes the most abundant group of lithics. Most likely, these pieces are the by-products of both freehand and bipolar reduction. All these specimens have been obtained from quartz blanks. Most waste (47%) is constituted by small debris (mean size 16.6 x 11 × 6 mm and 1.5 g). Flake fragments represent 26% of this category and can be tentatively ascribed to handheld knapping, although no clear technical traits can be identified due to the relevance of snapping accidents in these pieces. Mean size of flake fragments is 25 x 20 × 11 mm and 4.5 g of mass. Shatter includes larger indeterminate positives with a blocky morphology whose mean size is 27 x 20 × 18 mm and 7 g. Finally, a small group of unspecific core fragments includes three angular, cubic, and non-diagnostic specimens that might be related to the exploitation of tabular slabs. Mean size of these specimens is 51 x 27 × 26 mm and a mass of 46 g.

## Conclusions

A small lithic collection has been retrieved from EAK. Although percussion activities carried out at the site have been documented, they are residual. A single hammerstone, with intense signs of battering, evidences this behavior linked either to the knapping processes or to other unspecific percussive tasks. The size and characteristics of the sample show that lithic behaviors at this spot are mostly related to the casual exploitation of quartz tabular slabs to produce small usable flakes. The preferred selection of this blank type is confirmed by the fact that all identified cores and purported core fragments show a cubic morphology, in agreement with the shape of natural slabs as found at the quartz outcrops. Considering their contribution to the core sample (n=3), it seems that most of these blanks were exploited following the anvil method. Although clear bipolar products are significantly less numerous than freehand flakes, it is important to note that a significant fraction of

waste (some debris, shatter, and fragments) is compatible with the bipolar reduction of quartz slabs. A single exhausted multifacial-multipolar core is indicative of the presence of handheld percussion. As might be the case of bipolar knapping, usable freehand flakes outnumber negative scar count in this exhausted core. However, considering that this specimen could have gone through an intense process of blank reduction, the imbalance between on-site detached and retrieved flakes could be considered moderate. By means of bipolar and freehand knapping, small-sized usable flakes with a mean cutting edge length of 32 mm were obtained. Considering that the mean maximum length of these small flakes is 33 mm, an efficient balance between detached flakes and usable natural cutting edges is observed. None of these natural edges shows signs of transformation or resharpening via retouch or trimming. Among the flake sample, a small-sized phonolite specimen has been retrieved. Considering both the lack of volcanic cores in the sample and the particularly efficient ratio between size and cutting-edge length in this specimen, this might be a good example of inward fluxes of high-quality flakes produced elsewhere and consumed on-site. This movement of products along different fractions of the ancient landscape might have affected some quartz cores and/or flakes, although it seems more plausible that quartz exploitation and consumption took place preferentially on-site. In sum, at EAK, hominins were undertaking casual quartz knapping activities and probably transporting some products obtained with the main goal of obtaining small usable natural cutting edges.

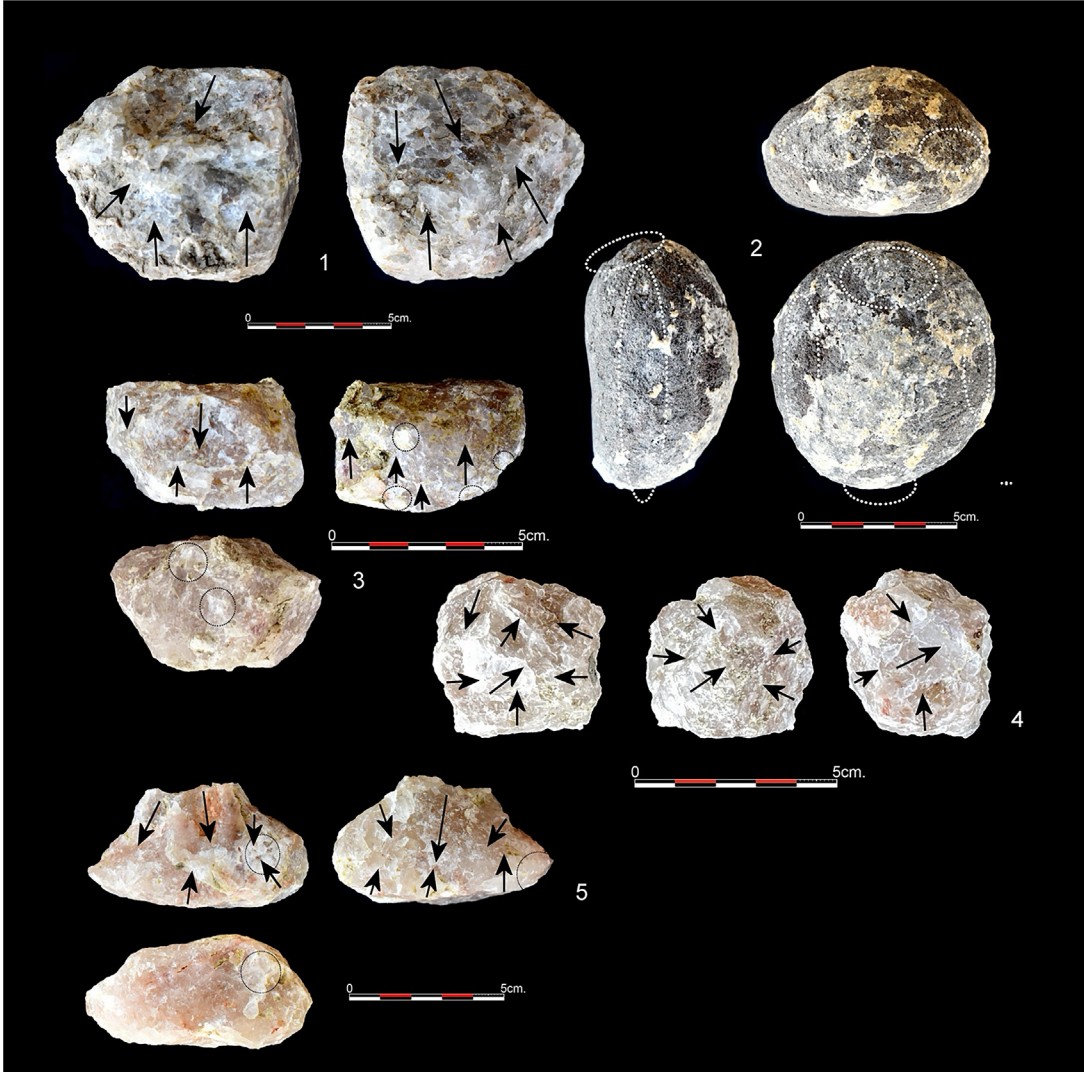

**Appendix 1—figure 26.** Lithics from EAK. Bipolar cores: 1 Cubic specimen (77 x 70 × 58 mm, 455 g) from which a medium-sized flake (47 x 63 mm) has been detached; 3. A cubic blank has been exploited in the thickness plane around the perimeter of the piece with a circular tendency (51 x 47 × 32 mm, 194 g). Frosting is observed on the basal area and on some ridges; 5. Triangular morphology and opposed detachments, showing clear frosting on the base (71 x 40 × 35 mm, 118 g). Hammerstone: 2. Oval basalt cobble (78 x 70 × 48 mm, 323 g) showing battering around the perimeter of the piece. Handheld cores: 4. Exhausted multifacial-multipolar core with a cubic shape (41 x 40 × 30 mm, 75 g), showing a minimum number of 10 negative scars crossing in multiple planes.

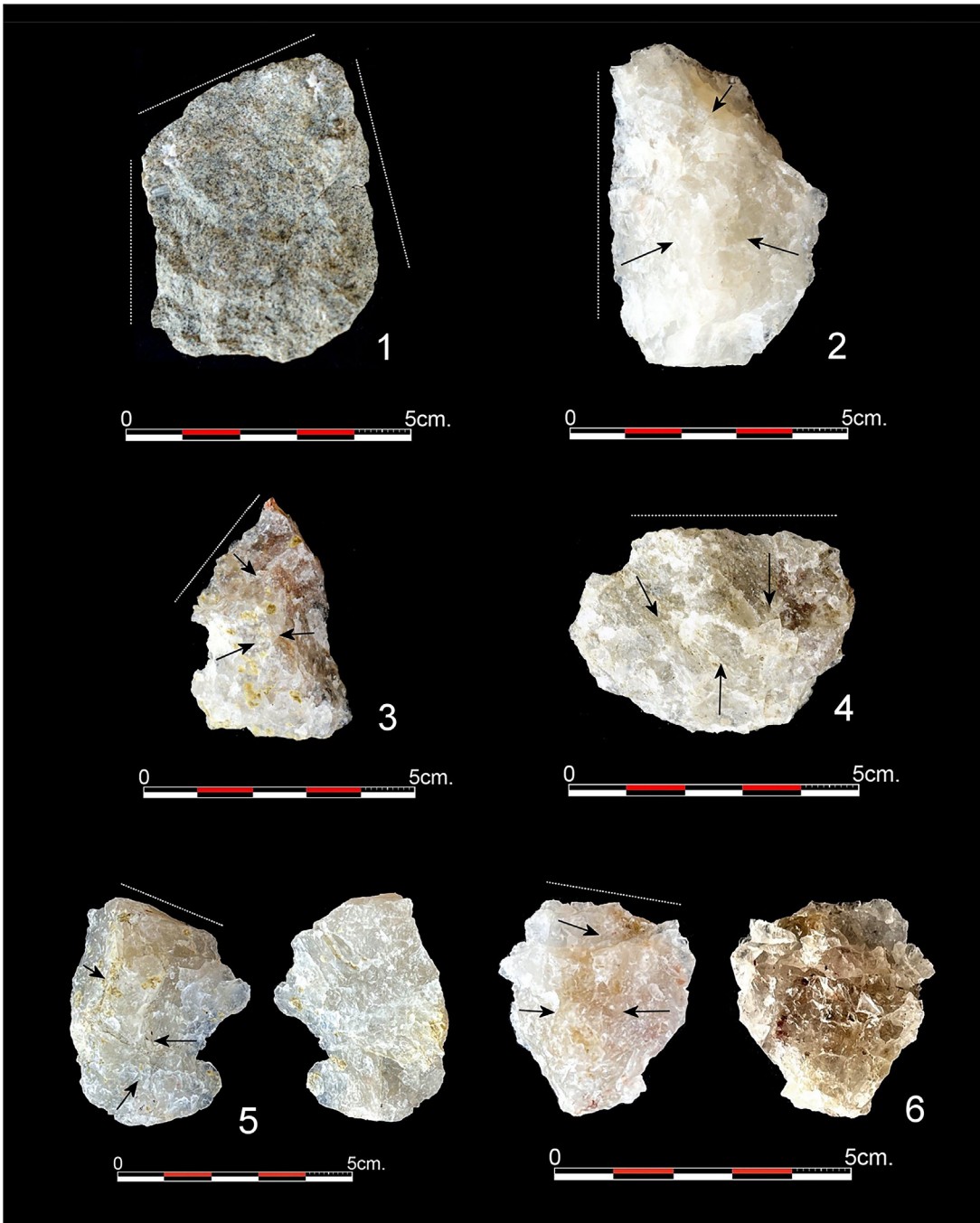

**Appendix 1—figure 27.** Flakes from EAK. Phonolite: 1. Rectangular, Type 6 specimen, unifaceted butt, one dorsal negative scar, with 92 mm of bilateral and distal cutting edge (48 x 38 × 13 mm, 29 g). Quartz: 2. Rectangular, Type 6, unifaceted platform, simple orthogonal dorsal pattern, and 50 mm lateral cutting edge (58 x 39 × 19 mm, 42 g); 3. Triangular, Type 6, unifaceted platform, simple orthogonal dorsal pattern, and 30 mm of latero-distal edge (recent notch on the left side; 43 x 28 × 9 mm, 13 g); 4. Oval, Type 6, lineal platform, opposed dorsal pattern, and 40 mm of transversal cutting edge (35 x 50 × 12 mm, 19 g); 5. Quadrangular, Type 6, unifaceted butt, orthogonal pattern, diffuse bulb on ventral face and 28 mm of distal cutting edge (48 x 37 × 14 mm, 24 g); 6. Oval, Type 6, unifaceted butt, lateral opposed dorsal pattern, and 20 mm of distal cutting edge (36 x 32 × 13 mm, 13 g).

**Appendix 1—table 2.** Distribution of the lithic specimens retrieved from EAK sorted by lithic category and raw material.

V, vesicular; B, basalt; P, phonolite; Q, quartz.

| Lithic category | Raw material | | | | Total | |
|---|---|---|---|---|---|---|
| | V | B | P | Q | n | % |
| Unmodified | | | | | 1 | 1.25 |
| Cobbles | 1 | | | | | |
| Percussion | | | | | 1 | 1.25 |
| Hemmerstones | | 1 | | | | |
| Cores | | | | | 4 | 5 |
| Handheld | | | | 1 | | |
| Bipolar | | | | 3 | | |
| Detached | | | | | 23 | 28.75 |
| Flakes | | | 1 | 14 | | |
| Broken flakes | | | | 4 | | |
| Bipolar flakes | | | | 4 | | |
| Waste | | | | | 51 | 63.75 |
| Flake fragments | | | | 14 | | |
| Debris | | | | 24 | | |
| Shatter | | | | 10 | | |
| Fragments | | | | 3 | | |
| Total n | 1 | 1 | 1 | 77 | | |
| Total % | 1.25 | 1.25 | 1.25 | 96.25 | | |

**Appendix 1—table 3.** Size (L=length, B=breadth; T=thickness) and mass (M) of complete flakes retrieved from EAK.

| | Min | Max | Mean | SD |
|---|---|---|---|---|
| L | 20 | 58 | 32.76 | 10.66 |
| B | 19 | 50 | 29.41 | 7.97 |
| T | 6 | 19 | 11.17 | 3.66 |
| M | 4 | 42 | 13.35 | 10.56 |

## Orientation of materials at EAK

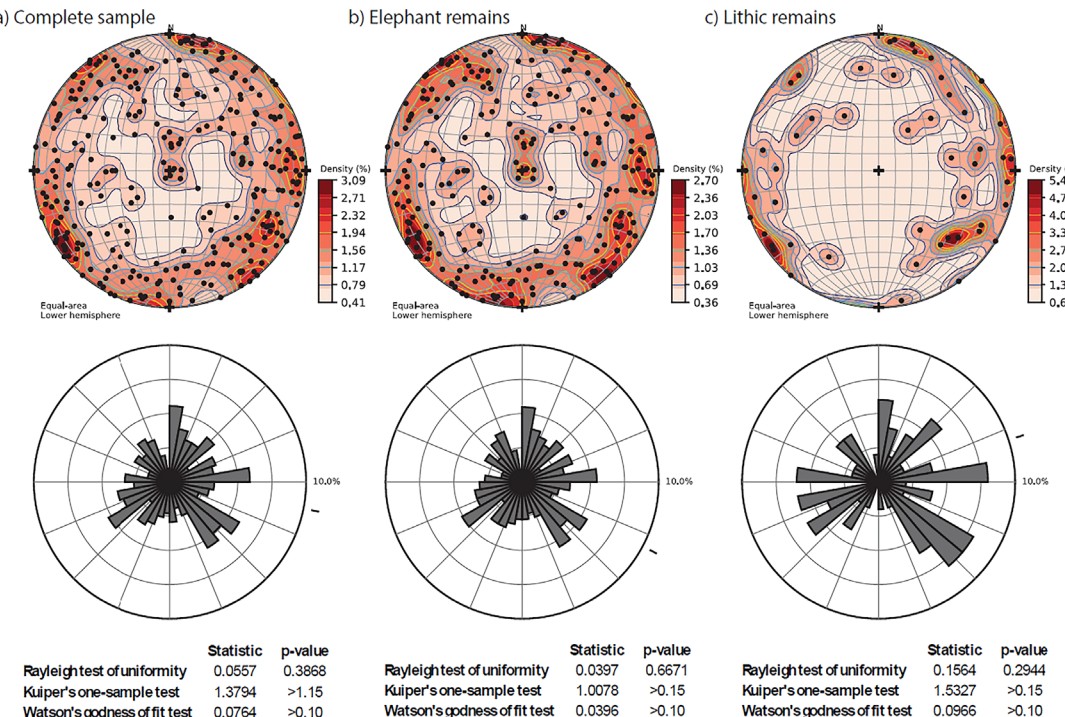

Appendix 1—figure 28. Stereograms, rose diagrams, and results obtained from the application of the Rayleigh, Kuiper, and Watson tests.

## Carnivore damage on an adult giraffe (Chobe National Park, Botswana)

In late May 2024, two of us (MDR and EB) carried out a taphonomic study of proboscidean carcasses in a National Park and a Reserve in Botswana (analysis in progress). Here we will advance some green-broken specimens (most likely caused by hyena ravaging) found on a cluster of bones belonging to an adult giraffe in Chone National Park. This is relevant to see the extent of damage that carnivores can do on megafaunal remains.

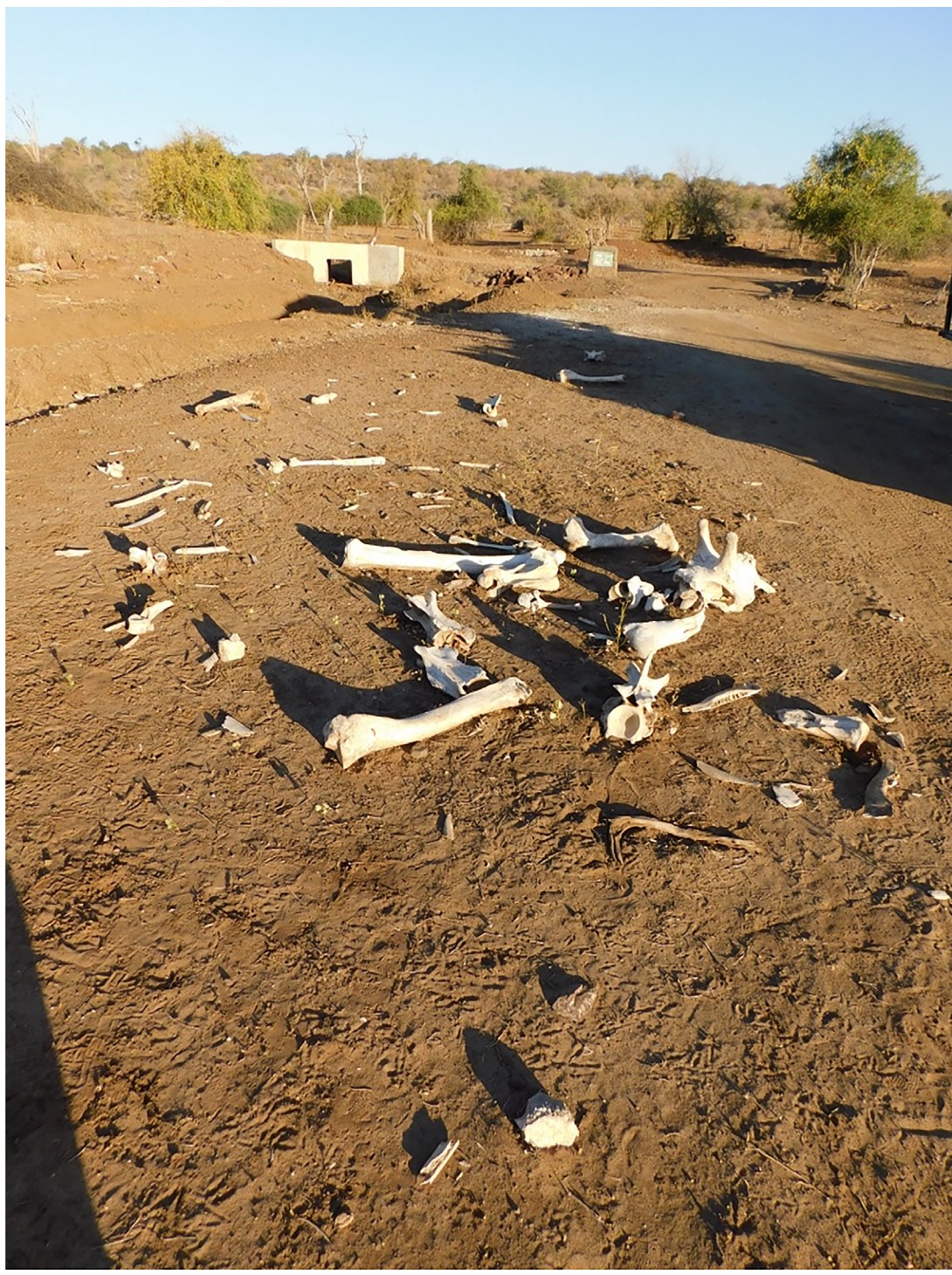

**Appendix 1—figure 29.** Cluster of hyena-ravaged bones from a giraffe at Chobe (Botswana).

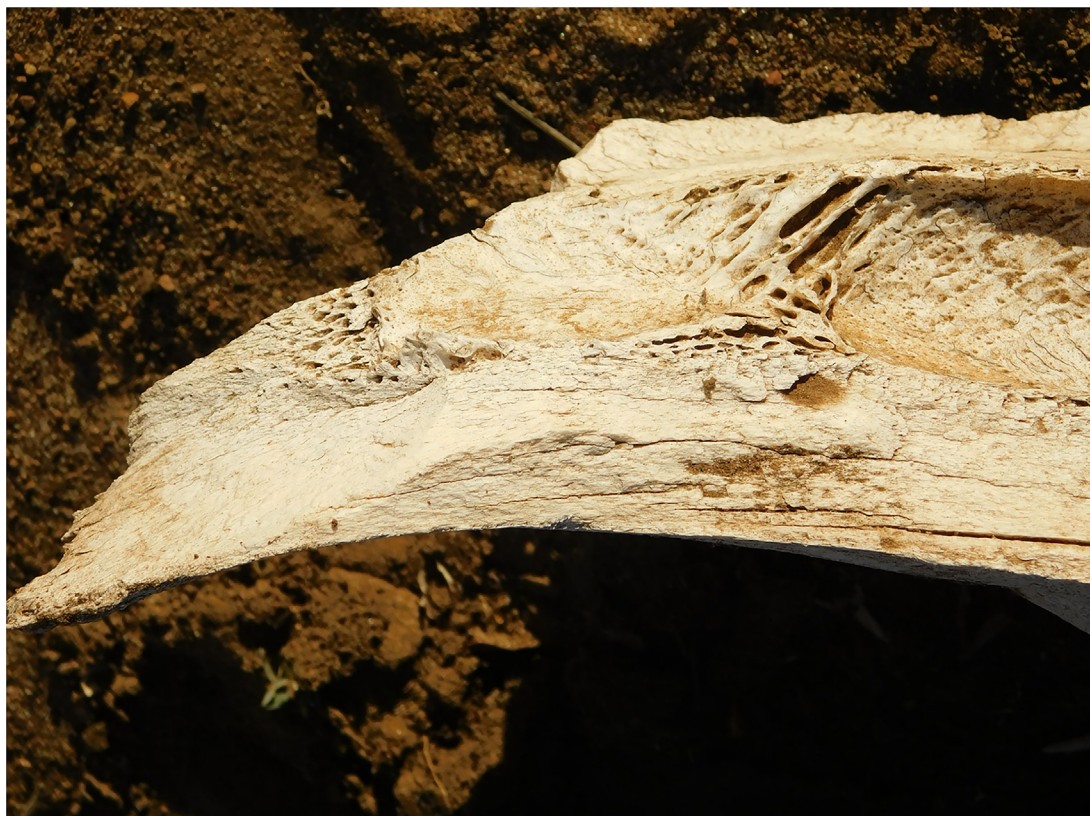

**Appendix 1—figure 30.** Spiral and green-broken planes on both sides of a giraffe humerus (broken by hyenas) from the cluster displayed in *Appendix 1—figure 22*.

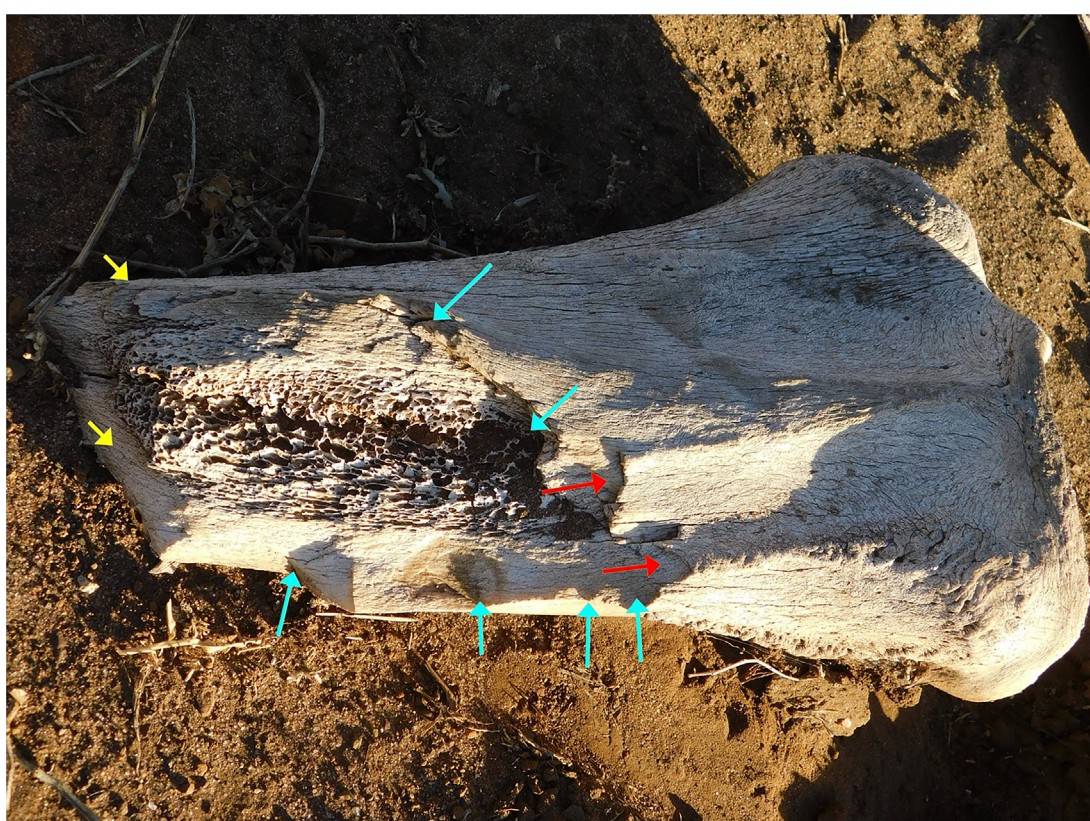

**Appendix 1—figure 31.** Giraffe scapula ravaged by hyenas displaying overlapping non-invasive notches (blue arrows), continuous green breaks (yellow arrows), and axial reflected scars parallel to the axis of the bone (also documented in hammerstone broken megafaunal limb bones; see *Appendix 1—figure 4*).

## The issue of bone tools

Here, we show some examples of additional carnivore damage on large-sized animals, displaying features that could be mistaken for anthropogenic agency.

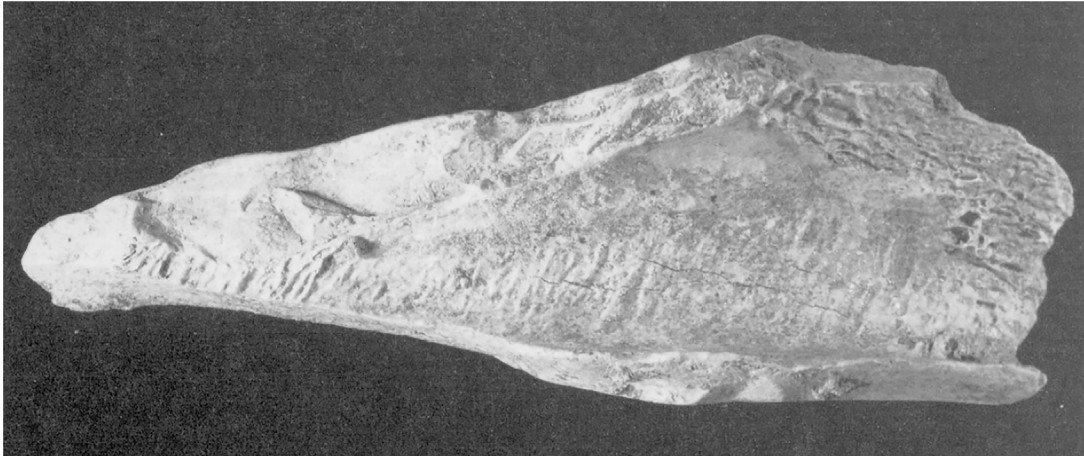

**Appendix 1—figure 32.** Example of Bos/Bison femoral shaft from the hyena den of Bois Roche (from *Villa and Bartram, 1996*, Flaked Bone from a Hyena Den, Paleo 8: 143–159). Notice the amount of scars, their overlapping and incomplete nature, and their continuous trajectory on what seems a pointed bone fragment.

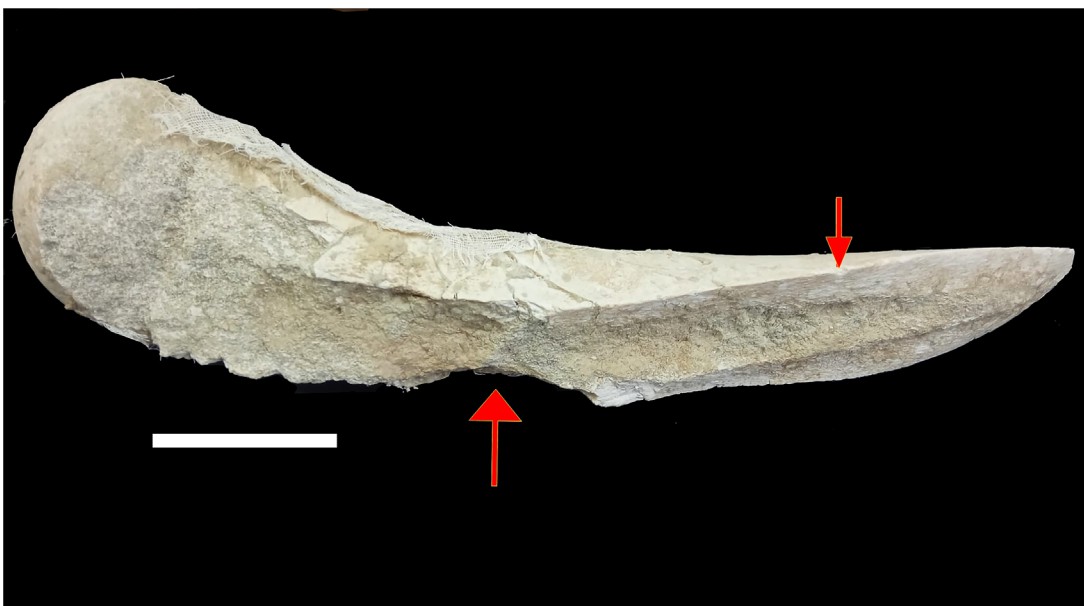

**Appendix 1—figure 33.** Example of green breakage on a proboscidean femur (Gomphoterium) from the Middle Miocene site of Virgen del Puerto (Madrid, Spain).

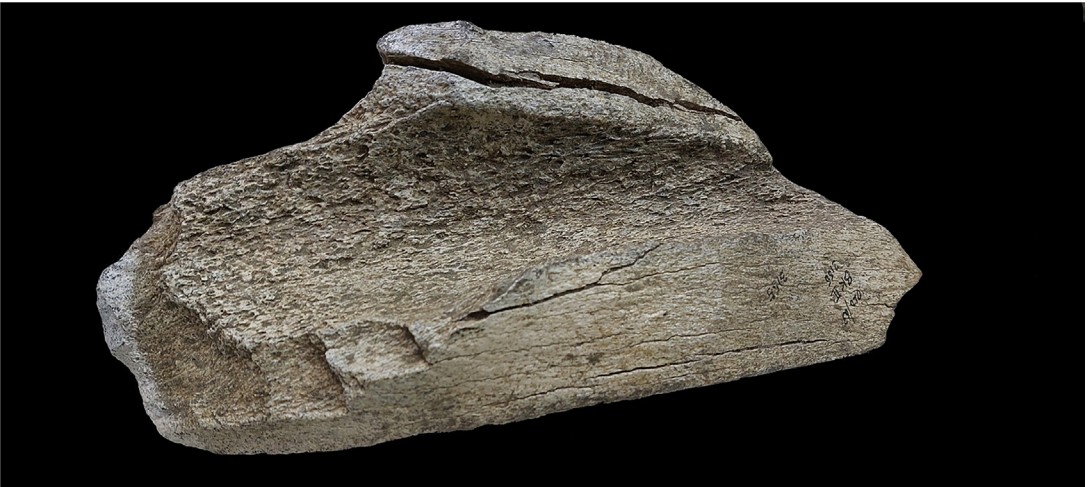

**Appendix 1—figure 34.** Example of an elephant green-broken femoral shaft fragment from BK (Olduvai, Upper Bed II) presented by Leakey as a case of bone tool. Notice the similar scar overlap and smooth green fracture surfaces resulting from bone breaking. The specimen's length is 46 cm. Currently displayed at the Olduvai Museum.

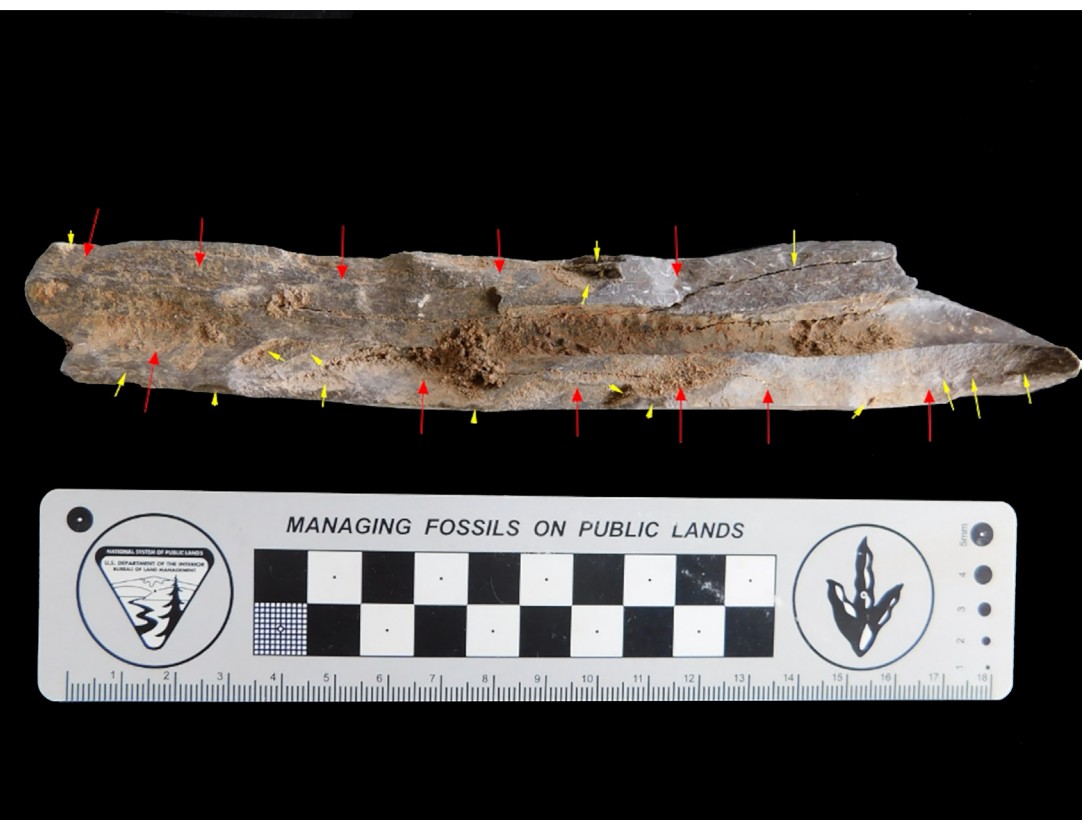

**Appendix 1—figure 35.** Tibia fragment from a large medium-sized bovid displaying multiple overlapping scars on both breakage planes inflicted by carnivore damage (red arrows). Within those scars, there are abundant reflected/incomplete smaller scars (yellow arrows). The specimen was found within the base of the LAS stratigraphic unit, where the most abundant number of megafaunal green-broken bones have been found.

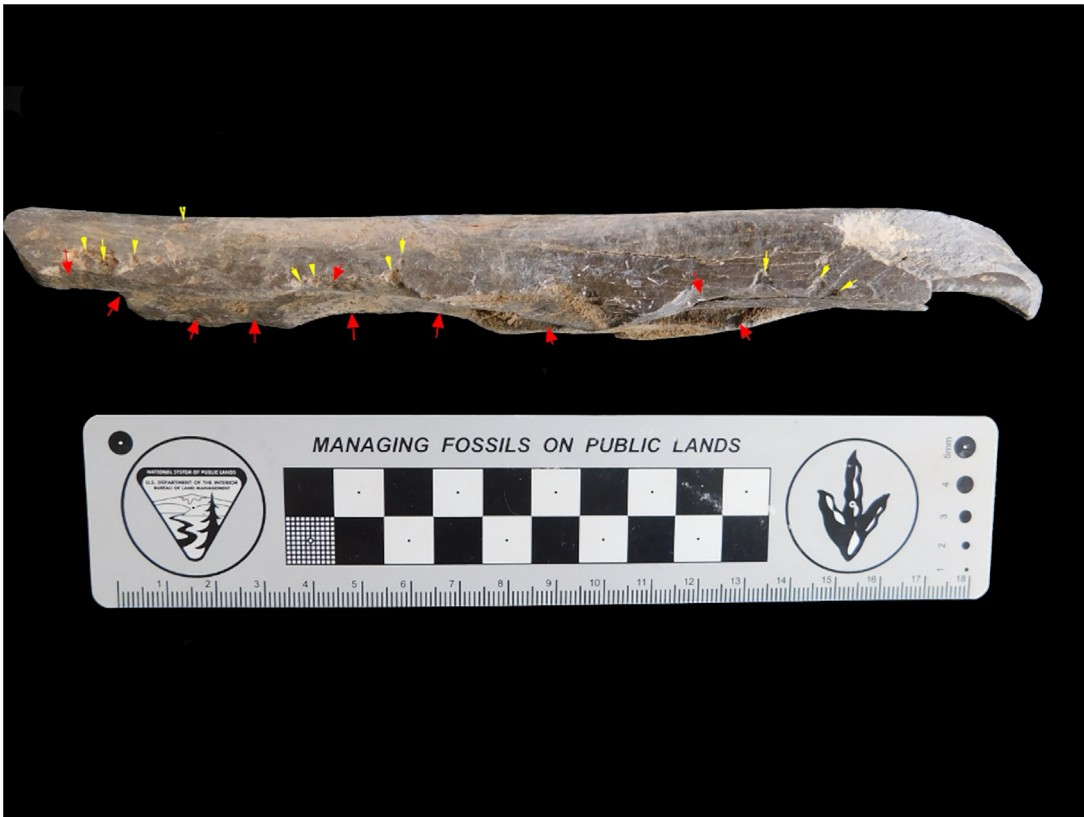

**Appendix 1—figure 36.** Same specimen as shown in *Appendix 1—figure 30*, viewed on its medial side. The profile of the overlapping scars and notches can be clearly seen (red arrows), together with traces of tooth marks (yellow arrows).

