## [Editor Report · eLife Assessment]

In this **valuable** study, the authors present traces of bone modification on ~1.8 million-year-old proboscidean remains from Tanzania, which they infer to be the earliest evidence for stone-tool-assisted megafaunal consumption by hominins. Challenging published claims, the authors argue that persistent megafaunal exploitation roughly coincided with the earliest Acheulean tools. Notwithstanding the rich descriptive and spatial data, the behavioral inferences about hominin agency rely on traces (such as bone fracture patterns and spatial overlap) that are not unequivocal; the evidence presented to support the inferences thus remains **incomplete**. Given the implications of the timing and extent of hominin consumption of nutritious and energy-dense food resources, as well as of bone toolmaking, the findings of this study will be of interest to paleoanthropologists and other evolutionary biologists.

---

## [Referee Report · Reviewer #1 (Public review)]

Domínguez-Rodrigo and colleagues make a moderately convincing case for habitual elephant butchery by Early Pleistocene hominins at Olduvai Gorge (Tanzania), ca. 1.8-1.7 million years ago. They present this at the site scale (the EAK locality, which they excavated), as well as across the penecontemporaneous landscape, analyzing a series of findspots that contain stone tools and large-mammal bones. The latter are primarily elephants, but giraffids and bovids were also butchered in a few localities. The authors claim that this is the earliest well-documented evidence for elephant butchery; doing so requires debunking other purported cases of elephant butchery in the literature, or in one case, reinterpreting elephant bone manipulation as being nutritional (fracturing to obtain marrow) rather than technological (to make bone tools). The authors' critical discussion of these cases may not be consensual, but it surely advances the scientific discourse. The authors conclude by suggesting that an evolutionary threshold was achieved at ca. 1.8 ma, whereby regular elephant consumption rich in fats and perhaps food surplus, more advanced extractive technology (the Acheulian toolkit), and larger human group size had coincided.

The fieldwork and spatial statistics methods are presented in detail and are solid and helpful, especially the excellent description (all too rare in zooarchaeology papers) of bone conservation and preservation procedures. However, the methods of the zooarchaeological and taphonomic analysis - the core of the study - are peculiarly missing. Some of these are explained along the manuscript, but not in a standard Methods paragraph with suitable references and an explicit account of how the authors recorded bone-surface modifications and the mode of bone fragmentation. This seems more of a technical omission that can be easily fixed than a true shortcoming of the study. The results are detailed and clearly presented.

By and large, the authors achieved their aims, showcasing recurring elephant butchery in 1.8-1.7 million-year-old archaeological contexts. Nevertheless, some ambiguity surrounds the evolutionary significance part. The authors emphasize the temporal and spatial correlation of (1) elephant butchery, (2) Acheulian toolkits, and (3) larger sites, but do not actually discuss how these elements may be causally related. Is it not possible that larger group size or the adoption of Acheulian technology have nothing to do with megafaunal exploitation? Alternative hypotheses exist, and at least, the authors should try to defend the causation, not just put forward the correlation. The only exception is briefly mentioning food surplus as a "significant advantage", but how exactly, in the absence of food-preservation technologies? Moreover, in a landscape full of aggressive scavengers, such excess carcass parts may become a death trap for hominins, not an advantage. I do think that demonstrating habitual butchery bears very significant implications for human evolution, but more effort should be invested in explaining how this might have worked.

Overall, this is an interesting manuscript of broad interest that presents original data and interpretations from the Early Pleistocene archaeology of Olduvai Gorge. These observations and the authors' critical review of previously published evidence are an important contribution that will form the basis for building models of Early Pleistocene hominin adaptation.

---

## [Referee Report · Reviewer #2 (Public review)]

The authors argue that the Emiliano Aguirre Korongo (EAK) assemblage from the base of Bed II at Olduvai Gorge shows systematic exploitation of elephants by hominins about 1.78 million years ago. They describe it as the earliest clear case of proboscidean butchery at Olduvai and link it to a larger behavioral shift from the Oldowan to the Acheulean.

The manuscript makes a valuable contribution to the Olduvai Gorge record, offering a detailed description of the EAK faunal assemblage. In particular, the paper provides a high-resolution record of a juvenile Elephas recki carcass, associated lithic artifacts, and several green-broken bone specimens. These data are inherently valuable and will be of significant interest to researchers studying Early Pleistocene taphonomy.

Comments on previous round of revisions:

The revised manuscript does a good job of using less definitive language, particularly by adding "possible" qualifiers to several interpretations. This addresses the concern about overstatement.

The main issue raised in the original review, however, remains unresolved. Only two elephant bone specimens at EAK show green-bone breakage interpreted as anthropogenic, and the diagnostic basis for that interpretation is not demonstrated clearly on the EAK material itself. The manuscript discusses a suite of fracture attributes described as diagnostic of dynamic percussive breakage, but these attributes are not explicitly documented on the EAK specimens. Instead, the diagnostic traits are illustrated using material from other Olduvai contexts, and that behavior is then extrapolated to make similar claims at EAK. For a paper making a potentially important behavioral argument, the key diagnostic evidence is not clearly demonstrated at the focal assemblage.

This problem is evident in the presentation of the EAK specimens. In their response, the authors state that one EAK specimen shows "overlapping scars" and constitutes a "long bone flake"; however, these features are not clearly identifiable in the figures or captions as currently presented. The authors state that Figures S21-S23 clearly indicate human agency, including a long bone flake with overlapping scars and a view of the medullary surface, but it is unclear which specimens or surfaces these descriptions refer to. Figure S21 does appear to show green fracture and is described only as an "elephant-sized flat bone fragment with green-bone curvilinear break." Figure S22 shows the same bone and cortical surface in a different orientation, providing no additional information. In Figure S23, I cannot clearly identify a medullary surface or evidence of green-bone fracture from this image. None of these images clearly demonstrates overlapping scars, and the figures would be substantially improved by explicitly identifying the features described in the text. Even if both EAK specimens are accepted as green-broken, they do not demonstrate the co-occurrence of multiple diagnostic fracture traits such as multiple green breaks, large step fractures, hackle marks, and overlapping scars that the authors state is required to attribute dynamic percussive activity to hominins and address equifinality.

I appreciate that the authors are careful to state that spatial association between stone tools and fossils alone does not demonstrate hominin behavior, and that they treat the spatial analyses as supportive rather than decisive. While the association is intriguing, the problem is downstream: spatial association is used to strengthen an interpretation of butchery at EAK that still depends on fracture evidence that is not clearly documented at the assemblage level.

The critique concerning Nyayanga is not addressed in the revision. The manuscript proposes alternative explanations for the Nyayanga material but does not demonstrate why these are more plausible than the interpretation advanced by Plummer et al. (2023). I am not arguing that the Nyayanga material should be accepted as butchery; rather, showing that trampling is possible does not establish it as more probable than cut marks. In contrast, the EAK material is treated as evidence of butchery on the basis of evidence that, in my opinion, is more limited and less clearly demonstrated. Even if this is not the authors' intention, the uneven treatment removes an earlier megafaunal case from the comparison and strengthens the case for interpreting EAK as marking a behavioral shift toward megafaunal butchery by excluding other early cases.

While I remain concerned about how the EAK evidence is documented and interpreted, I think the manuscript is appropriate for publication and will generate useful discussion. Readers can then assess for themselves whether the available evidence supports the strength of the behavioral claims.

[Editors' note: the authors are encouraged to make this version the Version of Record.]

---

## [Author Response]

The following is the authors’ response to the previous reviews

**Reviewer #2 (Public review):**
This problem is evident in the presentation of the EAK specimens. In their response, the authors state that one EAK specimen shows "overlapping scars" and constitutes a "long bone flake"; however, these features are not clearly identifiable in the figures or captions as currently presented. The authors state that Figures S21-S23 clearly indicate human agency, including a long bone flake with overlapping scars and a view of the medullary surface, but it is unclear which specimens or surfaces these descriptions refer to. Figure S21 does appear to show green fracture and is described only as an "elephant-sized flat bone fragment with green-bone curvilinear break." Figure S22 shows the same bone and cortical surface in a different orientation, providing no additional information. In Figure S23, I cannot clearly identify a medullary surface or evidence of green-bone fracture from this image. None of these images clearly demonstrates overlapping scars, and the figures would be substantially improved by explicitly identifying the features described in the text. Even if both EAK specimens are accepted as green-broken, they do not demonstrate the co-occurrence of multiple diagnostic fracture traits such as multiple green breaks, large step fractures, hackle marks, and overlapping scars that the authors state is required to attribute dynamic percussive activity to hominins and address equifinality.

We appreciate the reviewer’s careful evaluation of the EAK specimens. We acknowledge that the overlapping scars and medullary surface of the specimen originally shown in Figure S23 were not sufficiently clear. To address this, we have extensively revised Figure S23. In the updated Supplementary File, we have provided new annotations and line drawings that explicitly trace the outlines of the overlapping scars and clearly shows the green-bone fracture features. These enhancements ensure that the diagnostic traits discussed in the text are now directly identifiable in the visual record. This demonstrates the co-occurrence of traits: green-broken outlines and overlapping scars, which meet the criteria for identifying dynamic percussive activity. This is so following Reviewer´s 2 partial handling of our arguments; since we argued in our previous response that clear simple green-broken elephant long limb bones were an anthropogenic signature per se, given that currently no durophagous predator/scavenger (including spotted hyenas) are able to produce them. Additional secondary features like hackle marks are supportive but not necessary to attribute human agency.

I appreciate that the authors are careful to state that spatial association between stone tools and fossils alone does not demonstrate hominin behavior, and that they treat the spatial analyses as supportive rather than decisive. While the association is intriguing, the problem is downstream: spatial association is used to strengthen an interpretation of butchery at EAK that still depends on fracture evidence that is not clearly documented at the assemblage level.

The association is inferred (not demonstrated) by the strong statistical spatial association between lithics and bones. Additional taphonomic evidence (like cut marks or green-broken bones) do further support the inference but they do not demonstrate it, given the highly subjective nature of cut mark identification and the plethora of alternative scenarios: one green-broken bone would not demonstrate complete elephant butchery (it could result from a marginal exploitation of just that bone); one cutmarked bone could equally reflect several alternative access types to the remains. The reviewer recognized above the presence of green-broken elements at EAK; again, this supports anthropogenic agency better than any other alternative scenario, because one of the green-broken bones is a long bone and modern hyenas are not able to produce this kind of specimens.

The critique concerning Nyayanga is not addressed in the revision. The manuscript proposes alternative explanations for the Nyayanga material but does not demonstrate why these are more plausible than the interpretation advanced by Plummer et al. (2023). I am not arguing that the Nyayanga material should be accepted as butchery; rather, showing that trampling is possible does not establish it as more probable than cut marks. In contrast, the EAK material is treated as evidence of butchery on the basis of evidence that, in my opinion, is more limited and less clearly demonstrated. Even if this is not the authors' intention, the uneven treatment removes an earlier megafaunal case from the comparison and strengthens the case for interpreting EAK as marking a behavioral shift toward megafaunal butchery by excluding other early cases.

Again, it was never our intention to “demonstrate” anything. The reviewer is misusing this term. These types of arguments are epistemologically impossible to demonstrate. One can just discuss the heuristics of alternative scenarios. The point that we tried to make was that the Nyayanga purported cut marks on megafaunal remains are (as identified and published) impossible to differentiate from natural sedimentary abrasive marks (like trampling). Therefore, they cannot be argued to represent anthropogenic butchery on a secure basis. Especially, when they do not occur in conjunction with green-broken elements of clear dynamic loading nature.